# Provably Efficient RL under Episode-Wise Safety in Constrained MDPs with Linear Function Approximation

**Toshinori Kitamura**
The University of Tokyo
Tokyo, Japan
toshinori-k@weblab.t.u-tokyo.ac.jp

**Arnob Ghosh**
New Jersey Institute of Technology
New Jersey, USA

**Tadashi Kozuno**
OMRON SINIC X, NexaScience
Tokyo, Japan

**Wataru Kumagai**
OMRON SINIC X, NexaScience
Tokyo, Japan

**Kazumi Kasaura**
OMRON SINIC X
Tokyo, Japan

**Kenta Hoshino**[*]
Kyoto University
Kyoto, Japan

**Yohei Hosoe**
Kyoto University
Kyoto, Japan

**Yutaka Matsuo**
The University of Tokyo
Tokyo, Japan

## Abstract

We study the reinforcement learning (RL) problem in a constrained Markov decision process (CMDP), where an agent explores the environment to maximize the expected cumulative reward while satisfying a single constraint on the expected total utility value in every episode. While this problem is well understood in the tabular setting, theoretical results for function approximation remain scarce. This paper closes the gap by proposing an RL algorithm for linear CMDPs that achieves $\widetilde{\mathcal{O}}(\sqrt{K})$ regret with an *episode-wise* zero-violation guarantee. Furthermore, our method is computationally efficient, scaling polynomially with problem-dependent parameters while remaining independent of the state space size. Our results significantly improve upon recent linear CMDP algorithms, which either violate the constraint or incur exponential computational costs.

## 1 Introduction

Safe decision-making is essential in real-world applications such as plant control and finance [19]. Constrained Markov decision process (CMDP) is a mathematical framework for developing decision-making algorithms with formal safety guarantees [2]. This paper studies the reinforcement learning (RL) problem in finite-horizon CMDPs, where an agent explores the environment to maximize the expected cumulative rewards while satisfying a single constraint on the expected total utility value.

Safe exploration has been established in the tabular CMDP settings. Several works [27, 8, 47] achieve **episode-wise** zero-violation RL with $\widetilde{\mathcal{O}}(\sqrt{K})$ regret for $K$ number of episodes, ensuring constraint satisfaction in every episode. Their approach consists of two phases: deploying a known strictly safe policy $\pi^{\mathrm{sf}}$ and then updating policies via linear programs (LPs), which optimizes an optimistic objective while satisfying a pessimistic constraint. Deploying $\pi^{\mathrm{sf}}$ is necessary to ensure feasible solutions for the LPs once enough environmental information is collected.

---

[*]Current affiliation: Institute of Science Tokyo, Tokyo, Japan, and DENSO IT Laboratory, Tokyo, Japan.

39th Conference on Neural Information Processing Systems (NeurIPS 2025).

Table 1: Representative CMDP results with $K$-dependent regrets. See Section 3 for CMDP notations.

| | Paper | Epi.-Wise Safe? | Comp. Efficient? | Regret |
|---|---|---|---|---|
| Tabular | Yu et al. [47] | Yes | $|\mathcal{S}|$ dependent | $\widetilde{\mathcal{O}}(\xi^{-1}\sqrt{|\mathcal{S}|^2|\mathcal{A}|H^5 K})$ |
| Linear | Ghosh et al. [18] | No | No | $\widetilde{\mathcal{O}}(\sqrt{d^3 H^4 K})$ |
| | Roknilamouki et al. [36] | Instantaneous | Yes | $\widetilde{\mathcal{O}}(\xi^{-1}\sqrt{d^3 H^4 K})$ |
| | **OPSE-LCMDP (Ours)** | Yes | Yes | $\widetilde{\mathcal{O}}(\xi^{-1}\sqrt{d^5 H^8 K})$ |
| | Bandit lower bound [32] | Yes | N/A | $\widetilde{\Omega}(\max\{\sqrt{|\mathcal{A}|K}, \xi^{-2}\})^*$ |

\* The regrets of [47] and ours include an additional $\widetilde{\mathcal{O}}(\xi^{-2})$ constant. We omit them from the table due to space limitations. [36, 32, 4] avoid the constant by assuming access to a safe action vector, but they are limited to instantaneous constraints (see Section 2.1). Including ours, existing safe algorithms [27, 8, 47, 36, 3, 32, 22] incur $\widetilde{\mathcal{O}}(\xi^{-1}\sqrt{K})$ regret. Whether this can be improved to $\widetilde{\mathcal{O}}(\sqrt{K})$ remains open.

While safe exploration is well-established in tabular CMDPs, extending it to large-scale CMDPs remains a major challenge. LP-based methods are impractical at scale due to their state-dependent computational cost.[2] As a result, even in linear CMDPs, where value functions have linear structure, episode-wise safe RL has not been achieved. The state-of-the-art linear CMDP algorithm [18], which achieves the $\widetilde{\mathcal{O}}(\sqrt{K})$ violation regret,[3] incurs an exponential computational cost of $K^H$, where $H$ is the horizon. Several studies achieve safe RL under instantaneous constraints [4, 36],[4] a special sub-class of the episode-wise safety that can be overly conservative (e.g., in drone control, temporary high energy consumption is tolerable, but full battery depletion is not). Table 1 summarizes representative algorithms, with additional literature in Appendix B. In short, a fundamental open question remains:

*Can we develop a computationally efficient[5] linear CMDP algorithm with sublinear regret and zero episode-wise constraint violation?*

**Contributions.** We propose **O**ptimistic-**P**essimistic **S**oftmax **E**xploration for **L**inear **CMDP** (OPSE-LCMDP), the first algorithm for linear CMDPs that achieves $\widetilde{\mathcal{O}}(\sqrt{K})$**-regret and episode-wise safety**. Our approach builds on the optimistic-pessimistic exploration framework with two key innovations for large-scale state-space problems: (i) a new **deployment rule for $\pi^{\mathbf{sf}}$**, and (ii) a **computationally efficient** method to implement optimism for the objective and pessimism for the constraint within the softmax policy framework [16, 18].

Section 2 first analyzes the linear constrained bandit problem as a "warm-up" for linear CMDPs ($H = 1$ with an expected instantaneous constraint), highlighting the key role of the $\pi^{\mathbf{sf}}$ deployment rule in avoiding linear regret. Previous instantaneous constraint literature limits $\pi^{\mathbf{sf}}$ deployments by representing the safe action as a vector $\mathbf{a}^{\mathbf{sf}} \in \mathbb{R}^d$ [32, 33, 22, 3, 4]. However, extending this approach to episode-wise safety is non-trivial, as the constraint is imposed on policies rather than actions, and policies may be nonlinear functions (e.g., softmax mapping from value functions) rather than single vectors. We overcome this challenge by showing that **if $\pi^{\mathbf{sf}}$ is deployed only when the agent is less confident in $\pi^{\mathbf{sf}}$'s safety**, the number of deployments is logarithmically bounded (Theorem 1).

Section 3 then extends the bandit result to RL in CMDPs. To enable optimistic-pessimistic exploration in linear CMDPs, OPSE-LCMDP employs the **composite softmax policy** (Definition 3), which adjusts optimism and pessimism by controlling a variable $\lambda$. OPSE-LCMDP efficiently searches for the best $\lambda$ through **bisection search**, achieving a **polynomial computational cost** in problem parameters, independent of state-space cardinality (Remark 2). Overall, our techniques—the novel $\pi^{\mathbf{sf}}$ deployment rule and softmax-based optimistic-pessimistic exploration—achieve the first episode-wise safe RL with sublinear regret and computational efficiency in linear CMDPs.

**Mathematical notations.** The set of probability distributions over a set $\mathcal{S}$ is denoted by $\mathscr{P}(\mathcal{S})$. For integers $a \le b$, let $[\![a, b]\!] := \{a, \dots, b\}$, and $[\![a, b]\!] := \emptyset$ if $a > b$. For $\mathbf{x} \in \mathbb{R}^N$, its $n$-th element is $\mathbf{x}_n$ or $\mathbf{x}(n)$. The clipping function $\mathrm{clip}\{\mathbf{x}, a, b\}$ returns $\mathbf{x}'$ with $\mathbf{x}'_i = \min\{\max\{\mathbf{x}_i, a\}, b\}$ for each $i$. We define $\mathbf{0} := (0, \dots, 0)^\top$ and $\mathbf{1} := (1, \dots, 1)^\top$, with dimensions inferred from the context. For a positive definite matrix $\mathbf{A} \in \mathbb{R}^{d \times d}$ and $\mathbf{x} \in \mathbb{R}^d$, we denote $\|\mathbf{x}\|_{\mathbf{A}} = \sqrt{\mathbf{x}^\top \mathbf{A} \mathbf{x}}$. For positive sequences $\{a_n\}$ and $\{b_n\}$ with $n = 1, 2, \dots$, we write $a_n = O(b_n)$ if there exist $C > 0$ and $N \in \mathbb{N}$

---

[2] While some works (e.g., 30) proposed LP methods for unconstrained linear MDPs, they remain unsuitable for our exploration setting or still incur state-dependent computational costs (see Appendix B).

[3] Violation regret denotes the total amount of constraint violation during exploration.

[4] Inst. constraint requires $u_h(s_h^{(k)}, a_h^{(k)}) \ge b \; \forall h, k \in [\![1, H]\!] \times [\![1, K]\!]$ (see Section 3 for notations).

[5] An algorithm is comp. efficient if its cost is polynomial with problem parameters, excluding the state space.

such that $a_n \leq C b_n$ for all $n \geq N$, and $a_n = \Omega(b_n)$ for the reverse inequality. We use $\widetilde{O}(\cdot)$ and $\widetilde{\Omega}(\cdot)$ to further hide the polylogarithmic factors. Finally, for $\mathbf{x} \in \mathbb{R}^d$, we denote its softmax distribution as $\mathrm{SoftMax}(\mathbf{x}) \in \mathscr{P}(\llbracket 1, d \rrbracket)$ with its $i$-th component $\mathrm{SoftMax}(\mathbf{x})_i = \exp(\mathbf{x}_i)/(\sum_i \exp(\mathbf{x}_i))$.

## 2 Warm Up: Safe Exploration in Linear Constrained Bandit

To better illustrate the core ideas of our CMDP algorithm, this section introduces a contextual linear bandit variant based on Pacchiano et al. [32]. All the proofs in this section are provided in Appendix D. Let $\mathcal{A} \subset \mathbb{R}^d$ be the action space, a compact set of bounded $d$-dimensional vectors. Without loss of generality, we assume $\|\mathbf{a}\|_2 \leq 1$ for any $\mathbf{a} \in \mathcal{A}$. At each round $k$, the agent selects a policy $\pi^{(k)} \in \mathscr{P}(\mathcal{A})$, samples an action $\mathbf{a}^{(k)} \sim \pi^{(k)}$, and observes the reward $r^{(k)} = \boldsymbol{\theta}_r^\top \mathbf{a}^{(k)} + \varepsilon_r^{(k)}$ and utility $u^{(k)} = \boldsymbol{\theta}_u^\top \mathbf{a}^{(k)} + \varepsilon_u^{(k)}$. Here, $\boldsymbol{\theta}_r, \boldsymbol{\theta}_u \in \mathbb{R}^d$ are vectors unknown to the agent such that $\|\boldsymbol{\theta}_r\|_2, \|\boldsymbol{\theta}_u\|_2 \leq B$, and $\varepsilon_r^{(k)}, \varepsilon_u^{(k)}$ are $R$-sub-Gaussian random noises. For any policy $\pi$ and $g \in \{r, u\}$, let $g_\pi := \mathbb{E}_{\mathbf{a} \sim \pi}[\langle \boldsymbol{\theta}_g, \mathbf{a} \rangle]$. We consider a constraint such that the expected utility must be above the threshold $b \in \mathbb{R}$. Formally, let $\Pi^{\mathrm{sf}} := \{\pi \mid u_\pi \geq b\}$ denote the set of safe policies. The agent's goal is to achieve sublinear regret while satisfying the expected **instantaneous** constraints defined as follows:

$$\mathrm{Regret}(K) := \sum_{k=1}^K r_{\pi^\star} - r_{\pi^{(k)}} = o(K) \text{ such that } \pi^{(k)} \in \Pi^{\mathrm{sf}} \; \forall k \in \llbracket 1, K \rrbracket , \tag{1}$$

where $\pi^\star \in \arg\max_{\pi \in \Pi^{\mathrm{sf}}} r_\pi$. A sublinear regret exploration is efficient, as its averaged reward approaches the optimal value, i.e., $\lim_{K \to \infty} \frac{1}{K} r_{\pi^{(K)}} \to r_{\pi^\star}$. Finally, we assume access to a strictly safe policy in $\Pi^{\mathrm{sf}}$, as deploying arbitrary policies without this assumption risks violating constraints[6].

**Assumption 1** (Safe policy). We have access to $\pi^{\mathrm{sf}} \in \Pi^{\mathrm{sf}}$ and $\xi > 0$ such that $u_{\pi^{\mathrm{sf}}} - b \geq \xi$ .

### 2.1 Technical Challenge: Zero-Violation with a Safe Policy

The key to efficient and safe exploration is the **optimistic-pessimistic** exploration, which constructs an optimistic reward $\bar{r}_\pi^{(k)} \geq r_\pi$ and a pessimistic utility $\underline{u}_\pi^{(k)} \leq u_\pi$, and then computes a policy by:

$$\max_{\pi \in \mathscr{P}(\mathcal{A})} \bar{r}_\pi^{(k)} \quad \text{such that} \quad \underline{u}_\pi^{(k)} \geq b . \tag{2}$$

Here, $\bar{r}_\pi^{(k)}$ and $\underline{u}_\pi^{(k)}$ are designed to quickly approach $r_\pi$ and $u_\pi$ as data accumulates for efficient exploration [1]. However, although Equation (2) can have feasible solutions when $\underline{u}_\pi \approx u_\pi$, the pessimistic constraint may not have any feasible solution in the early stages of exploration.

To ensure that (2) always has a solution, a common bandit approach assumes access to a safe action $\mathbf{a}^{\mathrm{sf}} \in \mathcal{A}$ such that $\boldsymbol{\theta}_u^\top \mathbf{a}^{\mathrm{sf}} \geq b + \xi$, and then ensures the feasibility of (2) by leveraging the **vector representation** of $\mathbf{a}^{\mathrm{sf}} \in \mathbb{R}^d$. For example, [32, 33, 3] designed $\underline{u}_\pi^{(k)}$ using the orthogonal direction $(\mathbf{a}^{\mathrm{sf}})^\perp := \mathbf{a}^{\mathrm{sf}} - \mathbf{a}^{\mathrm{sf}}/\|\mathbf{a}^{\mathrm{sf}}\|_2$, while [22] assume $\mathbf{a}^{\mathrm{sf}} = \mathbf{0} \in \mathcal{A}$ with a negative constraint threshold $b < 0$. Both approaches ensure that a policy playing $\mathbf{a}^{\mathrm{sf}}$ with probability 1 is always feasible in (2).

However, extending this safe action technique to episode-wise safe RL is non-trivial, as the episode-wise constraint is imposed on policies rather than actions, and policies in linear CMDPs may be nonlinear functions (e.g., softmax mappings from value functions) rather than single vectors. To address this challenge, we first develop a safe bandit algorithm without relying on safe action techniques.

### 2.2 Algorithm and Analysis

We summarize the proposed **O**ptimistic-**P**essimistic **L**inear **B**andit with **S**afe **P**olicy (OPLB-SP) in Algorithm 1, which follows the standard linear bandit framework (see Abbasi-Yadkori et al. [1]). Throughout this section, we analyze Algorithm 1 under the parameters listed in its **Input** line. Let $\widehat{\boldsymbol{\theta}}_r^{(k)} := (\boldsymbol{\Lambda}^{(k)})^{-1} \sum_{i=1}^{k-1} \mathbf{a}^{(i)} r^{(i)}$ and $\widehat{\boldsymbol{\theta}}_u^{(k)} := (\boldsymbol{\Lambda}^{(k)})^{-1} \sum_{i=1}^{k-1} \mathbf{a}^{(i)} u^{(i)}$ denote the regularized least-squares estimates of $\boldsymbol{\theta}_r$ and $\boldsymbol{\theta}_u$, respectively, where $\boldsymbol{\Lambda}^{(k)} := \rho \mathbf{I} + \sum_{i=1}^{k-1} \mathbf{a}^{(i)} (\mathbf{a}^{(i)})^\top$. Let $\widehat{r}_\pi^{(k)} := \mathbb{E}_{\mathbf{a} \sim \pi}[\mathbf{a}^\top \widehat{\boldsymbol{\theta}}_r^{(k)}]$ and $\widehat{u}_\pi^{(k)} := \mathbb{E}_{\mathbf{a} \sim \pi}[\mathbf{a}^\top \widehat{\boldsymbol{\theta}}_u^{(k)}]$ be the estimated reward and utility functions.

---

[6]The knowledge of $\xi$ is for simplicity. If unknown, we can estimate it by deploying $\pi^{\mathrm{sf}}$ with a little overhead.

---

**Algorithm 1:** Optimistic-Pessimistic Linear Bandit with Safe Policy

---

**Input:** Regression coefficient $\rho = 1$, bonus scalers $C_u = B + R\sqrt{d \ln 4K\delta^{-1}}$ and
$\quad\quad C_r = C_u(1 + 2B\xi^{-1})$, safe policy $\pi^{\mathrm{sf}}$, and iteration length $K \in \mathbb{N}$

1 **for** $k = 1, \ldots, K$ **do**
2 $\quad$ Let $\beta_\pi^{(k)}, \widehat{r}_\pi^{(k)}$, and $\widehat{u}_\pi^{(k)}$ be bonus, estimated reward and utility, respectively (see Section 2.2)
3 $\quad$ **if** $C_u\beta_{\pi^{\mathrm{sf}}}^{(k)} > \frac{\xi}{2}$ **then** $\pi^{(k)} := \pi^{\mathrm{sf}}$ $\quad\quad\quad$ /* Deploy $\pi^{\mathrm{sf}}$ if $\pi^{\mathrm{sf}}$ is unconfident */
4 $\quad$ **else** $\pi^{(k)} \in \arg\max_{\pi \in \mathscr{P}(\mathcal{A})} \widehat{r}_\pi^{(k)} + C_r\beta_\pi^{(k)}$ such that $\widehat{u}_\pi^{(k)} - C_u\beta_\pi^{(k)} \geq b$
5 $\quad$ Sample an action $\mathbf{a}^{(k)} \sim \pi^{(k)}$ and observe reward $r^{(k)}$ and utility $u^{(k)}$.

---

Using the bonus function $\beta_\pi^{(k)} := \mathbb{E}_{\mathbf{a} \sim \pi}\|\mathbf{a}\|_{\left(\mathbf{\Lambda}^{(k)}\right)^{-1}}$, with the well-established elliptical confidence bound argument for linear bandits [1], the following confidence bounds hold:

**Lemma 1** (Confidence bounds). *For any $\pi$ and $k$, with probability (w.p.) at least $1 - \delta$,*

$$r_\pi + 2C_u\beta_\pi^{(k)} \geq \widehat{r}_\pi^{(k)} + C_u\beta_\pi^{(k)} \geq r_\pi \quad and \quad u_\pi \geq \widehat{u}_\pi^{(k)} - C_u\beta_\pi^{(k)} \geq u_\pi - 2C_u\beta_\pi^{(k)} \; .$$

Based on Lemma 1, Algorithm 1 solves the following optimistic-pessimistic (Opt-Pes) problem. The optimistic objective promotes efficient exploration, while the pessimism enforces the constraint satisfaction:

**Opt-Pes (Line 4)** $\quad \pi^{(k)} \in \arg\max_{\pi \in \mathscr{P}(\mathcal{A})} \widehat{r}_\pi^{(k)} + C_r\beta_\pi^{(k)}$ such that $\widehat{u}_\pi^{(k)} - C_u\beta_\pi^{(k)} \geq b \; .$ $\quad$ (3)

This is a convex optimization problem when the set $\mathcal{A}$ satisfies certain structural assumptions, such as being discrete or ellipsoidal (see, e.g., Section 19.3 of Lattimore and Szepesvari [26]). To emphasize our approach to the technical challenge in Section 2.1, this section omits the computational details of (3) and focuses instead on the core technique for efficient exploration under episode-wise safety.

### 2.2.1 Zero-Violation and Logarithmic Number of $\pi^{\mathrm{sf}}$ Deployments

Since $\pi^{(k)}$ is either $\pi^{\mathrm{sf}}$ or the solution to Opt-Pes (if feasible), all deployed policies in Algorithm 1 satisfy the constraint with high probability due to the pessimistic constraint. However, as noted in Section 2.1, the pessimistic constraint may render Opt-Pes infeasible, requiring Line 4 to wait until the bonus $\beta_\pi^{(k)}$ shrinks sufficiently. Yet, waiting too long overuses the suboptimal $\pi^{\mathrm{sf}}$, leading to poor regret. Thus, exploration must keep the number of iterations where Equation (2) is infeasible bounded.

The core technique of Algorithm 1 lies in the $\pi^{\mathbf{sf}}$ **deployment trigger** based on the confidence of $\pi^{\mathrm{sf}}$. Specifically, we solve the optimistic-pessimistic optimization whenever $\beta_{\pi^{\mathrm{sf}}}^{(k)} \leq \frac{\xi}{2C_u}$; otherwise, we correct the data by deploying $\pi^{\mathrm{sf}}$ (see Line 3). Under this trigger, the following Theorem 1 ensures that the number of $\pi^{\mathrm{sf}}$ deployments grows **logarithmically** with the iteration length $K$.

**Definition 1** ($\pi^{\mathrm{sf}}$ unconfident iterations). Let $\mathcal{U}$ be the set of iterations when Algorithm 1 is unconfident in $\pi^{\mathrm{sf}}$, i.e., $\mathcal{U} := \{k \in [\![1, K]\!] \mid \beta_{\pi^{\mathrm{sf}}}^{(k)} > \xi/(2C_u)\}$. Let $\mathcal{U}^{\complement} := [\![1, K]\!] \setminus \mathcal{U}$ be its complement.

**Theorem 1** (Logarithmic $|\mathcal{U}|$ bound). *It holds w.p. at least $1 - \delta$ that $|\mathcal{U}| \leq \mathcal{O}\big(dC_u^2\xi^{-2}\ln\big(K\delta^{-1}\big)\big)$.*

The proof utilizes the well-known elliptical potential lemma [1]. Intuitively, it ensures that the confidence bounds shrink on average, thereby limiting the number of iterations where the algorithm remains unconfident in $\pi^{\mathrm{sf}}$. He et al. [21], Zhang et al. [50] employed a similar technique in linear bandits to ensure the suboptimality of policies after sufficient iterations.

Moreover, combined with Lemma 1, the following Lemma 2 ensures that, after logarithmic iterations, **policies around $\pi^{\mathbf{sf}}$ will become feasible solutions to Opt-Pes and Line 4**.

**Lemma 2** (Mixture policy feasibility). *Consider $k \in \mathcal{U}^{\complement}$. Let $\alpha^{(k)} := \frac{\xi - 2C_u\beta_{\pi^{\mathrm{sf}}}^{(k)}}{\xi - 2C_u\beta_{\pi^{\mathrm{sf}}}^{(k)} + 2C_u\beta_{\pi^\star}^{(k)}}$. For any*

$\alpha \in \big[0, \alpha^{(k)}\big]$, *the mixture policy $\pi_\alpha := (1 - \alpha)\pi^{\mathrm{sf}} + \alpha\pi^\star$ satisfies $u_{\pi_\alpha} - 2C_u\beta_{\pi_\alpha}^{(k)} \geq b$.*

Note that the mixture policy $\pi_\alpha$ is introduced only to ensure the feasibility of Opt-Pes; it does not need to be computed in the algorithm.

Finally, Lemma 1 and Lemma 2 directly imply the following zero-violation guarantee:

**Corollary 1** (Zero-violation). *W.p. at least $1 - \delta$, Algorithm 1 satisfies $\pi^{(k)} \in \Pi^{\mathrm{sf}}$ for any $k$.*

### 2.2.2 Regret Analysis

The remaining task is to ensure sublinear regret. By Theorem 1 and Lemma 1, the regret is decomposed as:

$$\mathrm{Regret}(K) \leq \widetilde{\mathcal{O}}\big(dBC_u^2\xi^{-2}\big) + \underbrace{3C_r \sum_{k \in \mathcal{U}^{\complement}} \beta_\pi^{(k)}}_{\textcircled{1}} + \underbrace{\sum_{k \in \mathcal{U}^{\complement}} \left(r_{\pi^\star} - \widehat{r}_{\pi^{(k)}}^{(k)} - C_r\beta_\pi^{(k)}\right)}_{\textcircled{2}}.$$

Using the elliptical potential lemma [1], we can bound $\textcircled{1} \leq \widetilde{\mathcal{O}}(C_r\sqrt{dK})$.

For the term $\textcircled{2}$, when there is no constraint in Opt-Pes, the common strategy is bounding $\textcircled{2}$ using $r_{\pi^\star} - \widehat{r}_{\pi^{(k)}}^{(k)} - C_r\beta_\pi^{(k)} \leq 0$, leveraging the optimism due to Lemma 1 with the maximality of $\pi^{(k)}$ in Opt-Pes (see, e.g., Abbasi-Yadkori et al. [1]). However, due to the pessimistic constraint in Opt-Pes, $\pi^\star$ may not be a solution to Opt-Pes, necessitating a modification to this approach.

Recall from Lemma 2 that, for $k \in \mathcal{U}^{\complement}$, the mixture policy $\pi_{\alpha^{(k)}} := (1 - \alpha^{(k)})\pi^{\mathrm{sf}} + \alpha^{(k)}\pi^\star$ satisfies $u_{\pi_{\alpha^{(k)}}} - 2C_u\beta_{\pi_{\alpha^{(k)}}}^{(k)} \geq b$. For this $\pi_{\alpha^{(k)}}$, the following optimism with respect to $\pi^\star$ holds:

**Lemma 3** ($\pi_{\alpha^{(k)}}$ optimism). *For any $k \in \mathcal{U}^{\complement}$, it holds $r_{\pi_{\alpha^{(k)}}} + (2BC_u\xi^{-1})\beta_{\pi_{\alpha^{(k)}}}^{(k)} \geq r_{\pi^\star}$.*

Using Lemmas 1 and 3 with $C_r = C_u(1 + 2B\xi^{-1})$, we have

$$\textcircled{2} \leq \sum_{k \in \mathcal{U}^{\complement}} \widehat{r}_{\pi_{\alpha^{(k)}}} + C_r\beta_{\pi_{\alpha^{(k)}}}^{(k)} - \widehat{r}_{\pi^{(k)}}^{(k)} - C_r\beta_{\pi^{(k)}}^{(k)} \leq 0 \, , \tag{4}$$

where the second inequality holds since $\pi_{\alpha^{(k)}}$ is a feasible solution to Opt-Pes and $\pi^{(k)}$ is its maximizer. This optimism via a mixture policy technique is adapted from tabular CMDPs [27, 8] to the linear bandit setup. By combining all the results, Algorithm 1 archives the following guarantees:

**Theorem 2.** *If OPLB-SP is run with the parameters listed in its **Input** line, w.p. at least $1 - \delta$,*

$$\pi^{(k)} \in \Pi^{\mathrm{sf}} \ \text{ for any } \ k \in [\![1, K]\!] \quad \text{and} \quad \mathrm{Regret}(K) \leq \widetilde{\mathcal{O}}(dBC_u^2\xi^{-2} + C_r\sqrt{dK}) \, .$$

*When $B = R = 1$, the regret bound simplifies to $\widetilde{\mathcal{O}}(d^2\xi^{-2} + \xi^{-1}\sqrt{d^3K})$.*

In summary, OPLB-SP relies on three components: (i) optimistic-pessimistic updates (Opt-Pes), (ii) a logarithmic number of $\pi^{\mathrm{sf}}$ deployments (Theorem 1), and (iii) compensation for the pessimism (Lemma 3). Building on these components, the next section develops a linear CMDP algorithm.

## 3 Safe Reinforcement Learning in Linear Constrained MDP

A finite-horizon CMDP is defined as a tuple $(\mathcal{S}, \mathcal{A}, H, P, r, u, b, s_1)$, where $\mathcal{S}$ is the finite but potentially exponentially large state space, $\mathcal{A}$ is the finite action space ($|\mathcal{A}| = A$),[7] $H \in \mathbb{N}$ is the episode horizon, $b \in [0, H]$ is the constrained threshold, and $s_1$ is the fixed initial state. The reward and utility functions $r, u : [\![1, H]\!] \times \mathcal{S} \times \mathcal{A} \to [0, 1]$ specify the reward $r_h(s, a)$ and constraint utility $u_h(s, a)$ when taking action $a$ at state $s$ in step $h$. Finally, $P_.(\cdot \mid \cdot, \cdot) : [\![1, H]\!] \times \mathcal{S} \times \mathcal{A} \times \mathcal{S} \to [0, 1]$ denotes the transition kernel, where $P_h(s' \mid s, a)$ denotes the state transition probability to a new state $s'$ from a state $s$ when taking an action $a$ in step $h$. With a slight abuse of notation, for functions $V : \mathcal{S} \to \mathbb{R}$ and $P_h$, we write $(P_hV)(x, a) = \sum_{y \in \mathcal{S}} V(y)P_h(y \mid x, a)$.

**Policy and (regularized) value functions.** A policy is defined as $\pi_.(\cdot \mid \cdot) : [\![1, H]\!] \times \mathcal{S} \times \mathcal{A} \to [0, 1]$, where $\pi_h(a \mid s)$ gives the probability of taking an action $a$ at state $s$ in step $h$. The set of all the policies is denoted as $\Pi$. With an abuse of notation, for any policy $\pi$ and $Q : \mathcal{S} \times \mathcal{A} \to \mathbb{R}$, let $\pi_h$ be an operator such that $(\pi_hQ)(s) = \sum_{a \in \mathcal{A}} \pi_h(a \mid s)Q(s, a)$. For a policy $\pi$, transition kernel $P$,

---

[7]While Section 2 permits infinite actions, we here restrict to the finite case. Even then, episode-wise safe exploration in linear CMDP is non-trivial, and infinite actions would further complicate the regret analysis.

reward function $g : [\![1, H]\!] \times \mathcal{S} \times \mathcal{A} \to \mathbb{R}$, and entropy coefficient $\kappa \geq 0$, let $Q_{P,h}^{\pi;g}[\kappa] : \mathcal{S} \times \mathcal{A} \to \mathbb{R}$ and $V_{P,h}^{\pi;g}[\kappa] : \mathcal{S} \to \mathbb{R}$ denote the entropy-regularized value functions at step $h$ satisfying:

$$Q_{P,h}^{\pi;g}[\kappa] = g_h + (P_h V_{h+1,P}^{\pi;g}[\kappa]), \ \ V_{P,h}^{\pi;g}[\kappa] = \pi_h(Q_{P,h}^{\pi;g}[\kappa] - \kappa \ln \pi_h), \ \ \text{and} \ \ V_{H+1,P}^{\pi;g}[\kappa] = \mathbf{0} \ .$$

For $\kappa = 0$, we omit $\kappa$, e.g., $Q_{P,h}^{\pi;g} := Q_{P,h}^{\pi;g}[0]$. We denote $h_\kappa := h(1 + \kappa \ln A)$ for $h \in [\![1, H]\!]$.

For $h \in [\![1, H]\!]$, let $w_{P,h}^\pi \in \Delta(\mathcal{S} \times \mathcal{A})$ denote the occupancy measure of $\pi$ in $P$ at step $h$ such that

$$w_{P,h}^\pi(s, a) = \mathbb{P}(s_h = s, a_h = a \mid \pi, P) \quad \forall (h, s, a) \in [\![1, H]\!] \times \mathcal{S} \times \mathcal{A} \ , \tag{5}$$

where the expectation is taken over all possible trajectories, in which $a_h \sim \pi_h(\cdot \mid s_h)$ and $s_{h+1} \sim P_h(\cdot \mid s_h, a_h)$. With a slight abuse of notation, we write $w_{P,h}^\pi(s) = \sum_{a \in \mathcal{A}} w_{P,h}^\pi(s, a)$.

**Learning Setup.** An agent interacts with the CMDP for $K$ episodes using policies $\pi^{(1)}, \ldots, \pi^{(K)} \in \Pi$. Each episode $k$ starts from $s_1$. At step $h$ in episode $k$, the agent observes a state $s_h^{(k)}$, selects an action $a_h^{(k)} \sim \pi_h^{(k)}(\cdot \mid s_h^{(k)})$, and transitions to $s_{h+1}^{(k)} \sim P_h(\cdot \mid s_h^{(k)}, a_h^{(k)})$. The algorithm lacks knowledge of the transition kernel $P$, while $r$ and $u$ are known for simplicity. Extending our setting to unknown stochastic reward and utility is straightforward (see, e.g., Efroni et al. [13]).

To handle a potentially large state space, we consider the following linear MDP assumption:

**Assumption 2** (Linear MDP). We have a known feature map $\phi : \mathcal{S} \times \mathcal{A} \to \mathbb{R}^d$ satisfying: there exist unknown $d$ (signed) measures $\boldsymbol{\mu}_h := (\boldsymbol{\mu}_h^1, \ldots, \boldsymbol{\mu}_h^d) \in \mathbb{R}^{S \times d}$ such that $P_h(s' \mid s, a) = \boldsymbol{\mu}_h(s')^\top \phi(s, a)$, and known vectors $\boldsymbol{\theta}_h^r, \boldsymbol{\theta}_h^u \in \mathbb{R}^d$ such that $r_h(s, a) = (\boldsymbol{\theta}_h^r)^\top \phi(s, a)$ and $u_h(s, a) = (\boldsymbol{\theta}_h^u)^\top \phi(s, a)$. We assume $\sup_{s,a} \|\phi(s, a)\|_2 \leq 1$ and $\left\|V^\top \boldsymbol{\mu}_h\right\|_2 \leq \sqrt{d}$ for any $V \in \mathbb{R}^\mathcal{S}$ such that $\|V\|_\infty \leq 1$.

Let $\pi^\star \in \arg\max_{\pi \in \Pi^{\mathrm{sf}}} V_{P,1}^{\pi,r}(s_1)$ be the optimal policy, where $\Pi^{\mathrm{sf}} := \{\pi \mid V_{P,1}^{\pi,u}(s_1) \geq b\}$ is the set of safe policies. The goal is to achieve sublinear regret under **episode-wise** constraints:

$$\mathrm{Regret}(K) := \sum_{k=1}^K V_{P,1}^{\pi^\star,r}(s_1) - V_{P,1}^{\pi^{(k)},r}(s_1) = o(K) \ \ \text{such that} \ \ \pi^{(k)} \in \Pi^{\mathrm{sf}} \quad \forall k \in [K] \ . \tag{6}$$

Finally, we assume the strictly safe policy similar to Section 2.

**Assumption 3** (Safe policy). We have access to $\pi^{\mathrm{sf}} \in \Pi^{\mathrm{sf}}$ and $\xi > 0$ such that $V_{P,1}^{\pi^{\mathrm{sf}},u}(s_1) - b \geq \xi$ .

### 3.1 Technical Challenge: Optimistic-Pessimistic Optimization in Linear CMDP

Our linear CMDP algorithm builds on OPLB-SP in Section 2: deploying an optimistic-pessimistic policy when confident in $\pi^{\mathrm{sf}}$; otherwise, it uses $\pi^{\mathrm{sf}}$. We will logarithmically bound the number of $\pi^{\mathrm{sf}}$ deployments, similar to Theorem 1, and ensure optimism through a linear mixture of policies, as in Lemma 2. However, computing an optimistic-pessimistic policy in the linear CMDP setting, similar to Opt-Pes, presents a non-trivial challenge. This section outlines the difficulties.

Following standard linear MDP algorithm frameworks (e.g., Jin et al. [23], Lykouris et al. [28]), for each $h, k$, let $\beta_h^{(k)} : (s, a) \mapsto \|\phi(s, a)\|_{(\Lambda_h^{(k)})^{-1}}$ be the bonus, where $\Lambda_h^{(k)} := \rho\mathbf{I} + \sum_{i=1}^{k-1} \phi(s_h^{(i)}, a_h^{(i)})\phi(s_h^{(i)}, a_h^{(i)})^\top$ and $\rho > 0$. For any $V : \mathcal{S} \to \mathbb{R}$, let $\widehat{P}_h^{(k)} V$ be the next-step value estimation defined as: $(\widehat{P}_h^{(k)} V)(s, a) := \phi(s, a)^\top (\Lambda_h^{(k)})^{-1} \sum_{i=1}^{k-1} \phi(s_h^{(i)}, a_h^{(i)}) V(s_{h+1}^{(i)})$. We construct the following optimistic and pessimistic value functions for reward and utility, respectively:

**Definition 2** (Clipped value functions). Let $C_r, C_u, C_\dagger, B_\dagger > 0$. For each $k, h, \pi$, and $\kappa \geq 0$, define $\overline{Q}_{(k),h}^{\pi,r}[\kappa], \overline{Q}_{(k),h}^{\pi,\dagger}, \underline{Q}_{(k),h}^{\pi,u} : \mathcal{S} \times \mathcal{A} \to \mathbb{R}$ and $\overline{V}_{(k),h}^{\pi,r}[\kappa], \overline{V}_{(k),h}^{\pi,\dagger}, \underline{V}_{(k),h}^{\pi,u} : \mathcal{S} \to \mathbb{R}$ such that:

$$\overline{Q}_{(k),h}^{\pi,r}[\kappa] := r_h + \mathrm{clip}\{C_r\beta_h^{(k)} + \widehat{P}_h^{(k)}\overline{V}_{(k),h+1}^{\pi,r}[\kappa], \ 0, \ H_\kappa - h_\kappa\}, \qquad \overline{V}_{(k),h}^{\pi,r}[\kappa] := \pi_h(\overline{Q}_{(k),h}^{\pi,r}[\kappa] - \kappa \ln \pi_h) \ ,$$

$$\overline{Q}_{(k),h}^{\pi,\dagger} := B_\dagger\beta_h^{(k)} + \mathrm{clip}\{C_\dagger\beta_h^{(k)} + \widehat{P}_h^{(k)}\overline{V}_{(k),h+1}^{\pi,\dagger}, \ 0, \ B_\dagger(H - h)\}, \quad \overline{V}_{(k),h}^{\pi,\dagger} := \pi_h\overline{Q}_{(k),h}^{\pi,\dagger} \ ,$$

$$\underline{Q}_{(k),h}^{\pi,u} := u_h + \mathrm{clip}\{-C_u\beta_h^{(k)} + \widehat{P}_h^{(k)}\underline{V}_{(k),h+1}^{\pi,u}, \ 0, \ H - h\}, \quad \text{and} \quad \underline{V}_{(k),h}^{\pi,u} := \pi_h\underline{Q}_{(k),h}^{\pi,u} \ .$$

We set $\overline{V}_{(k),H+1}^{\pi,r}[\kappa] = \overline{V}_{(k),H+1}^{\pi,\dagger} = \underline{V}_{(k),H+1}^{\pi,u} = \mathbf{0}$. For $\kappa = 0$, omit $\kappa$, e.g., $\overline{Q}_{(k),h}^{\pi,r} := \overline{Q}_{(k),h}^{\pi,r}[0]$.

---

**Algorithm 2:** Optimistic-Pessimistic Softmax Exploration for Linear CMDP

---

**Input:** Regr. coeff. $\rho = 1$, bonus scalers $C_r = \widetilde{\mathcal{O}}(dH)$, $C_u = \widetilde{\mathcal{O}}(dH)$, $C_\dagger = \widetilde{\mathcal{O}}(d^2 H^3 \xi^{-1})$,
$B_\dagger = \widetilde{\mathcal{O}}(dH^2\xi^{-1})$, entropy coeff. $\kappa = \widetilde{\Omega}(\xi^3 H^{-4} d^{-1} K^{-0.5})$, search length $T = \widetilde{\mathcal{O}}(H)$,
$\lambda$-threshold $C_\lambda = \widetilde{\mathcal{O}}(dH^4\xi^{-2})$, safe policy $\pi^{\text{sf}}$, and iter. length $K \in \mathbb{N}$

1  **for** $k = 1, \ldots, K$ **do**
2     Let $\underline{V}^{\pi,u}_{(k),h}$ be value function (Definition 2) and $\pi^{(k),\lambda}$ be softmax policy (Definition 3)
    /\* $\pi^{\text{sf}}$ trigger is implicitly tied to $\pi^{\text{sf}}$ confidence (Lemma 5)     \*/
3     **if** $\underline{V}^{\pi^{(k),C_\lambda},u}_{(k),1}(s_1) < b$ **then** Set $\pi^{(k)} := \pi^{\text{sf}}$
4     **else if** $\underline{V}^{\pi^{(k),0},u}_{(k),1}(s_1) \geq b$ **then** Set $\pi^{(k)} := \pi^{(k),0}$
5     **else** /\* Do bisection-search to find safe $\pi^{(k),\lambda}$ with small $\lambda$     \*/
6        Set $\underline{\lambda}^{(k,1)} := 0$ and $\overline{\lambda}^{(k,1)} := C_\lambda$. Let $\lambda^{(k,t)} := (\underline{\lambda}^{(k,t)} + \overline{\lambda}^{(k,t)})/2$
7        **for** $t = 1, \ldots, T$ **do**
8           **if** $\underline{V}^{\pi^{(k),\lambda^{(k,t)}},u}_{(k),1}(s_1) \geq b$ **then** $\underline{\lambda}^{(k,t+1)} := \underline{\lambda}^{(k,t)}$ and $\overline{\lambda}^{(k,t+1)} := \lambda^{(k,t)}$
9           **else** $\underline{\lambda}^{(k,t+1)} := \lambda^{(k,t)}$ and $\overline{\lambda}^{(k,t+1)} := \overline{\lambda}^{(k,t)}$
10       Set $\pi^{(k)} := \pi^{(k),\overline{\lambda}^{(k,T)}}$
11    Sample a trajectory $(s_1^{(k)}, a_1^{(k)}, \ldots, s_H^{(k)}, a_H^{(k)})$ by deploying $\pi^{(k)}$

---

We will utilize $\overline{Q}^{\pi,\dagger}_{(k),h}$ and $\overline{V}^{\pi,\dagger}_{(k),h}$ to compensate for the pessimism, similar to the bandit proof in (4).[8] Entropy regularization in $\overline{Q}^{\pi,r}_{(k),h}[\kappa]$ is for the later analysis. The clipping operators are essential to avoid the propagation of unreasonable value estimates [48].

Using these value functions, one might consider extending Opt-Pes to linear CMDPs by solving:

$$\max_{\pi \in \Pi} \overline{V}^{\pi,r}_{(k),1}(s_1) + \overline{V}^{\pi,\dagger}_{(k),1}(s_1) \text{ such that } \underline{V}^{\pi,u}_{(k),1}(s_1) \geq b. \tag{7}$$

However, solving this (7) is challenging due to (i) **the large state space** in the linear CMDP setting ($|\mathcal{S}| \gg 1$) and (ii) **the clipping operators** in $\overline{Q}^{\pi,r}_{(k),h}$, $\overline{Q}^{\pi,\dagger}_{(k),h}$, and $\underline{Q}^{\pi,u}_{(k),h}$.

In tabular CMDPs with small $|\mathcal{S}|$, Liu et al. [27] and Bura et al. [8] used linear programming (LP) to solve similar optimistic-pessimistic optimization problems, achieving zero violation. However, the computational cost of LP scales with $|\mathcal{S}|$, making it impractical for linear CMDPs.

Another option is the Lagrangian method, which reformulates the constrained optimization as a min-max optimization: $\min_{\lambda \geq 0} \max_{\pi \in \Pi} \overline{V}^{\pi,r}_{(k),1}(s_1) + \overline{V}^{\pi,\dagger}_{(k),1}(s_1) + \lambda(\underline{V}^{\pi,u}_{(k),1}(s_1) - b)$. When the value functions are exact, i.e., $\overline{V}^{\pi,\dagger}_{(k),h} + \overline{V}^{\pi,r}_{(k),h} + \underline{V}^{\pi,u}_{(k),h} = V^{\pi,r+B_\dagger \beta^{(k)} + \lambda u}_{P,h}$, this min-max is equivalent to (7), and the inner maximization reduces to a standard policy optimization [2]. Both favorable properties arise due to the linearity of the value function in the occupancy measure (see, e.g., Paternain et al. [34]). However, due to clipping, the value functions in Definition 2 may not be representable via occupancy measures, making the Lagrangian approach inapplicable.

To address this large-scale optimization challenge, instead of directly solving (7), we realize optimism and pessimism through a novel adaptation of the recent **softmax policy** technique for linear CMDPs [18, 16], combined with **the $\pi^{\text{sf}}$ deployment technique** from Section 2.

### 3.2 Algorithm and Analysis

We summarize the proposed OPSE-LCMDP in Algorithm 2 and analyze it under the parameters in its **Input** line. All formal theorems and proofs in this section are in Appendix E. A key component of our algorithm is the **composite softmax policy**, which balances optimism and pessimism via $\lambda \geq 0$:

---

[8]Increasing $C_r$ and clip-threshold could offer similar compensation, but separated values simplify analysis.

**Definition 3** (Composite softmax policy). For $\lambda \geq 0$, $\kappa > 0$, let $\pi^{(k),\lambda} \in \Pi$ be a policy such that

$$\pi_h^{(k),\lambda}(\cdot \mid s) = \mathrm{SoftMax}\left(\frac{1}{\kappa}\left(\overline{Q}_{(k),h}^{\pi^{(k),\lambda},\dagger}(s,\cdot) + \overline{Q}_{(k),h}^{\pi^{(k),\lambda},r}[\kappa](s,\cdot) + \lambda \underline{Q}_{(k),h}^{\pi^{(k),\lambda},u}(s,\cdot)\right)\right) .$$

$\pi^{(k),\lambda}$ can be computed iteratively in a backward manner for $h = H, \ldots, 1$. For this $\pi^{(k),\lambda}$, using the Lipschitz continuity of $\mathrm{SoftMax}(\cdot)$ (see Ghosh et al. [16]), the following confidence bounds hold:

**Lemma 4** (Confidence bounds). *For any $(k,h)$, $\lambda \in [0, C_\lambda]$, $\pi \in \{\pi^{(k),\lambda}, \pi^{\mathrm{sf}}\}$, w.p. at least $1 - \delta$,*

$$V_{P,h}^{\pi,r} \leq \overline{V}_{(k),h}^{\pi,r} \leq V_{P,h}^{\pi,r+2C_r\beta^{(k)}}, \quad V_{P,h}^{\pi,B_\dagger\beta^{(k)}} \leq \overline{V}_{(k),h}^{\pi,\dagger} \leq V_{P,h}^{\pi,(B_\dagger+2C_\dagger)\beta^{(k)}}, \quad V_{P,h}^{\pi,u-2C_u\beta^{(k)}} \leq \underline{V}_{(k),h}^{\pi,u} \leq V_{P,h}^{\pi,u}.$$

Using Lemma 4, analogous to Section 2.2.1, we next establish the zero-violation guarantee.

### 3.2.1 Zero-Violation and Logarithmic Number of $\pi^{\mathrm{sf}}$ Deployments

In the softmax policy (Definition 3), $\lambda$ balances optimism and pessimism: a small $\lambda$ promotes exploration, while a large $\lambda$ prioritizes constraint satisfaction. Building on this, Algorithm 2 conducts a **bisection search** to find the smallest feasible $\lambda$ while ensuring the pessimistic constraint holds (Line 4 to Line 10). If a large $\lambda = C_\lambda$ fails to satisfy the constraint, the algorithm assumes no feasible pessimistic policy exists and deploys $\pi^{\mathrm{sf}}$ (Line 3). Since the softmax policy is only deployed for $\lambda$ satisfying $\underline{V}_{(k),1}^{\pi^{(k),\lambda},u}(s_1) \geq b$, Lemma 4 implies the following zero-violation guarantees:

**Corollary 2** (Zero-violation). *W.p. at least $1 - \delta$, Algorithm 2 satisfies $\pi^{(k)} \in \Pi^{\mathrm{sf}}$ for any $k$.*

Next, we bound the number of $\pi^{\mathrm{sf}}$ deployments to achieve sublinear regret. To this end, similar to the bandit warm-up (Section 2), **we relate $\pi^{\mathrm{sf}}$ deployment to $\pi^{\mathrm{sf}}$ uncertainty level** and logarithmically bound the number of uncertain iterations. The following Lemma 5 ensures that, if Algorithm 2 is confident in $\pi^{\mathrm{sf}}$ and runs with appropriate $C_\lambda$ and $\kappa$, then $\pi^{\mathrm{sf}}$ is not deployed.

**Definition 4** ($\pi^{\mathrm{sf}}$ unconfident iterations). Let $\mathcal{U}$ be the iterations when Algorithm 2 is unconfident in $\pi^{\mathrm{sf}}$, i.e., $\mathcal{U} := \{k \in [\![1, K]\!] \mid V_{P,1}^{\pi^{\mathrm{sf}},\beta^{(k)}}(s_1) > \frac{\xi}{4C_u}\}$. Let $\mathcal{U}^{\complement} := [\![1, K]\!] \setminus \mathcal{U}$ be its complement.

**Lemma 5** (Implicit $\pi^{\mathrm{sf}}$ deployment trigger). *When $C_\lambda \geq \frac{8H_\kappa^2(B_\dagger+1)}{\xi}$ and $\kappa \leq \frac{\xi^2}{32H_\kappa^2(B_\dagger+1)}$, then w.p. at least $1 - \delta$, it holds that $\underline{V}_{(k),1}^{\pi^{(k),C_\lambda},u}(s_1) \geq b$ for all $k \in \mathcal{U}^{\complement}$.*

Essentially, the proof of Lemma 5 relies on the following monotonic property of the value function for the softmax policy: if the value estimation is exact, increasing $\lambda$ monotonically improves safety.

**Lemma 6** (Softmax value monotonicity). *For $\lambda \geq 0$, let $\pi^\lambda$ be a softmax policy such that $\pi_h^\lambda(\cdot \mid s) = \mathrm{SoftMax}(\frac{1}{\kappa}(Q_{P,h}^{\pi,r}[\kappa](s,\cdot) + \lambda Q_{P,h}^{\pi,u}(s,\cdot)))$. Then, $V_{P,1}^{\pi^\lambda,u}(s_1)$ is monotonically increasing in $\lambda$.*

While the true value function enjoys this monotonicity, the estimated value $\underline{V}_{(k),1}^{\pi^{(k),\lambda},u}(s_1)$ may not, as $\widehat{P}_h^{(k)}V$ can take negative values even when $V$ is positive. This complicates the proof of Lemma 5. To address this, we leverage Lemma 4, which sandwiches the estimated values by some true values. We prove Lemma 5 by showing that, for sufficiently large $C_\lambda$, any sandwiched value satisfies the constraint under pessimism, implying that the estimated value also satisfies it. This novel result enables bisection search to adjust $\lambda$, making OPSE-LCMDP more computationally efficient than Ghosh et al. [18]. The detailed proofs of Lemmas 5 and 6 are provided in Appendix E.4.1.

Finally, the following theorem ensures that the number of $\pi^{\mathrm{sf}}$ deployment scales logarithmic to $K$, as in Theorem 1. The proof follows from extending the bandit's proof of Theorem 1 to CMDPs.

**Theorem 3** (Logarithmic $|\mathcal{U}|$ bound). *It holds w.p. at least $1 - \delta$ that $|\mathcal{U}| \leq \mathcal{O}\left(d^3H^4\xi^{-2}\ln KH\delta^{-1}\right)$.*

### 3.2.2 Regret Analysis

The remaining task is to ensure sublinear regret. By Theorem 3 and Lemma 4, the regret is decomposed as:

$$\mathrm{Regret}(K) \leq \widetilde{\mathcal{O}}\left(\frac{d^3H^4}{\xi^2}\right) + \underbrace{\sum_{k \in \mathcal{U}^{\complement}} V_{P,1}^{\pi^{(k)},2C_r\beta^{(k)}}(s_1)}_{①} + \underbrace{\sum_{k \in \mathcal{U}^{\complement}}\left(V_{P,1}^{\pi^\star,r}(s_1) - \overline{V}_{(k),1}^{\pi^{(k)},r}[\kappa](s_1)\right)}_{②} + \kappa KH\ln A ,$$

where the last term arises from the entropy regularization $(V_{P,1}^{\pi,r}(s_1)[\kappa] - V_{P,1}^{\pi,r}(s_1) \leq \kappa H \ln A)$. Using the elliptical potential lemma for linear MDPs [23], we obtain ① $\leq \widetilde{\mathcal{O}}(C_r H \sqrt{dK})$.

We now bound ②. Note that for any $k \in \mathcal{U}^\complement$, due to Lemma 5, $\pi^{(k)}$ is the softmax policy by Line 10. To bound ②, following a similar approach to Lemma 3 in the bandit, we replace $\pi^\star$ with a mixture policy that satisfies the pessimistic constraint. To this end, we utilize the following lemmas.

**Definition 5** (Mixture policy). *For $\alpha \in [0,1]$, let $\pi^\alpha$ be a mixture policy such that, for any $h$, $w_{P,h}^{\pi^\alpha} = (1-\alpha)w_{P,h}^{\pi^{\mathrm{sf}}} + \alpha w_{P,h}^{\pi^\star}$. Such a $\pi^\alpha$ is ensured to exists for any $\alpha \in [0,1]$ [7].*

**Lemma 7** (Safe and optimistic mixture policy). *Let $\alpha^{(k)} := \frac{\xi}{\xi + 2V_{P,1}^{\pi^\star, 2C_u\beta^{(k)}}(s_1)}$. If $B_\dagger \geq \frac{4C_u H}{\xi}$, then for any $k \in \mathcal{U}^\complement$, it holds* (i) $V_{P,1}^{\pi^{\alpha^{(k)}}, u - 2C_u\beta^{(k)}}(s_1) \geq b$ *and* (ii) $V_{P,1}^{\pi^{\alpha^{(k)}}, r + B_\dagger\beta^{(k)}}(s_1) \geq V_{P,1}^{\pi^\star, r}(s_1)$.

We note that the mixture policy $\pi^{\alpha^{(k)}}$ is introduced only for the regret analysis; it is not required in the actual algorithm. Since $\overline{\lambda}^{(k,T)}$ is chosen to satisfy $\underline{V}_{(k),1}^{\pi^{(k)}, u}(s_1) < b$ and $b \leq V_{P,1}^{\pi^{\alpha^{(k)}}, u - 2C_u\beta^{(k)}}(s_1)$ holds by Lemma 7,

$$
\begin{aligned}
② \leq \quad & \sum_{k \in \mathcal{U}^\complement} \left( V_{P,1}^{\pi^{\alpha^{(k)}}, B_\dagger\beta^{(k)}}(s_1) + V_{P,1}^{\pi^{\alpha^{(k)}}, r}[\kappa](s_1) + \overline{\lambda}^{(k,T)} V_{P,1}^{\pi^{\alpha^{(k)}}, u - 2C_u\beta^{(k)}}(s_1) \right. \\
& \left. \left. - \overline{V}_{(k),1}^{\pi^{(k)}, \dagger}(s_1) - \overline{V}_{(k),1}^{\pi^{(k)}, r}[\kappa](s_1) - \overline{\lambda}^{(k,T)} \underline{V}_{(k),1}^{\pi^{(k)}, u}(s_1) \right) \right\} ③ \\
& + \underbrace{\sum_{k \in \mathcal{U}^\complement} \overline{V}_{(k),1}^{\pi^{(k)}, \dagger}(s_1)}_{④} + C_\lambda \underbrace{\sum_{k \in \mathcal{U}^\complement} \left( \underline{V}_{(k),1}^{\pi^{(k)}, \overline{\lambda}^{(k,T)}, u}(s_1) - \underline{V}_{(k),1}^{\pi^{(k)}, \underline{\lambda}^{(k,T)}, u}(s_1) \right)}_{⑤} .
\end{aligned}
$$

Using Lemma 4, similar to ①, we have ④ $\leq \widetilde{\mathcal{O}}\left( (B_\dagger + C_\dagger) H \sqrt{dK} \right)$. The term ⑤ is controlled by the bisection search width $(\overline{\lambda}^{(k,T)} - \underline{\lambda}^{(k,T)})$ and the following sensitivity of $\underline{V}_{(k),1}^{\pi^{(k)}, \lambda, u}(s_1)$ to $\lambda$.

**Lemma 8.** *For any $k$ and $\lambda \in [0, C_\lambda]$, we have $\left| \underline{V}_{(k),1}^{\pi^{(k)}, \lambda, u}(s_1) - \underline{V}_{(k),1}^{\pi^{(k)}, \lambda + \varepsilon, u}(s_1) \right| \leq \mathcal{O}\left( (KH)^H \right) \varepsilon$*

Ghosh et al. [18] also derived a similar exponential bound (see their Appendix C). Due to the update rule of the bisection search, setting the search iteration to $T = \widetilde{\mathcal{O}}(H)$ ensures that ⑤ $\leq \widetilde{\mathcal{O}}(1)$.

For ③, using a modification of the so-called value-difference lemma [40], we have

$$
③ = \sum_{k \in \mathcal{U}^\complement} V_{P,1}^{\pi^{\alpha^{(k)}}, f^1}(s_1) - V_{P,1}^{\pi^{\alpha^{(k)}}, f^2}(s_1) - \overline{\lambda}^{(k,T)} V_{P,1}^{\pi^{\alpha^{(k)}}, 2C_u\beta^{(k)}}(s_1) , \tag{8}
$$

where $f^1, f^2 : [\![1, H]\!] \times \mathcal{S} \times \mathcal{A} \to \mathbb{R}$ are functions such that, for any $h$,

$$
f_h^1 = \left( \pi_h^{\alpha^{(k)}} - \pi_h^{(k)} \right) \left( \overline{Q}_{(k),h}^{\pi^{(k)}, \dagger} + \overline{Q}_{(k),h}^{\pi^{(k)}, r}[\kappa] + \overline{\lambda}^{(k)} \underline{Q}_{(k),h}^{\pi^{(k)}, u} \right) - \kappa \pi_h^{\alpha^{(k)}} \ln \pi_h^{\alpha^{(k)}} + \kappa \pi_h^{(k)} \ln \pi_h^{(k)}
$$

$$
\text{and} \quad f_h^2 = \left( \overline{Q}_{(k),h}^{\pi^{(k)}, r}[\kappa] - r_h - P_h \overline{V}_{(k),h+1}^{\pi^{(k)}, r}[\kappa] \right) + \overline{\lambda}^{(k,T)} \left( u_h + P_h \underline{V}_{(k),h+1}^{\pi^{(k)}, u} - \underline{Q}_{(k),h}^{\pi^{(k)}, u} \right) + \left( \overline{Q}_{(k),h}^{\pi^{(k)}, \dagger} - B_\dagger\beta^{(k)} - P_h \overline{V}_{(k),h+1}^{\pi^{(k)}, \dagger} \right) .
$$

**Our use of the softmax policy with entropy regularization is crucial for bounding ③.** Since the analytical maximizer of the regularized optimization $\max_{\pi \in \mathscr{P}(\mathcal{A})} \sum_{a \in \mathcal{A}} \pi(a)(\mathbf{x}(a) - \kappa \ln \pi(a))$ is given by $\mathrm{SoftMax}\left(\frac{1}{\kappa}\mathbf{x}(\cdot)\right)$, it follows that $f^1$ is non-positive, implying $V_{P,1}^{\pi^{\alpha^{(k)}}, f^1}(s_1) \leq 0$. Additionally, applying Lemma 4, we derive $f_h^2 \geq -\overline{\lambda}^{(k,T)} 2C_u\beta_h^{(k)}$, which leads to $-V_{P,1}^{\pi^{\alpha^{(k)}}, f^2}(s_1) - \overline{\lambda}^{(k,T)} V_{P,1}^{\pi^{\alpha^{(k)}}, 2C_u\beta^{(k)}}(s_1) \leq 0$. By substituting these bounds into Equation (8), we obtain ③ $\leq 0$.

By combining all the results, Algorithm 2 achieves the following guarantees:

**Theorem 4.** *If OPSE-LCMDP is run with the parameters listed in its **Input** line, w.p. at least $1 - \delta$,*

$$
\pi^{(k)} \in \Pi^{\mathrm{sf}} \ \forall k \in [\![1, K]\!] \quad \text{and} \quad \mathrm{Regret}(K) \leq \underbrace{\widetilde{\mathcal{O}}(H^2\sqrt{d^3 K})}_{(i)} + \underbrace{\widetilde{\mathcal{O}}(d^3 H^4 \xi^{-2})}_{(ii)} + \underbrace{\widetilde{\mathcal{O}}(H^4 \xi^{-1}\sqrt{d^5 K})}_{(iii)} .
$$

Notably, **Theorem 4 is the first linear CMDP result achieving zero episode-wise constraint violations and sublinear regret.** We conclude this section by discussing the regret bound quality and computational cost of OPSE-LCMDP.

**Remark 1** (Can we do better?). *Without the constraint—i.e., removing* (ii) *and* (iii)—*our bound matches the* $\widetilde{\mathcal{O}}(H^2\sqrt{d^3K})$ *regret of the fundamental LSVI-UCB algorithm [23]. The* $\xi^{-2}$ *term in* (ii) *is unavoidable [32]. The* $\xi^{-1}$ *dependence in* (iii) *remains unresolved, yet appears in all the existing safe RL literature [27, 8, 47, 36, 3, 32, 22]. These observations suggest that the bound is tight in* $\xi^{-1}$ *and* $K$. *As for* $d$ *and* $H$, *the term* (iii) *introduces an extra* $dH^2$ *factor over* (i). *Similar deterioration has been observed in tabular CMDPs [27, 42] and partially mitigated via Bernstein-type bonus analysis [47]. While improvement may be possible, it is unclear whether our* $\sqrt{d^5H^8}$ *dependence is overly loose when compared with existing CMDP regret bounds (e.g.,* $\sqrt{d^3H^4}$ *by Ghosh et al. [18]), since none of the existing results achieve episode-wise safe exploration. In general, regret or sample complexity bounds under different safety requirements are not directly comparable, even if the problem settings appear similar. For example, Vaswani et al. [42] shows that the sample complexity lower bound for tabular CMDPs exhibits a worse dependence on the horizon under a strict safe policy requirement than when small violations are allowed. Establishing a formal regret lower bound for our setting would be necessary to assess the tightness of our result, but this is beyond the scope of the current paper.*

**Remark 2** (Computational cost). *Algorithm 2 requires up to* $T$ *value evaluations (Definition 2) and policy computation (Definition 3). Using the bisection search, we bound* $T = \widetilde{\mathcal{O}}(H)$, *reducing the computational cost per-iteration to* $\widetilde{\mathcal{O}}(H \times [value \& policy comp.])$. *As this cost scales polynomially with* $A, H,$ *and* $d$ *[28],* **OPSE-LCMDP** *runs in polynomial time—an improvement over recent Ghosh et al. [18], which achieves* $\widetilde{\mathcal{O}}(\sqrt{K})$ *violation regret but incurs an exponential* $K^H$ *cost.*

## 4   Conclusion

This paper proposed OPSE-LCMDP, the first RL algorithm achieving both sublinear regret and episode-wise constraint satisfaction in linear CMDPs (Theorem 4). Our approach builds on optimistic-pessimistic exploration with two key innovations: (i) a novel deployment rule for $\pi^{\mathrm{sf}}$ and (ii) a softmax-based approach for efficiently implementing optimistic-pessimistic policies in linear CMDPs.

**Experiments.**   We numerically evaluate OPSE-LCMDP on several linear CMDP environments to support our theoretical results. We compare OPSE-LCMDP with the prior state-of-the-art linear CMDP algorithm of Ghosh et al. [18] and the tabular algorithm called DOPE [8]. Across all environments, OPSE-LCMDP achieves sublinear regret with zero constraint violation, while Ghosh et al. [18] shows positive violation regret. These results empirically validate Theorem 4. While DOPE also achieves zero violation, its use is limited to the tabular settings where $S$ is small. This highlights the computational tractability of our OPSE-LCMDP in large $S$, which supports Remark 2. All the results and details are deferred to Appendix F.

**Limitation and future work.**   OPSE-LCMDP achieves computational efficiency by the bisection search over $\lambda \in [0, C_\lambda]$, which works in the single-constraint setting thanks to the monotonicity in Lemma 6. However, extending our method to the **multi-constraint setting** is non-trivial, as $\lambda$ becomes a vector, requiring a vectorized version of the monotonicity lemma. Nonetheless, all theoretical results in Table 1 are also limited to single-constraint settings, meaning our work still advances the state of the art in safety. An efficient and safe algorithm for multi-constraint settings remains open for future work.

Another future direction is to extend the analysis to adversarial initial states $s_1$. This extension is non-trivial, as our core techniques rely on the fixed-state assumption. For example, we control the number of safe policy deployments by evaluating the bonus-return function $V_{P,1}^{\pi^{\mathrm{sf}}, \beta^{(k)}}(s_1)$ (Definition 4), which explicitly depends on $s_1$. The bound on the number of $\pi^{\mathrm{sf}}$ deployments (Theorem 3) and the existence of optimistic–pessimistic policies (Lemma 7) also depend on this initial state. Extending these analyses to handle adversarial $s_1$ would entail non-trivial technical challenges and is left for future work.

## Acknowledgments and Disclosure of Funding

This work is supported by JST Moonshot R&D Program Grant Number JPMJMS2236. AG acknowledges NJIT Start-up fund indexed number 172884.

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

# Contents

# A  List of Symbols

The next list describes several symbols used within this paper.

**Mathematical Notations**

$\mathscr{P}(\mathcal{S})$   Set of probability distributions over a set $\mathcal{S}$

$[\![a, b]\!]$   Set of integers defined by $\{a, \dots, b\}$

$\mathrm{clip}\{\mathbf{x}, a, b\}$   Clipping function which returns $\mathbf{x}'$ with $\mathbf{x}'_i = \min\{\max\{\mathbf{x}_i, a\}, b\}$ for each $i$

$\mathbf{0}, \mathbf{1}$   Vectors such that $\mathbf{0} := (0, \dots, 0)^\top$ and $\mathbf{1} := (1, \dots, 1)^\top$

$\|\mathbf{x}\|_{\mathbf{A}}$   For a positive definite matrix $\mathbf{A} \in \mathbb{R}^{d \times d}$ and $\mathbf{x} \in \mathbb{R}^d$, we denote $\|\mathbf{x}\|_{\mathbf{A}} = \sqrt{\mathbf{x}^\top \mathbf{A} \mathbf{x}}$

$a_n = O(b_n)$   There exist constants $C > 0$ and $N \in \mathbb{N}$ such that $a_n \leq Cb_n$ for all $n \geq N$

$\widetilde{O}(\cdot)$   Hide the polylogarithmic factors from $O(\cdot)$

$\mathrm{SoftMax}(\mathbf{x})$   Softmax distribution satisfying $\mathrm{SoftMax}(\mathbf{x})_i = \exp(\mathbf{x}_i)/(\sum_i \exp(\mathbf{x}_i))$

$\mathrm{dist}_\infty$   Distance metric for two functions $Q, Q' : \mathcal{S} \times \mathcal{A} \to \mathbb{R}$ or $V, V' : \mathcal{S} \to \mathbb{R}$ (see Definition 7)

$\mathrm{dist}_1$   Distance metric for two functions $\pi, \pi' : \mathcal{S} \to \mathscr{P}(\mathcal{A})$ (see Definition 7)

**Constrained Bandit**

$\mathcal{A} \subset \mathbb{R}^d$   Set of actions

$K$   Round length of the bandit problem

$r^{(k)}, u^{(k)}$   Reward and utility at $k$-th round

$\boldsymbol{\theta}_r, \boldsymbol{\theta}_u \in \mathbb{R}^d$   Unknown vectors for reward and utility

$\varepsilon_r, \varepsilon_u$   $R$-sub-Gaussian random noises

$r_\pi, u_\pi$   $g_\pi = \mathbb{E}_{\mathbf{a} \sim \pi}[\langle \boldsymbol{\theta}_g, \mathbf{a} \rangle]$ for both $g \in \{r, u\}$

$\Pi^{\mathrm{sf}}$   Set of safe policies $\{\pi \mid u_\pi \geq b\}$

$\pi^{\mathrm{sf}}$   Safe policy

$\xi > 0$   Safety of $\pi^{\mathrm{sf}}$ such that $u_{\pi^{\mathrm{sf}}} - b \geq \xi$

$\boldsymbol{\Lambda}^{(k)}$   Gram matrix defined by $\boldsymbol{\Lambda}^{(k)} := \rho \mathbf{I} + \sum_{i=1}^{k-1} \mathbf{a}^{(i)}(\mathbf{a}^{(i)})^\top$

$\widehat{\boldsymbol{\theta}}_r^{(k)}, \widehat{\boldsymbol{\theta}}_u^{(k)}$   Estimates of $\boldsymbol{\theta}_r$ and $\boldsymbol{\theta}_u$

$\beta_\pi^{(k)}$   Bonus function

$C_r, C_u$   Bonus scalers for reward and utility, respectively

$\mathcal{U}$   Set of iterations when Algorithm 1 is unconfident in $\pi^{\mathrm{sf}}$ (see Definition 1)

**Constrained MDP**

$K$   Number of episodes of the CMDP problem

$H$   Horizon

$\mathcal{S}, \mathcal{A}$   State space and action spaces

$P$   Transition kernel

$r, u$   Reward and utility functions

$s_1$   Initial state

$V_{P,h}^{\pi;g}[\kappa]$   Regularized state value function for a reward function $g$ with an entropy coefficient $\kappa$

$Q_{P,h}^{\pi;g}[\kappa]$   Regularized action value function for a reward function $g$ with an entropy coefficient $\kappa$

$h_\kappa$   Shorthand of $h(1 + \kappa \ln A)$

$w_{P,h}^\pi$   Occupancy measure of $\pi$ in $P$ at step $h$

$\phi : \mathcal{S} \times \mathcal{A} \to \mathbb{R}^d$  Feature map of the linear CMDP (see Assumption 2)

$\boldsymbol{\mu}_h$  $d$-signed measures specifying the transition probabilities (see Assumption 2)

$\boldsymbol{\theta}_h^r, \boldsymbol{\theta}_r^u \in \mathbb{R}^d$  Known vectors specifying the reward and utilitiy functions (see Assumption 2)

$\Pi^{\mathrm{sf}}$  Set of safe policies $\{\pi \mid V_{P,1}^{\pi,u}(s_1) \geq b\}$

$\pi^{\mathrm{sf}}$  Safe policy

$\xi > 0$  Safety of $\pi^{\mathrm{sf}}$ such that $V_{P,1}^{\pi,u}(\pi^{\mathrm{sf}}) - b \geq \xi$

$\boldsymbol{\Lambda}_h^{(k)}$  Gram matrix defined by $\boldsymbol{\Lambda}_h^{(k)} \coloneqq \rho \mathbf{I} + \sum_{i=1}^{k-1} \phi(s_h^{(i)}, a_h^{(i)}) \phi(s_h^{(i)}, a_h^{(i)})^\top$

$\widehat{P}_h^{(k)} V$  Next value estimation: $(\widehat{P}_h^{(k)} V)(s, a) \coloneqq \phi(s,a)^\top (\boldsymbol{\Lambda}_h^{(k)})^{-1} \sum_{i=1}^{k-1} \phi(s_h^{(i)}, a_h^{(i)}) V(s_{h+1}^{(i)})$

$\overline{Q}_{(k),h}^{\pi,r}[\kappa], \underline{Q}_{(k),h}^{\pi,u}, \overline{Q}_{(k),h}^{\pi,\dagger}$  Clipped value functions defined in Definition 2

$C_r, C_u, C_\dagger, B_\dagger$  Bonus scalers used in Definition 2

$\mathcal{N}_\varepsilon$  $\varepsilon$-cover of a certain set

$\mathcal{U}$  Set of iterations when Algorithm 2 is unconfident in $\pi^{\mathrm{sf}}$ (see Definition 4)

$\pi_h^{(k),\lambda}$  Softmax policy with a parameter $\lambda$ (see Definition 3)

$\lambda$  Parameter to balance optimism and pessimism of $\pi_h^{(k),\lambda}$ (see Definition 3)

$C_\lambda$  Maximum value of $\lambda$

$[\underline{\lambda}^{(k,t)}, \overline{\lambda}^{(k,t)}]$  Search space of the bisection search at iteration $t$ in episode $k$ (see Algorithm 2)

$T$  Iteration length of the bisection search (see Algorithm 2)

$\mathcal{Q}^r, \mathcal{Q}^u, \mathcal{Q}^\dagger$  Function classes for $Q$-functions defined in Definition 10

$\mathcal{Q}^\circ$  Function class for the composite of $Q$-functions defined in Definition 11

$\widetilde{\Pi}$  Class for softmax policies defined in Definition 12

$\mathcal{V}^r, \mathcal{V}^u, \mathcal{V}^\dagger$  Function classes for $V$-functions defined in Definition 13

$\delta_{(k)}^{\pi,r}, \delta_{(k)}^{\pi,u}, \delta_{(k)}^{\pi,\dagger}$  Discrepancies between the estimated and true $Q$-functions (see Definition 14)

$\Delta_r^{(k)}, \Delta_u^{(k)}, \Delta_\dagger^{(k)}$  Function classes for $\delta_{(k)}^{\pi,r}, \delta_{(k)}^{\pi,u}, \delta_{(k)}^{\pi,\dagger}$ (see Definition 14)

# B Related Work

## B.1 Related Algorithms

Building on the seminal work of Efroni et al. [13], numerous safe RL algorithms for CMDPs have been developed, broadly categorized into linear programming (LP) approaches and Lagrangian-based approaches.

**Linear programming.** LP approaches formulate CMDPs as linear optimization problems [2], solving them using an estimated transition kernel. Efroni et al. [13] introduced a basic sublinear regret algorithm, while HasanzadeZonuzy et al. [20] provided $(\varepsilon, \delta)$-PAC guarantees, ensuring the algorithm outputs a near-optimal policy. However, these methods permit constraint violations during exploration, making them unsuitable for safety-critical applications. Liu et al. [27] and Bura et al. [8] developed LP-based algorithms that achieve sublinear regret while maintaining episode-wise zero-violation guarantees by incorporating optimistic-pessimistic estimation into the LP formulation.

LP-based approaches in tabular settings, however, suffer from computational costs that scale with the size of the state space, making them impractical for linear CMDPs. While several studies propose LP algorithms for linear MDPs [31, 5, 30, 25, 14], these methods either use occupancy measures as decision variables—which can be exponentially large for large state spaces—or require a set of feature vectors that sufficiently cover the state space, which may not be feasible in our exploration settings. Moreover, as described in Section 3.1, the estimated value functions in linear CMDPs with exploration require clipping operators, further complicating the use of occupancy-measure-based approaches like LP methods in our setting.

**Lagrangian approach.** Lagrangian approaches reformulate the constrained optimization $\max_\pi \{f(\pi) \mid h(\pi) \geq 0\}$ as a min-max optimization $\min_{\lambda \geq 0} \max_\pi \{f(\pi) + \lambda h(\pi)\}$, and simultaneously optimize both $\pi$ and $\lambda$. When an algorithm gradually updates $\pi$ and then adjusts $\lambda$ incrementally, it is referred to as a **primal-dual (PD)** algorithm [11]. In contrast, if $\lambda$ is updated only after fully optimizing $\pi$ in the inner maximization, it is known as a **dual** approach [46]. Since the inner maximization reduces to standard policy optimization, Lagrangian methods integrate naturally with scalable methods such as policy gradient and value iteration.

For the tabular settings, Wei et al. [44], Müller et al. [29] develop model-free primal-dual algorithms with sublinear regret, while Wei et al. [45] extends this approach to the average-reward setting. Zeng et al. [49], Kitamura et al. [24] propose $(\varepsilon, \delta)$-PAC primal-dual algorithms, and Vaswani et al. [42] achieved the PAC guarantee via dual approach.

Beyond tabular settings, Ding et al. [10] propose PD algorithms with linear function approximation, achieving sublinear regret guarantees. Ghosh et al. [17] extend this to the average-reward linear CMDPs. Ghosh et al. [16] take a dual approach, also attaining sublinear regret in the finite-horizon settings.

These PD and dual algorithms, however, do not ensure episode-wise zero violation. Intuitively, the key issue lies in their $\lambda$-adjustment strategy, which updates $\lambda$ only incrementally. For example, the basic PD and dual algorithms by Efroni et al. [13] updates $\lambda$ using $\lambda^{(k+1)} \leftarrow \lambda^{(k)} + \alpha \cdot [\text{violation}]$, where $\alpha$ is a small learning rate. Since $\lambda$ controls constraint satisfaction, if the current policy fails to satisfy constraints adequately, $\lambda$ should be increased sufficiently before the next policy deployment.

Following this principle, Ghosh et al. [18] propose a dual approach that searches for an appropriate $\lambda$ within each episode, leading to a tighter violation regret guarantee than Ghosh et al. [16]. However, due to the lack of pessimistic constraint estimation, their method does not ensure episode-wise safety and allows constraint violations. Like Ghosh et al. [18], our OPSE-LCMDP searches for the best $\lambda$ in each episode. However, unlike their approach, OPSE-LCMDP controls $\lambda$ with pessimism, ensuring zero violation, and guarantees the existence of a feasible $\lambda$ by deploying a sufficient number of $\pi^{\text{sf}}$.

## B.2 Related Safety Types

**Instantaneous safety.** Unlike our episode-wise safety, instantaneous safety defines exploration as safe if it satisfies $u_h(s_h^{(k)}, a_h^{(k)}) \geq b$ for all $h$ and $k$ [32, 33, 22, 41, 4]. In other words, states and actions must belong to predefined safe sets, $\mathcal{S}_{\text{sf}} \times \mathcal{A}_{\text{sf}}$. Instantaneous safety is a special case of the

episode-wise constraint. Indeed, by defining $u_h(s,a) = -\mathbb{I}\{(s,a) \notin \mathcal{S}_{\text{sf}} \times \mathcal{A}_{\text{sf}}\}$ and setting $b = 0$, an episode-wise safe algorithm safeties the instantaneous constraint for all $h$ and $k$.

**Cancel Safety.** Cancel safety is another common safety measure in CMDP literature Wei et al. [44], Ghosh et al. [16]. It allows a strict constraint satisfaction in one episode to compensate for a violation in another. Formally, cancel safety ensures that the following cumulative **cancel violation regret** remains non-positive:

$$\text{Vio}_{\text{cancel}}(K) := \sum_{k=1}^{K} b - V_{P,1}^{\pi^{(k)},u}(s_1) \ .$$

Note that the "hard" violation regret $\text{Vio}_{\text{hard}}(K) := \sum_{k=1}^{K} \max\{b - V_{P,1}^{\pi^{(k)},u}(s_1), 0\}$ which considers violations in each individual episode [18, 13, 29], always upper-bounds the cancel regret. This means cancel regret is a weaker measure. Since episode-wise safety ensures $\text{Vio}_{\text{hard}} = 0$, our OPSE-LCMDP always satisfies cancel safety, but cancel safety does not necessarily guarantee episode-wise safety.

## C  Useful Lemmas

**Definition 6.** For a set of positive values $\{a_n\}_{n=1}^N$, we write $x = \text{polylog}(a_1, \dots, a_N)$ if there exists an absolute constants $\{b_n\}_{n=0}^N > 0$ and $\{c_n\}_{n=1}^N > 0$ such that $x \le b_0 + b_1(\ln a_1)^{c_1} + \dots + b_N(\ln a_N)^{c_N}$.

**Definition 7** (Distance metrics). Let $\text{dist}_\infty$ be the distance metric such that, for two functions $Q, Q' : \mathcal{S} \times \mathcal{A} \to \mathbb{R}$, $\text{dist}_\infty(Q, Q') = \sup_{(s,a) \in \mathcal{S} \times \mathcal{A}} |Q(s,a) - Q'(s,a)|$. Similarly, for two functions $V, V' : \mathcal{S} \to \mathbb{R}$, $\text{dist}_\infty(V, V') = \sup_{s \in \mathcal{S}} |V(s) - V'(s)|$. Finally, $\text{dist}_1$ denotes the distance metric such that, for two functions $\pi, \pi' : \mathcal{S} \to \mathscr{P}(\mathcal{A})$, $\text{dist}_1(\pi, \pi') = \sup_{s \in \mathcal{S}} \|\pi(\cdot \mid s) - \pi'(\cdot \mid s)\|_1$.

**Definition 8** ($\varepsilon$-cover). Let $\Theta = \{\boldsymbol{\theta} \in \mathbb{R}^d : \|\boldsymbol{\theta}\|_2 \le R\}$ be a ball with radius $R$. Fix an $\varepsilon$. An $\varepsilon$-net $\mathcal{M}_\varepsilon \subset \Theta$ is a finite set such that for any $\boldsymbol{\theta} \in \Theta$, there exists a $\boldsymbol{\theta}' \in \mathcal{M}_\varepsilon$ such that $\text{dist}(\boldsymbol{\theta}, \boldsymbol{\theta}') \le \varepsilon$ for some distance metric $\text{dist}(\cdot, \cdot)$. The smallest $\varepsilon$-net is called $\varepsilon$-cover and denoted as $\mathcal{N}_\varepsilon$. The size of the $\varepsilon$-net is called the $\varepsilon$-covering number.

**Lemma 9** (**Lemma 5.2** in Vershynin [43]). *The $\varepsilon$-covering number of the ball $\Theta = \{\theta \in \mathbb{R}^d : \|\theta\|_2 \le R\}$ with the distance metric $\|\cdot\|_2$ is upper bounded by $(1 + 2R/\varepsilon)^d$.*

**Lemma 10** (Danskin's Theorem [6]). *Let $f : \mathbb{R}^n \times \mathcal{Z} \to \mathbb{R}$ be a continuous function where $\mathcal{Z} \in \mathbb{R}^m$ is a compact set and $g(x) := \max_{z \in \mathcal{Z}} f(x, z)$.*
*Let $\mathcal{Z}_0(x) := \{\bar{z} \mid f(x, \bar{z}) = \max_{z \in \mathcal{Z}} f(x, z)\}$ be the maximizing points of $f(x, z)$. Assume that $f(x, z)$ is convex in $x$ for every $z \in \mathcal{Z}$. Then, $g(x)$ is convex. Furthermore, if $\mathcal{Z}_0(x)$ consists of a single element $\bar{z}$, i.e., $\mathcal{Z}_0(x) = \{\bar{z}\}$, it holds that $\frac{\partial g(x)}{\partial x} = \frac{\partial f(x, \bar{z})}{\partial x}$.*

**Lemma 11** (**Lemma D.4** in Rosenberg et al. [37]). *Let $\left(X^{(k)}\right)_{k=1}^\infty$ be a sequence of random variables with expectation adapted to the filtration $\left(\mathcal{F}^{(k)}\right)_{k=0}^\infty$. Suppose that $0 \le X^{(k)} \le B$ almost surely. Then, with probability at least $1 - \delta$, the following holds for all $k \ge 1$ simultaneously:*

$$\sum_{i=1}^{k} \mathbb{E}\left[X^{(i)} \mid \mathcal{F}^{(i-1)}\right] \le 2\sum_{i=1}^{k} X^{(i)} + 4B \ln \frac{2k}{\delta}$$

**Lemma 12** (**Lemma 11** in Abbasi-Yadkori et al. [1]). *Let $\{\mathbf{x}^{(k)}\}_{k=1}^K$ be a sequence in $\mathbb{R}^d$. Let $\boldsymbol{\Lambda}^{(k)} = \rho \mathbf{I} + \sum_{i=1}^{k-1} \mathbf{x}^{(i)}\left(\mathbf{x}^{(i)}\right)^\top$. If $\left\|\mathbf{x}^{(k)}\right\|_2 \le B$ for all $k$,*

$$\sum_{k=1}^{K} \min\left\{1, \left\|\mathbf{x}^{(k)}\right\|_{\left(\boldsymbol{\Lambda}^{(k)}\right)^{-1}}^2\right\} \le 2d \ln\left(\frac{\rho d + KB^2}{\rho d}\right) \ .$$

Additionally, if $\left\|\mathbf{x}^{(k)}\right\|_2 \leq 1$ for all $k$ and $\rho \geq 1$[9], we have

$$\sum_{k=1}^{K} \left\|\mathbf{x}^{(k)}\right\|_{\left(\boldsymbol{\Lambda}^{(k)}\right)^{-1}}^{2} \leq 2d \ln\left(\frac{\rho d + K}{\rho d}\right) \ .$$

**Lemma 13 (Theorem 2** in Abbasi-Yadkori et al. [1]**).** *Let $\left\{\mathcal{F}^{(k)}\right\}_{k=0}^{\infty}$ be a filtration. Let $\left\{\varepsilon^{(k)}\right\}_{k=1}^{\infty}$ be a real-valued stochastic process such that $\varepsilon^{(k)}$ is $\mathcal{F}^{(k)}$-measurable and $\varepsilon^{(k)}$ is conditionally $R$-sub-Gaussian for some $R \geq 0$. Let $\left\{\boldsymbol{\phi}^{(k)}\right\}_{k=1}^{\infty}$ be an $\mathbb{R}^d$-valued stochastic process such that $\boldsymbol{\phi}^{(k)}$ is $\mathcal{F}^{(k-1)}$ measurable and $\|\boldsymbol{\phi}^{(k)}\|_2 \leq L$ for all $k$. For any $k \geq 0$, define $Y_k := \boldsymbol{\theta}^{\top} \boldsymbol{\phi}^{(k)} + \varepsilon_t$ for some $\boldsymbol{\theta} \in \mathbb{R}^d$ such that $\|\boldsymbol{\theta}\|_2 \leq B$, $\boldsymbol{\Lambda}^{(k)} := \rho \mathbf{I} + \sum_{i=1}^{k} \boldsymbol{\phi}^{(i)} \left(\boldsymbol{\phi}^{(i)}\right)^{\top}$, and $\widehat{\boldsymbol{\theta}}^{(k)} := \left(\boldsymbol{\Lambda}^{(k)}\right)^{-1} \sum_{i=1}^{k} \boldsymbol{\phi}^{(i)} Y^{(i)}$. Then for any $\delta > 0$, with probability at least $1 - \delta$, for all $k \geq 0$, we have*

$$\left\|\widehat{\boldsymbol{\theta}}^{(k)} - \boldsymbol{\theta}\right\|_{\boldsymbol{\Lambda}^{(k)}} \leq \rho^{1/2} B + R\sqrt{d \ln\left(\frac{1 + kL^2/\rho}{\delta}\right)} \ .$$

**Lemma 14 (Lemma D.4** in Jin et al. [23]**).** *Let $\left\{s^{(k)}\right\}_{k=1}^{\infty}$ be a stochastic process on state space $\mathcal{S}$ with corresponding filtration $\left\{\mathcal{F}^{(k)}\right\}_{k=0}^{\infty}$. Let $\left\{\boldsymbol{\phi}^{(k)}\right\}_{k=0}^{\infty}$ be an $\mathbb{R}^d$-valued stochastic process where $\boldsymbol{\phi}^{(k)}$ is $\mathcal{F}^{(k-1)}$-measurable and $\left\|\boldsymbol{\phi}^{(k)}\right\| \leq 1$. Let $\boldsymbol{\Lambda}^{(k)} = \rho \mathbf{I} + \sum_{k=1}^{k} \boldsymbol{\phi}^{(k)} \left(\boldsymbol{\phi}^{(k)}\right)^{\top}$ and let $\mathcal{V}$ be a class of real-valued function over the state space $\mathcal{S}$ such that $\sup_s |V(s)| \leq B$ for a $B > 0$. Let $\mathcal{N}_\varepsilon^{\mathcal{V}}$ be the $\varepsilon$-cover of $\mathcal{V}$ with respect to the distance $\mathrm{dist}_{\infty}$. Then for any $\delta > 0$, with probability at least $1 - \delta$, for all $K \geq 0$, and any $V \in \mathcal{V}$, we have:*

$$\left\|\sum_{k=1}^{K} \boldsymbol{\phi}^{(k)} \left(V\left(s^{(k)}\right) - \mathbb{E}\left[V\left(s^{(k)}\right) \mid \mathcal{F}^{(k-1)}\right]\right)\right\|_{\left(\boldsymbol{\Lambda}^{(k)}\right)^{-1}}^{2} \leq 4B^2\left(\frac{d}{2}\ln\left(\frac{K+\rho}{\rho}\right) + \ln\frac{|\mathcal{N}_\varepsilon^{\mathcal{V}}|}{\delta}\right) + \frac{8K^2\varepsilon^2}{\rho} \ .$$

**Lemma 15 (Lemma A.1** in Shalev-Shwartz and Ben-David [39]**).** *Let $a > 0$. Then, $x \geq 2a\ln(a)$ yields $x \geq a\ln(x)$. It follows that a necessary condition for the inequality $x \leq a\ln(x)$ to hold is that $x \leq 2a\ln(a)$.*

**Lemma 16.** *For any positive real numbers $x_1, x_2, \ldots, x_n$, $\sum_{i=1}^{n} \sqrt{x_i} \leq \sqrt{n}\sqrt{\sum_{i=1}^{n} x_i}$.*

*Proof.* Due to the Cauchy-Schwarz inequality, we have $\left(\frac{\sum_{i=1}^{n} \sqrt{x_i}}{n}\right)^2 \leq \frac{\sum_{i=1}^{n} x_i}{n}$. Taking the square root of the inequality proves the claim. $\square$

**Lemma 17 (Lemma 1** in Shani et al. [40]**).** *Let $\widetilde{\pi}, \pi$ be two policies, $P$ be a transition kernel, and $g$ be a reward function. Let $\widetilde{V}_h^\pi : \mathcal{S} \rightarrow \mathbb{R}$ be a function such that*

$$\widetilde{V}_h^\pi(s) = \sum_{a \in \mathcal{A}} \widetilde{\pi}_h(a \mid s) \widetilde{Q}_h(s, a) \ ,$$

*for all $h \in [\![1, H]\!]$ with some function $\widetilde{Q}_h : [\![1, H]\!] \times \mathcal{S} \times \mathcal{A} \rightarrow \mathbb{R}$. Then, for any $(h, s) \in [\![1, H]\!] \times \mathcal{S}$*

$$\widetilde{V}_h^{\widetilde{\pi}}(s) - V_{P,h}^{\pi,g}(s) = V_{P,h}^{\pi,g^1}(s) + V_{P,h}^{\pi,g^2}(s) \ ,$$

*where $g^1$ and $g^2$ are reward functions such that*

$$g_h^1(s, a) = \sum_{a \in \mathcal{A}} (\widetilde{\pi}_h(a \mid s) - \pi_h(a \mid s)) \widetilde{Q}_h(s, a) \quad \text{and} \quad g_h^2(s, a) = \widetilde{Q}_h(s, a) - g_h(s, a) - \left(P_h \widetilde{V}_{h+1}^{\widetilde{\pi}}\right)(s, a) \ .$$

---

[9]The second argument follows since $\|\mathbf{x}\|_{\boldsymbol{\Lambda}^{-1}}^2 \leq \sigma_{\max}\left(\boldsymbol{\Lambda}^{-1}\right)\|\mathbf{x}\|^2 \leq \rho^{-1} \leq 1$, where $\sigma_{\max}(\boldsymbol{\Lambda}^{-1})$ denotes the maximum eigen value of $\boldsymbol{\Lambda}^{-1}$.

**Lemma 18** (Regularized value difference lemma). *Let $\kappa \geq 0$ be a non-negative value, $\pi, \pi'$ be two policies, $P$ be a transition kernel, and $g$ be a reward function. Let $\widetilde{V}_h^{\widetilde{\pi}}[\kappa] : \mathcal{S} \to \mathbb{R}$ be a function such that*

$$\widetilde{V}_h^{\widetilde{\pi}}[\kappa](s) = \sum_{a \in \mathcal{A}} \widetilde{\pi}_h(a \mid s)\Big(\widetilde{Q}_h(s,a) - \kappa \ln \widetilde{\pi}_h(a \mid s)\Big) ,$$

*for all $h \in [\![1, H]\!]$ with some function $\widetilde{Q}_h : [\![1, H]\!] \times \mathcal{S} \times \mathcal{A} \to \mathbb{R}$. Then, for any $(h, s) \in [\![1, H]\!] \times \mathcal{S}$*

$$\widetilde{V}_h^{\widetilde{\pi}}[\kappa](s) - V_{P,h}^{\pi,g}[\kappa](s) = V_{P,h}^{\pi,f^1}(s) + V_{P,h}^{\pi,f^2}(s) ,$$

*where $f^1$ and $f^2$ are reward functions such that*

$$f_h^1(s,a) = \sum_{a \in \mathcal{A}} \widetilde{\pi}_h(a \mid s)\Big(\widetilde{Q}_h(s,a) - \kappa \ln \widetilde{\pi}_h(a \mid s)\Big) - \pi_h(a \mid s)\Big(\widetilde{Q}_h(s,a) - \kappa \ln \pi_h(a \mid s)\Big)$$

*and* $f_h^2(s,a) = \widetilde{Q}_h(s,a) - g_h(s,a) - \Big(P_h \widetilde{V}_{h+1}^{\widetilde{\pi}}[\kappa]\Big)(s,a)$ .

*Proof.* Since

$$\widetilde{V}_h^{\widetilde{\pi}}[\kappa](s) = \sum_{a \in \mathcal{A}} \widetilde{\pi}_h(a \mid s)\Big(\widetilde{Q}_h(s,a) - \kappa \ln \widetilde{\pi}_h(a \mid s)\Big) \text{ and } V_{P,h}^{\pi,g}[\kappa](s) = V_{P,h}^{\pi,g-\kappa \ln \pi}(s) ,$$

using Lemma 17, we have

$$\widetilde{V}_1^{\widetilde{\pi}}[\kappa](s_1) - V_{P,1}^{\pi,g}[\kappa](s_1) = V_{P,1}^{\pi,g^1}(s_1) + V_{P,1}^{\pi,g^2}(s_1) ,$$

where $g^1$ and $g^2$ are reward functions such that

$$g_h^1(s,a) = \sum_{a \in \mathcal{A}} (\widetilde{\pi}_h(a \mid s) - \pi_h(a \mid s))\Big(\widetilde{Q}_h(s,a) - \kappa \ln \widetilde{\pi}_h(a \mid s)\Big)$$

$$= \sum_{a \in \mathcal{A}} \widetilde{\pi}_h(a \mid s)\Big(\widetilde{Q}_h(s,a) - \kappa \ln \widetilde{\pi}_h(a \mid s)\Big) - \pi_h(a \mid s)\Big(\widetilde{Q}_h(s,a) - \kappa \ln \pi_h(a \mid s)\Big)$$

$$+ \underbrace{\sum_{a \in \mathcal{A}} \pi_h(a \mid s)(\kappa \ln \widetilde{\pi}_h(a \mid s) - \kappa \ln \pi_h(a \mid s))}_{(a)}$$

and $g_h^2(s,a) = \widetilde{Q}_h(s,a) - g_h(s,a) - \Big(P_h \widetilde{V}_{h+1}^{\widetilde{\pi}}[\kappa]\Big)(s,a) \underbrace{-\kappa \ln \widetilde{\pi}_h(a \mid s) + \kappa \ln \pi_h(a \mid s)}_{(b)}$ .

The claim holds since the terms (a) and (b) are canceled out in $V_{P,h}^{\pi,g^1}(s) + V_{P,h}^{\pi,g^2}(s)$. $\qquad\square$

**Lemma 19.** *Let $Q, \widetilde{Q} : \mathcal{A} \to \mathbb{R}$ be two functions. Let $\kappa > 0$ be a positive constant. Define two softmax distributions $\pi, \widetilde{\pi} \in \mathscr{P}(\mathcal{A})$ such that $\pi = \mathrm{SoftMax}\big(\frac{Q}{\kappa}\big)$ and $\widetilde{\pi} = \mathrm{SoftMax}\big(\frac{\widetilde{Q}}{\kappa}\big)$. Then, $\|\pi - \widetilde{\pi}\|_1 \leq \frac{8}{\kappa}\big\|Q - \widetilde{Q}\big\|_\infty$.*

*Proof.* It holds that

$$\frac{1}{2}\|\pi - \widetilde{\pi}\|_1 \overset{(a)}{\leq} 2 \sum_{a \in \mathcal{A}} \pi(a)|\ln \pi(a) - \ln \widetilde{\pi}(a)| \leq 2 \max_a |\ln \pi(a) - \ln \widetilde{\pi}(a)|$$

$$= 2 \max_a \left|\frac{1}{\kappa}Q(a) - \frac{1}{\kappa}\widetilde{Q}(a) - \ln \sum_a \exp\left(\frac{1}{\kappa}Q(a)\right) + \ln \sum_a \exp\left(\frac{1}{\kappa}\widetilde{Q}(a)\right)\right|$$

$$\leq 2 \max_a \left|\frac{1}{\kappa}Q(a) - \frac{1}{\kappa}\widetilde{Q}(a)\right| + 2\left|\ln \sum_a \exp\left(\frac{1}{\kappa}Q(a)\right) - \ln \sum_a \exp\left(\frac{1}{\kappa}\widetilde{Q}(a)\right)\right|$$

$$\overset{(b)}{\leq} 4 \max_a \left|\frac{1}{\kappa}Q(a) - \frac{1}{\kappa}\widetilde{Q}(a)\right| ,$$

where (a) uses **Theorem 17** in Sason and Verdú [38] and (b) uses the fact that $\ln \sum_i \exp(\mathbf{x}_i) - \ln \sum_i \exp(\mathbf{y}_i) \leq \max_i(\mathbf{x}_i - \mathbf{y}_i)$ (see, e.g., **Theorem 1** in Dutta and Furuichi [12]). This concludes the proof. $\qquad\square$

# D   Regret Analysis (Linear Constrained Bandit)

**Lemma 20** (Good event 1). *Suppose Algorithm 1 is run with $\rho = 1$. Let $\delta \in (0, 1]$. Define $\mathcal{E}_1$ as the event where the following inequality holds:*

$$\sum_{k=1}^{K} \mathbb{E}_{\mathbf{a} \sim \pi^{(k)}} \|\mathbf{a}\|^2_{\left(\mathbf{\Lambda}^{(k)}\right)^{-1}} \leq 2 \sum_{k=1}^{K} \left\|\mathbf{a}^{(k)}\right\|^2_{\left(\mathbf{\Lambda}^{(k)}\right)^{-1}} + 4 \ln \frac{2K}{\delta} .$$

*Then, $\mathbb{P}(\mathcal{E}_1) \geq 1 - \delta$.*

*Proof.* The claim immediately follows from Lemma 11 with $\|\mathbf{a}\|_2 \leq 1$ and $\rho = 1$. $\qquad\square$

**Lemma 21** (Good event 2). *Define $\mathcal{E}_2$ as the event where the following two hold: For any $\pi \in \Pi$, $k \in [\![1, K]\!]$,*

$$\left|\widehat{r}^{(k)}_{\pi} - r_{\pi}\right| \leq C_u \beta^{(k)}_{\pi} \quad and \quad \left|\widehat{u}^{(k)}_{\pi} - u_{\pi}\right| \leq C_u \beta^{(k)}_{\pi} .$$

*Then, if Algorithm 1 is run with $\rho = 1$ and the value of $C_u \geq B + R\sqrt{d \ln \frac{4K}{\delta}}$, it holds that $\mathbb{P}(\mathcal{E}_2) \geq 1 - \delta$.*

*Proof.* Using Lemma 13 with $\rho = 1$, with probability at least $1 - \delta$, for any $k \in [\![1, K]\!]$ and for both $g \in \{r, u\}$, we have

$$\left|\mathbf{a}^{\top}\left(\widehat{\boldsymbol{\theta}}^{(k)}_g - \boldsymbol{\theta}_g\right)\right| \leq \left\|\widehat{\boldsymbol{\theta}}^{(k)}_g - \boldsymbol{\theta}_g\right\|_{\mathbf{\Lambda}^{(k)}} \|\mathbf{a}\|_{\left(\mathbf{\Lambda}^{(k)}\right)^{-1}}$$
$$\overset{(a)}{\leq} \left(B + R\sqrt{d \ln \frac{2(1 + K)}{\delta}}\right) \|\mathbf{a}\|_{\left(\mathbf{\Lambda}^{(k)}\right)^{-1}}$$
$$\leq \left(B + R\sqrt{d \ln \frac{4K}{\delta}}\right) \|\mathbf{a}\|_{\left(\mathbf{\Lambda}^{(k)}\right)^{-1}} ,$$

where (a) uses Lemma 13. The claim holds by $\left|\widehat{g}^{(k)}_{\pi} - g_{\pi}\right| \leq \mathbb{E}_{\mathbf{a} \sim \pi}\left|\mathbf{a}^{\top}\left(\widehat{\boldsymbol{\theta}}^{(k)}_g - \boldsymbol{\theta}_g\right)\right|$ for $g \in \{r, u\}$. $\qquad\square$

**Lemma 22** (Cumulative bonus bound). *Suppose $\mathcal{E}_1$ holds. Then, $\sum_{k=1}^{K} \beta^{(k)}_{\pi^{(k)}} \leq \sqrt{K}\sqrt{2d \ln\left(1 + \frac{K}{d}\right) + 4 \ln \frac{2K}{\delta}}$.*

*Proof.* It holds that

$$\sum_{k=1}^{K} \beta^{(k)}_{\pi^{(k)}} \overset{(a)}{\leq} \sqrt{K \sum_{k=1}^{K} \left(\mathbb{E}_{\mathbf{a} \sim \pi^{(k)}} \|\mathbf{a}\|_{\left(\mathbf{\Lambda}^{(k)}\right)^{-1}}\right)^2} \overset{(b)}{\leq} \sqrt{K \sum_{k=1}^{K} \mathbb{E}_{\mathbf{a} \sim \pi^{(k)}} \|\mathbf{a}\|^2_{\left(\mathbf{\Lambda}^{(k)}\right)^{-1}}}$$
$$\overset{(c)}{\leq} \sqrt{K}\sqrt{2 \sum_{k=1}^{K} \left\|\mathbf{a}^{(k)}\right\|^2_{\left(\mathbf{\Lambda}^{(k)}\right)^{-1}} + 4 \ln \frac{2K}{\delta}} \overset{(d)}{\leq} \sqrt{K}\sqrt{2d \ln\left(1 + \frac{K}{d}\right) + 4 \ln \frac{2K}{\delta}} ,$$

where (a) and (b) use Cauchy–Schwarz inequality, (c) is due to $\mathcal{E}_1$, and (d) uses Lemma 12. $\qquad\square$

**Lemma 23** (Restatement of Lemma 1). *Suppose $\mathcal{E}_2$ holds. Then, for any $\pi \in \Pi$ and $k \in [\![1, K]\!]$,*

$$r_{\pi} + 2C_u \beta^{(k)}_{\pi} \geq \widehat{r}^{(k)}_{\pi} + C_u \beta^{(k)}_{\pi} \geq r_{\pi} \quad and \quad u_{\pi} \geq \widehat{u}^{(k)}_{\pi} - C_u \beta^{(k)}_{\pi} \geq u_{\pi} - 2C_u \beta^{(k)}_{\pi} .$$

*Proof.* We have

$$u_{\pi} \geq \widehat{u}^{(k)}_{\pi} - \left|\widehat{u}^{(k)}_{\pi} - u_{\pi}\right| \geq \widehat{u}^{(k)}_{\pi} - C_u \beta^{(k)}_{\pi} \geq \widehat{u}^{(k)}_{\pi} - \left|\widehat{u}^{(k)}_{\pi} - u_{\pi}\right| - C_u \beta^{(k)}_{\pi} \geq u_{\pi} - 2C_u \beta^{(k)}_{\pi} .$$

Similarly,

$$r_{\pi} + 2C_u \beta^{(k)}_{\pi} \geq \widehat{r}^{(k)}_{\pi} + \left|\widehat{r}^{(k)}_{\pi} - r_{\pi}\right| + C_u \beta^{(k)}_{\pi} \geq \widehat{r}^{(k)}_{\pi} + C_u \beta^{(k)}_{\pi} \geq \widehat{r}^{(k)}_{\pi} + \left|\widehat{r}^{(k)}_{\pi} - r_{\pi}\right| \geq r_{\pi} .$$

$\qquad\square$

**Lemma 24** (Restatement of Lemma 2). *Consider $k \in \mathcal{U}^{\complement}$. For any $\alpha \in \left[0, \frac{\xi - 2C_u \beta_{\pi^{\mathrm{sf}}}^{(k)}}{\xi - 2C_u \beta_{\pi^{\mathrm{sf}}}^{(k)} + 2C_u \beta_{\pi^\star}^{(k)}}\right]$, a mixture policy $\pi_\alpha := (1 - \alpha)\pi^{\mathrm{sf}} + \alpha\pi^\star$ satisfies $u_{\pi_\alpha} - 2C_u \beta_{\pi_\alpha}^{(k)} \geq b$.*

*Proof.* For any $k$ and $\alpha \in [0, 1]$, we have

$$u_{\pi_\alpha} - b - 2C_u \beta_{\pi_\alpha}^{(k)} = (1 - \alpha)\underbrace{(u_{\pi^{\mathrm{sf}}} - b)}_{\geq \xi} + \alpha \underbrace{(u_{\pi^\star} - b)}_{\geq 0} - 2C_u(1 - \alpha)\beta_{\pi^{\mathrm{sf}}}^{(k)} - 2C_u\alpha\beta_{\pi^\star}^{(k)}$$

$$\geq (1 - \alpha)\left(\xi - 2C_u\beta_{\pi^{\mathrm{sf}}}^{(k)}\right) - 2\alpha C_u\beta_{\pi^\star}^{(k)}.$$

To make $(1 - \alpha)\left(\xi - 2C_u\beta_{\pi^{\mathrm{sf}}}^{(k)}\right) - 2\alpha C_u\beta_{\pi^\star}^{(k)} \geq 0$, a sufficient condition is

$$\alpha \leq \frac{\xi - 2C_u\beta_{\pi^{\mathrm{sf}}}^{(k)}}{\xi - 2C_u\beta_{\pi^{\mathrm{sf}}}^{(k)} + 2C_u\beta_{\pi^\star}^{(k)}}, \tag{9}$$

where the right hand side is non-negative since $k \in \mathcal{U}^{\complement}$. This concludes the proof. $\qquad\square$

**Lemma 25** (Restatement of Theorem 1). *Suppose Algorithm 1 is run with $\rho = 1$. Assume the event $\mathscr{E}_1$ holds. Then, $|\mathcal{U}| \leq 32dC_u^2\xi^{-2}\ln(2K\delta^{-1})$.*

*Proof.* We have

$$\sum_{k=1}^{K} \mathbb{E}_{\mathbf{a} \sim \pi^{(k)}} \|\mathbf{a}\|_{\left(\mathbf{\Lambda}^{(k)}\right)^{-1}}^{2} \geq \sum_{k \in \mathcal{U}} \mathbb{E}_{\mathbf{a} \sim \pi^{(k)}} \|\mathbf{a}\|_{\left(\mathbf{\Lambda}^{(k)}\right)^{-1}}^{2}$$

$$\overset{(a)}{\geq} \sum_{k \in \mathcal{U}} \underbrace{\left(\mathbb{E}_{\mathbf{a} \sim \pi^{(k)}} \|\mathbf{a}\|_{\left(\mathbf{\Lambda}^{(k)}\right)^{-1}}\right)^{2}}_{= \left(\beta_{\pi^{\mathrm{sf}}}^{(k)}\right)^2 \text{ since } \pi^{(k)} = \pi^{\mathrm{sf}}} \overset{(b)}{\geq} |\mathcal{U}|\frac{\xi^2}{4C_u^2},$$

where (a) is due to Jensen's inequality, and (b) is due to Definition 1. Due to $\mathscr{E}_1$, we have

$$\sum_{k=1}^{K} \mathbb{E}_{\mathbf{a} \sim \pi^{(k)}} \|\mathbf{a}\|_{\left(\mathbf{\Lambda}^{(k)}\right)^{-1}}^{2} \leq 2\sum_{k=1}^{K} \left\|\mathbf{a}^{(k)}\right\|_{\left(\mathbf{\Lambda}^{(k)}\right)^{-1}}^{2} + 4\ln\frac{2K}{\delta}.$$

Using Lemma 12 and since $\|\mathbf{a}\|_2 \leq 1$ and $\rho = 1$, the first term is bounded by: $\leq 2d\ln\left(1 + \frac{K}{d}\right)$. Thus,

$$\frac{\xi^2}{4C_u^2}|\mathcal{U}| \leq 2d\ln\underbrace{\left(1 + \frac{K}{d}\right)}_{\leq 2K} + 4\ln\frac{2K}{\delta} \leq 8d\ln\left(\frac{2K}{\delta}\right).$$

The claim holds by rearranging the above inequality. $\qquad\square$

**Lemma 26** (Restatement of Lemma 3). *For any $k \in \mathcal{U}^{\complement}$, $\pi_{\alpha^{(k)}}$ satisfies $r_{\pi_{\alpha^{(k)}}} + \frac{2BC_u}{\xi}\beta_{\pi_{\alpha^{(k)}}}^{(k)} \geq r_{\pi^\star}$.*

*Proof.* Let $\alpha^{(k)} := \frac{\xi - 2C_u\beta_{\pi^{\mathrm{sf}}}^{(k)}}{\xi - 2C_u\beta_{\pi^{\mathrm{sf}}}^{(k)} + 2C_u\beta_{\pi^\star}^{(k)}}$ and $C := \frac{2BC_u}{\xi}$. Note that $\frac{\alpha^{(k)}}{1 - \alpha^{(k)}} = \frac{\xi - 2C_u\beta_{\pi^{\mathrm{sf}}}^{(k)}}{2C_u\beta_{\pi^\star}^{(k)}}$. We have,

$$r_{\pi_{\alpha^{(k)}}} + C\beta_{\pi_{\alpha^{(k)}}}^{(k)} = (1 - \alpha^{(k)})r_{\pi^{\mathrm{sf}}} + \alpha^{(k)}r_{\pi^\star} + C(1 - \alpha^{(k)})\beta_{\pi^{\mathrm{sf}}}^{(k)} + C\alpha^{(k)}\beta_{\pi^\star}^{(k)}$$

$$\geq \alpha^{(k)}r_{\pi^\star} + C\left(\left(1 - \alpha^{(k)}\right)\beta_{\pi^{\mathrm{sf}}}^{(k)} + \alpha^{(k)}\beta_{\pi^\star}^{(k)}\right).$$

A sufficient condition to have $\alpha^{(k)}r_{\pi^\star} + C\left(\left(1 - \alpha^{(k)}\right)\beta_{\pi^{\mathrm{sf}}}^{(k)} + \alpha^{(k)}\beta_{\pi^\star}^{(k)}\right) \geq r_{\pi^\star}$ is, since $r_{\pi^\star} = \mathbb{E}_{\mathbf{a} \sim \pi^\star}[\langle \boldsymbol{\theta}, \mathbf{a}\rangle] \leq \|\boldsymbol{\theta}\|_2 \mathbb{E}_{\mathbf{a} \sim \pi^\star}\|\mathbf{a}\|_2 \leq B$,

$$B \leq C\left(\beta_{\pi^{\mathrm{sf}}}^{(k)} + \frac{\alpha^{(k)}}{1 - \alpha^{(k)}}\beta_{\pi^\star}^{(k)}\right)$$

$$= C\left(\beta_{\pi^{\mathrm{sf}}}^{(k)} + \frac{1}{2C_u}\xi - \beta_{\pi^{\mathrm{sf}}}^{(k)}\right) \leq \frac{C}{2C_u}\xi.$$

Therefore, when $C \geq \frac{2BC_u}{\xi}$, we have $r_{\pi_{\alpha^{(k)}}} + C\beta_{\pi_{\alpha^{(k)}}}^{(k)} \geq r_{\pi^\star}$. $\qquad\square$

**Theorem 5** (Restatement of Theorem 2). *Suppose that Algorithm 1 is run with $\rho = 1$,*

$$C_u = B + R\sqrt{d\ln\frac{4K}{\delta}}, \quad \text{and } C_r = C_u\left(1 + \frac{2B}{\xi}\right).$$

*Then, with probability at least $1 - 2\delta$, the following two hold simultaneously:*

- $\pi^{(k)} \in \Pi^{\mathrm{sf}}$ *for any $k \in [K]$*

- $\mathrm{Regret}(K) \leq 64dBC_u^2\xi^{-2}\ln(2K\delta^{-1}) + 4C_r\sqrt{K}\sqrt{2d\ln(1 + \frac{K}{d}) + 4\ln\frac{2K}{\delta}}$

*Proof.* Suppose the good events $\mathscr{E}_1 \cap \mathscr{E}_2$ hold. Recall that $\pi^{(k)}$ is either $\pi^{\mathrm{sf}}$ in $k \in \mathcal{U}$ or the solution to Opt-Pes in $k \in \mathcal{U}^{\complement}$. Since Opt-Pes is ensured to have feasible solutions by Lemma 24 for $k \in \mathcal{U}^{\complement}$, the first claim follows immediately.

We will prove the second claim. It holds that

$$\mathrm{Regret}(K) = \sum_{k=1}^K r_{\pi^\star} - r_{\pi^{(k)}} = \underbrace{\sum_{k\in\mathcal{U}} r_{\pi^\star} - r_{\pi^{(k)}}}_{\pi^{(k)} = \pi^{\mathrm{sf}}} + \underbrace{\sum_{k\notin\mathcal{U}} r_{\pi^\star} - r_{\pi^{(k)}}}_{\pi^{(k)} \text{ is computed by Opt-Pes}}$$

$$\leq 2B|\mathcal{U}| + \sum_{k\notin\mathcal{U}} r_{\pi^\star} - r_{\pi^{(k)}}$$

$$\overset{(a)}{\leq} 64dBC_u^2\xi^{-2}\ln(2K\delta^{-1}) + \underbrace{\sum_{k\notin\mathcal{U}}\left(r_{\pi^\star} - \widehat{r}_{\pi^{(k)}}^{(k)} - C_r\beta_{\pi^{(k)}}^{(k)}\right)}_{\text{①}} + \underbrace{\sum_{k\notin\mathcal{U}}\left(\widehat{r}_{\pi^{(k)}}^{(k)} + C_r\beta_{\pi^{(k)}}^{(k)} - r_{\pi^{(k)}}\right)}_{\text{②}},$$

where (a) uses the bound of $|\mathcal{U}|$ (Lemma 25). Using Lemma 23, the term ② is bounded by ② $\leq \sum_{k\notin\mathcal{U}} 3C_r\beta_{\pi^{(k)}}^{(k)}$. On the other hand, ① is bounded by

$$\text{①} \overset{(a)}{\leq} \sum_{k\notin\mathcal{U}} r_{\pi_{\alpha^{(k)}}} + \frac{2BC_u}{\xi}\beta_{\pi_{\alpha^{(k)}}}^{(k)} - \widehat{r}_{\pi^{(k)}}^{(k)} - C_r\beta_{\pi^{(k)}}^{(k)}$$

$$\overset{(b)}{\leq} \sum_{k\notin\mathcal{U}} \widehat{r}_{\pi_{\alpha^{(k)}}}^{(k)} + C_u\beta_{\pi_{\alpha^{(k)}}}^{(k)} + \frac{2BC_u}{\xi}\beta_{\pi_{\alpha^{(k)}}}^{(k)} - \widehat{r}_{\pi^{(k)}}^{(k)} - C_r\beta_{\pi^{(k)}}^{(k)} \overset{(c)}{\leq} 0,$$

where (a) uses the optimism of mixture policy (Lemma 26), (b) uses Lemma 23, and (c) uses $C_r = (1 + 2BC_u\xi^{-1})C_u$ and since $\pi_{\alpha^{(k)}}$ is a feasible solution to Opt-Pes due to Lemma 24.

Finally, by combining all the results, we have

$$\mathrm{Regret}(K) \leq 64dBC_u^2\xi^{-2}\ln(2K\delta^{-1}) + 3C_r\sum_{k\notin\mathcal{U}}\beta_{\pi^{(k)}}^{(k)}$$

$$\leq 64dBC_u^2\xi^{-2}\ln(2K\delta^{-1}) + 3C_r\sqrt{K}\sqrt{2d\ln\left(1 + \frac{K}{d}\right) + 4\ln\frac{2K}{\delta}}$$

where the second inequality uses Lemma 22. Since the good event $\mathscr{E}_1 \cap \mathscr{E}_2$ occurs with probability at least $1 - 2\delta$ due to Lemmas 20 and 21, the claim holds. □

# E   Regret Analysis (Linear CMDP)

## E.1   Definitions and Useful Lemmas

**Definition 9** ($\mu$-estimator). Let $\mathbf{e}(s) \in \mathbb{R}^\mathcal{S}$ denote a one-hot vector such that only the element at $s \in \mathcal{S}$ is 1 and otherwise 0. In Algorithm 2, for all $h$ and $k$, define $\boldsymbol{\mu}_h^{(k)} \in \mathbb{R}^{S\times d}$ and $\boldsymbol{\epsilon}_h^{(k)} \in \mathbb{R}^\mathcal{S}$ such

that

$$\boldsymbol{\mu}_h^{(k)} := \sum_{i=1}^{k-1} \mathbf{e}\left(s_{h+1}^{(i)}\right)\boldsymbol{\phi}\left(s_h^{(i)}, a_h^{(i)}\right)^\top \left(\boldsymbol{\Lambda}_h^{(k)}\right)^{-1} \text{ and } \boldsymbol{\epsilon}_h^{(k)} := \mathbf{e}\left(s_{h+1}^{(k)}\right) - P\left(\cdot \,\Big|\, s_h^{(k)}, a_h^{(k)}\right). \quad (10)$$

We remark that $\left(\widehat{P}_h^{(k)} V\right)(s, a) = \boldsymbol{\phi}(s, a)^\top \left(\boldsymbol{\mu}_h^{(k)}\right)^\top V$ for any $V \in \mathbb{R}^{\mathcal{S}}$.

**Lemma 27.** *For all $k$ and $h$, it holds that:*

$$\boldsymbol{\mu}_h^{(k)} - \boldsymbol{\mu}_h = -\rho\boldsymbol{\mu}_h \left(\boldsymbol{\Lambda}_h^{(k)}\right)^{-1} + \sum_{i=1}^{k-1} \boldsymbol{\epsilon}_h^{(i)} \boldsymbol{\phi}\left(s_h^{(i)}, a_h^{(i)}\right)^\top \left(\boldsymbol{\Lambda}_h^{(k)}\right)^{-1}$$

*Proof.* Due to the definition of $\boldsymbol{\mu}_h^{(k)}$, we have

$$\boldsymbol{\mu}_h^{(k)} = \sum_{i=1}^{k-1} \mathbf{e}\left(s_{h+1}^{(i)}\right)\boldsymbol{\phi}\left(s_h^{(i)}, a_h^{(i)}\right)^\top \left(\boldsymbol{\Lambda}_h^{(k)}\right)^{-1} = \sum_{i=1}^{k-1}\left(P\left(\cdot \,\Big|\, s_h^{(k)}, a_h^{(k)}\right) + \boldsymbol{\epsilon}_h^{(k)}\right)\boldsymbol{\phi}\left(s_h^{(i)}, a_h^{(i)}\right)^\top \left(\boldsymbol{\Lambda}_h^{(k)}\right)^{-1}$$

$$= \sum_{i=1}^{k-1}\left(\boldsymbol{\mu}_h\boldsymbol{\phi}\left(s_h^{(k)}, a_h^{(k)}\right) + \boldsymbol{\epsilon}_h^{(k)}\right)\boldsymbol{\phi}\left(s_h^{(i)}, a_h^{(i)}\right)^\top \left(\boldsymbol{\Lambda}_h^{(k)}\right)^{-1}$$

$$= \sum_{i=1}^{k-1}\boldsymbol{\mu}_h\boldsymbol{\phi}\left(s_h^{(k)}, a_h^{(k)}\right)\boldsymbol{\phi}\left(s_h^{(i)}, a_h^{(i)}\right)^\top \left(\boldsymbol{\Lambda}_h^{(k)}\right)^{-1} + \sum_{i=1}^{k-1}\boldsymbol{\epsilon}_h^{(k)}\boldsymbol{\phi}\left(s_h^{(i)}, a_h^{(i)}\right)^\top \left(\boldsymbol{\Lambda}_h^{(k)}\right)^{-1}$$

$$= \boldsymbol{\mu}_h\left(\boldsymbol{\Lambda}_h^{(k)} - \rho\mathbf{I}\right)\left(\boldsymbol{\Lambda}_h^{(k)}\right)^{-1} + \sum_{i=1}^{k-1}\boldsymbol{\epsilon}_h^{(k)}\boldsymbol{\phi}\left(s_h^{(i)}, a_h^{(i)}\right)^\top \left(\boldsymbol{\Lambda}_h^{(k)}\right)^{-1}$$

$$= \boldsymbol{\mu}_h - \rho\boldsymbol{\mu}_h\left(\boldsymbol{\Lambda}_h^{(k)}\right)^{-1} + \sum_{i=1}^{k-1}\boldsymbol{\epsilon}_h^{(k)}\boldsymbol{\phi}\left(s_h^{(i)}, a_h^{(i)}\right)^\top \left(\boldsymbol{\Lambda}_h^{(k)}\right)^{-1}.$$

$\square$

**Lemma 28.** *Let $\mathcal{V}$ be a class of real-valued function over the state space $\mathcal{S}$ such that $\sup_s |V(s)| \leq B$ for a $B > 0$. Let $\mathcal{N}_\varepsilon$ be the $\varepsilon$-cover of $\mathcal{V}$ with respect to the distance $\text{dist}_\infty$. In Algorithm 2, for all $k, h, s, a$, for any $V \in \mathcal{V}$, with probability at least $1 - \delta$, we have*

$$\left|\left(\left(\widehat{P}_h^{(k)} - P_h\right)V\right)(s, a)\right|$$

$$\leq \|\boldsymbol{\phi}(s, a)\|_{\left(\boldsymbol{\Lambda}_h^{(k)}\right)^{-1}}\left(\sqrt{d\rho}B + 2B\sqrt{\frac{d}{2}\ln\left(\frac{k+\rho}{\rho}\right)} + 2B\sqrt{\ln\frac{|\mathcal{N}_\varepsilon|}{\delta}} + \frac{4k\varepsilon}{\sqrt{\rho}}\right).$$

*Proof.* Using Lemma 14 and due to the definition of $\boldsymbol{\Lambda}^{(k)}$ in Algorithm 2, with probability at least $1 - \delta$, for all $k, h$, we have

$$\left\|\sum_{i=1}^{k-1}\boldsymbol{\phi}\left(s_h^{(k)}, a_h^{(k)}\right)\left(V^\top \boldsymbol{\epsilon}_h^{(i)}\right)\right\|_{\left(\boldsymbol{\Lambda}^{(k)}\right)^{-1}} \leq \sqrt{4B^2\left(\frac{d}{2}\ln\left(\frac{k+\rho}{\rho}\right) + \ln\frac{|\mathcal{N}_\varepsilon|}{\delta}\right) + \frac{8k^2\varepsilon^2}{\rho}}$$

$$\leq 2B\sqrt{\frac{d}{2}\ln\left(\frac{k+\rho}{\rho}\right)} + 2B\sqrt{\ln\frac{|\mathcal{N}_\varepsilon|}{\delta}} + \frac{4k\varepsilon}{\sqrt{\rho}},$$

where the second inequality uses $\sqrt{a+b} \le \sqrt{a} + \sqrt{b}$. By inserting this to Definition 9, we have

$$\left| \left( \left( \widehat{P}_h^{(k)} - P_h \right) V \right)(s,a) \right|$$

$$= \left| \phi(s,a)^\top \left( \boldsymbol{\mu}_h^{(k)} - \boldsymbol{\mu}_h \right)^\top V \right|$$

$$= \left| \phi(s,a)^\top \left( -\rho \boldsymbol{\mu}_h \left( \boldsymbol{\Lambda}_h^{(k)} \right)^{-1} + \sum_{i=1}^{k-1} \boldsymbol{\epsilon}_h^{(i)} \phi \left( s_h^{(i)}, a_h^{(i)} \right)^\top \left( \boldsymbol{\Lambda}_h^{(k)} \right)^{-1} \right)^\top V \right|$$

$$\le \rho \left| \phi(s,a)^\top \left( \boldsymbol{\Lambda}_h^{(k)} \right)^{-1} (\boldsymbol{\mu}_h)^\top V \right| + \left| \phi(s,a)^\top \left( \boldsymbol{\Lambda}_h^{(k)} \right)^{-1} \sum_{i=1}^{k-1} \phi \left( s_h^{(i)}, a_h^{(i)} \right) \left( V^\top \boldsymbol{\epsilon}_h^{(i)} \right) \right|$$

$$\le \rho \|\phi(s,a)\|_{\left( \boldsymbol{\Lambda}_h^{(k)} \right)^{-1}} \underbrace{\left\| (\boldsymbol{\mu}_h)^\top V \right\|_{\left( \boldsymbol{\Lambda}_h^{(k)} \right)^{-1}}}_{\le B\sqrt{d/\rho} \text{ by Assumption 2}} + \|\phi(s,a)\|_{\left( \boldsymbol{\Lambda}_h^{(k)} \right)^{-1}} \left\| \sum_{i=1}^{k-1} \phi \left( s_h^{(i)}, a_h^{(i)} \right) \left( \boldsymbol{\epsilon}_h^{(i)} \right)^\top V \right\|_{\left( \boldsymbol{\Lambda}_h^{(k)} \right)^{-1}}$$

$$\le \|\phi(s,a)\|_{\left( \boldsymbol{\Lambda}_h^{(k)} \right)^{-1}} \left( \sqrt{d\rho} B + 2B \sqrt{\frac{d}{2} \ln \left( \frac{k+\rho}{\rho} \right)} + 2B \sqrt{\ln \frac{|\mathcal{N}_\varepsilon|}{\delta}} + \frac{4k\varepsilon}{\sqrt{\rho}} \right).$$

$\square$

### E.2 Function Classes and Covering Argument

**Definition 10** ($Q$ function class). For any $h$ and for a pair of $(\mathbf{w}, \boldsymbol{\Lambda})$, where $\mathbf{w} \in \mathbb{R}^d$ and $\boldsymbol{\Lambda} \in \mathbb{R}^{d \times d}$, define $Q_h^{(\mathbf{w},\boldsymbol{\Lambda}),r} : \mathcal{S} \times \mathcal{A} \to \mathbb{R}$, $Q_h^{(\mathbf{w},\boldsymbol{\Lambda}),u} : \mathcal{S} \times \mathcal{A} \to \mathbb{R}$, and $Q_h^{(\mathbf{w},\boldsymbol{\Lambda}),\dagger} : \mathcal{S} \times \mathcal{A} \to \mathbb{R}$ such that

$$Q_h^{(\mathbf{w},\boldsymbol{\Lambda}),r}(s,a) = r_h(s,a) + \text{clip}\left\{ C_r \|\phi(s,a)\|_{\boldsymbol{\Lambda}^{-1}} + \mathbf{w}^\top \phi(s,a), \ 0, \ H_\kappa - h_\kappa \right\}$$

$$Q_h^{(\mathbf{w},\boldsymbol{\Lambda}),u}(s,a) = u_h(s,a) + \text{clip}\left\{ -C_u \|\phi(s,a)\|_{\boldsymbol{\Lambda}^{-1}} + \mathbf{w}^\top \phi(s,a), \ 0, \ H - h \right\}$$

$$Q_h^{(\mathbf{w},\boldsymbol{\Lambda}),\dagger}(s,a) = B_\dagger \|\phi(s,a)\|_{\boldsymbol{\Lambda}^{-1}} + \text{clip}\left\{ C_\dagger \|\phi(s,a)\|_{\boldsymbol{\Lambda}^{-1}} + \mathbf{w}^\top \phi(s,a), \ 0, \ B_\dagger(H - h) \right\},$$

where $\kappa, C_r, C_u, B_\dagger, C_\dagger \ge 0$. We denoted $h_\kappa := h(1 + \kappa \ln A)$ for $h \in [\![1, H]\!]$. Let $\mathcal{Q}_h^r, \mathcal{Q}_h^u, \mathcal{Q}_h^\dagger$ denote function classes such that

$$\mathcal{Q}_h^r := \left\{ Q_h^{(\mathbf{w},\boldsymbol{\Lambda}),r} \ \middle| \ \|\mathbf{w}\|_2 \le KH_\kappa, \ \sigma_{\min}(\boldsymbol{\Lambda}) \ge 1 \right\},$$

$$\mathcal{Q}_h^u := \left\{ Q_h^{(\mathbf{w},\boldsymbol{\Lambda}),u} \ \middle| \ \|\mathbf{w}\|_2 \le KH, \ \sigma_{\min}(\boldsymbol{\Lambda}) \ge 1 \right\},$$

$$\text{and } \mathcal{Q}_h^\dagger := \left\{ Q_h^{(\mathbf{w},\boldsymbol{\Lambda}),\dagger} \ \middle| \ \|\mathbf{w}\|_2 \le KHB_\dagger, \ \sigma_{\min}(\boldsymbol{\Lambda}) \ge 1 \right\}.$$

We let $\mathcal{N}_\varepsilon^{\mathcal{Q}_h^r}, \mathcal{N}_\varepsilon^{\mathcal{Q}_h^u}$, and $\mathcal{N}_\varepsilon^{\mathcal{Q}_h^\dagger}$, be the $\varepsilon$-covers of $\mathcal{Q}_h^r$, $\mathcal{Q}_h^u$, and $\mathcal{Q}_h^\dagger$ with the distance metric $\text{dist}_\infty$.

**Lemma 29** ($Q$ covers). *When Algorithm 2 is run with $\rho = 1$, it hold that:*

(i) *For all $k, h$ and for any $\pi \in \Pi$, $\overline{Q}_{(k),h}^{\pi,r}[\kappa] \in \mathcal{Q}_h^r$, $\underline{Q}_{(k),h}^{\pi,u} \in \mathcal{Q}_h^u$, and $\overline{Q}_{(k),h}^{\pi,\dagger} \in \mathcal{Q}_h^\dagger$*

(ii) $\ln |\mathcal{N}_\varepsilon^{\mathcal{Q}_h^r}| \le d \ln \left( 1 + \frac{4KH_\kappa}{\varepsilon} \right) + d^2 \ln \left( 1 + \frac{8\sqrt{d}C_r^2}{\varepsilon^2} \right) = \mathcal{O}(d^2) \, \text{polylog}\left(d, K, H_\kappa, C_r, \varepsilon^{-1}\right)$,

$\ln |\mathcal{N}_\varepsilon^{\mathcal{Q}_h^u}| \le d \ln \left( 1 + \frac{4KH}{\varepsilon} \right) + d^2 \ln \left( 1 + \frac{8\sqrt{d}C_u^2}{\varepsilon^2} \right) = \mathcal{O}(d^2) \, \text{polylog}\left(d, K, H, C_u, \varepsilon^{-1}\right)$,

*and* $\quad \ln |\mathcal{N}_\varepsilon^{\mathcal{Q}_h^\dagger}| \quad \le \quad d \ln \left( 1 + \frac{4KB_\dagger H}{\varepsilon} \right) \quad + \quad d^2 \ln \left( 1 + \frac{8\sqrt{d}C_\dagger^2}{\varepsilon^2} \right) \quad = $

$\mathcal{O}(d^2) \, \text{polylog}\left(d, K, H, B_\dagger, C_\dagger, \varepsilon^{-1}\right)$

*Proof.* The statements in (ii) immediately follow from the proof of **Lemma D.6** in Jin et al. [23].

We prove the first claim (i). For $\overline{Q}_{(k),h}^{\pi,r}$, we have

$$\overline{Q}_{(k),h}^{\pi,r}[\kappa] = r_h + \text{clip}\left\{C_r\beta^{(k)} + \widehat{P}^{(k)}\overline{V}_{(k),h+1}^{\pi,r}[\kappa],\ 0,\ (H-h)(1+\kappa\ln A)\right\}$$

$$= r_h + \text{clip}\left\{C_r\sqrt{\phi(s,a)^\top\left(\Lambda_h^{(k)}\right)^{-1}\phi(s,a)} + \phi(s,a)^\top\left(\mu_h^{(k)}\right)^\top\overline{V}_{(k),h+1}^{\pi,r}[\kappa],\ 0,\ (H-h)(1+\kappa\ln A)\right\}.$$

According to the definition of $Q_h^{(\mathbf{w},\Lambda),r}$ (Definition 10), the claim immediately holds by showing the L2 bound of $\left(\mu_h^{(k)}\right)^\top\overline{V}_{(k),h+1}^{\pi,r}[\kappa]$. For any $h \in [\![1,H]\!]$ and $k \in [\![1,K]\!]$, we have

$$\left\|\left(\mu_h^{(k)}\right)^\top\overline{V}_{(k),h+1}^{\pi,r}[\kappa]\right\|_2 = \left\|\sum_{i=1}^{k-1}\overline{V}_{(k),h+1}^{\pi,r}[\kappa]\left(s_{h+1}^{(i)}\right)\phi\left(s_h^{(i)},a_h^{(i)}\right)^\top\left(\Lambda_h^{(k)}\right)^{-1}\right\|_2$$

$$\overset{(a)}{\leq} H_\kappa\left\|\left(\Lambda_h^{(k)}\right)^{-1}\sum_{i=1}^{k-1}\phi\left(s_h^{(i)},a_h^{(i)}\right)\right\|_2 \leq KH_\kappa.$$

where (a) uses $\|\phi\|_2 \leq 1$ with $\rho = 1$ and $0 \leq \overline{V}_{(k),h+1}^{\pi,r}[\kappa] \leq H_\kappa$.

The remaining claims for $\underline{Q}_{(k),h}^{\pi,u}(s,a) \in \mathcal{Q}_h^u$ and $\overline{Q}_{(k),h}^{\pi,\dagger}(s,a) \in \mathcal{Q}_h^\dagger$ can be similarly proven. $\square$

**Definition 11** (Composite Q function class). For each $h$, let $\mathcal{Q}_h^\circ$ denote a function class such that

$$\mathcal{Q}_h^\circ := \left\{Q^\dagger + Q^r + \lambda Q^u \mid Q^\dagger \in \mathcal{Q}_h^\dagger,\ Q^r \in \mathcal{Q}_h^r,\ Q^u \in \mathcal{Q}_h^u,\ \text{and}\ \lambda \in [0,C_\lambda]\right\}.$$

where $C_\lambda > 0$. We let $\mathcal{N}_\varepsilon^{\mathcal{Q}_h^\circ}$ be the $\varepsilon$-cover of $\mathcal{Q}_h^\circ$ with the distance metric $\text{dist}_\infty$.

**Lemma 30** (Composite Q cover). *When Algorithm 2 is run with $\rho = 1$, the following statements hold:*

(i) *For all $(k,h)$, for any $\pi \in \Pi$, and for any $\lambda \in [0,C_\lambda]$, $\overline{Q}_{(k),h}^{\pi,\dagger} + \overline{Q}_{(k),h}^{\pi,r}[\kappa] + \lambda\underline{Q}_{(k),h}^{\pi,u} \in \mathcal{Q}_h^\circ$*

(ii) $\ln\left|\mathcal{N}_\varepsilon^{\mathcal{Q}_h^\circ}\right| = \mathcal{O}(d^2)\,\text{polylog}(d,K,H_\kappa,C_r,C_u,B_\dagger,C_\dagger,C_\lambda,\varepsilon^{-1})$

*Proof.* The claim (i) clearly holds by Lemma 29 and Definition 11.

We prove the second claim (ii). Let $\mathcal{N}_\varepsilon^\lambda$ be the $\varepsilon$-cover of a set $\{\lambda \mid \lambda \in [0,C_\lambda]\}$ with the distance metric $\|\cdot\|_2$. Let $\varepsilon_\dagger,\varepsilon_r,\varepsilon_u,\varepsilon_\lambda > 0$ be positive scalars. Consider $\widetilde{Q}^\dagger \in \mathcal{N}_{\varepsilon_\dagger}^{\mathcal{Q}_h^\dagger}$, $\widetilde{Q}^r \in \mathcal{N}_{\varepsilon_r}^{\mathcal{Q}_h^r}$, $\widetilde{Q}^u \in \mathcal{N}_{\varepsilon_u}^{\mathcal{Q}_h^u}$, and $\widetilde{\lambda} \in \mathcal{N}_{\varepsilon_\lambda}^\lambda$. For any $Q^\dagger \in \mathcal{Q}_h^\dagger$, $Q^r \in \mathcal{Q}_h^r$, $Q^u \in \mathcal{Q}_h^u$, and $\lambda \in [0,C_\lambda]$, we have

$$\text{dist}_\infty\left(Q^\dagger + Q^r + \lambda Q^u, \widetilde{Q}^\dagger + \widetilde{Q}^r + \widetilde{\lambda}\widetilde{Q}^u\right)$$

$$\leq \underbrace{\sup_{s,a}\left|Q^\dagger(s,a) - \widetilde{Q}^\dagger(s,a)\right|}_{\leq\varepsilon_\dagger} + \underbrace{\sup_{s,a}\left|Q^r(s,a) - \widetilde{Q}^r(s,a)\right|}_{\leq\varepsilon_r}$$

$$+ \underbrace{\lambda\sup_{s,a}\left|\left(Q^u(s,a) - \widetilde{Q}^u(s,a)\right)\right|}_{\leq C_\lambda\varepsilon_u} + \underbrace{\sup_{s,a}\left|\left(\lambda - \widetilde{\lambda}\right)Q^u(s,a)\right|}_{\varepsilon_\lambda H}$$

$$\overset{(a)}{\leq} \varepsilon_\dagger + \varepsilon_r + C_\lambda\varepsilon_u + \varepsilon_\lambda H,$$

where (a) appropriately chooses $\widetilde{Q}^\dagger,\widetilde{Q}^r,\widetilde{Q}^u,\widetilde{\lambda}$. By replacing $\varepsilon_\dagger$ with $\varepsilon/4$, $\varepsilon_r$ with $\varepsilon/4$, $\varepsilon_u$ with $1/4C_\lambda$, and $\varepsilon_\lambda$ with $\varepsilon/4H$, the above inequality is upper bounded by $\varepsilon$. Thus,

$$\ln\left|\mathcal{N}_\varepsilon^{\mathcal{Q}_h^\circ}\right| \leq \ln\left|\mathcal{N}_{\varepsilon/4H}^\lambda\right| + \ln\left|\mathcal{N}_{\varepsilon/4C_\lambda}^{\mathcal{Q}_h^u}\right| + \ln\left|\mathcal{N}_{\varepsilon/4}^{\mathcal{Q}_h^r}\right| + \ln\left|\mathcal{N}_{\varepsilon/4}^{\mathcal{Q}_h^\dagger}\right|$$

$$\leq \mathcal{O}(d^2)\,\text{polylog}(d,K,H_\kappa,C_r,C_u,B_\dagger,C_\dagger,C_\lambda,\varepsilon^{-1}).$$

where the second inequality uses Lemma 9 and Lemma 29. $\square$

**Definition 12** (Policy class). $\widetilde{\Pi} := \widetilde{\Pi}_1 \times \cdots \times \widetilde{\Pi}_H$ denotes a softmax policy class such that

$$\widetilde{\Pi}_h := \{\pi_Q \in \Pi \mid Q \in \mathcal{Q}_h^\circ\} \quad \text{where} \quad \pi_Q(\cdot \mid s) = \mathrm{SoftMax}\left(\frac{1}{\kappa} Q(s, \cdot)\right) \forall s \in \mathcal{S} ,$$

where $\kappa > 0$. We let $\mathcal{N}_\varepsilon^{\widetilde{\Pi}_h}$ be the $\varepsilon$-cover of $\widetilde{\Pi}_h$ with the distance metric $\mathrm{dist}_1$.

**Lemma 31** ($\pi^{(k),\lambda}$ cover). *When Algorithm 2 is run with $\rho = 1$ and $\kappa > 0$, for all $h$, the following statements hold:*

(i) *For all $(k, h)$ and $\lambda \in [0, C_\lambda]$ in Algorithm 2, $\pi_h^{(k),\lambda} \in \widetilde{\Pi}_h$*

(ii) $\ln\left|\mathcal{N}_\varepsilon^{\widetilde{\Pi}_h}\right| = \mathcal{O}(d^2) \,\mathrm{polylog}(d, K, H_\kappa, C_r, C_u, B_\dagger, C_\dagger, C_\lambda, \varepsilon^{-1}, \kappa^{-1})$

*Proof.* The claim (i) immediately follows from Lemma 30 and Definition 3.

We prove the second claim. For a $Q : \mathcal{S} \times \mathcal{A} \to \mathbb{R}$, let $\pi_Q$ be a softmax policy such that $\pi_Q(\cdot \mid s) = \mathrm{SoftMax}\left(\frac{Q(s,\cdot)}{\kappa}\right)$. Consider $\widetilde{Q}$ from $\mathcal{N}_\varepsilon^{\mathcal{Q}_h^\circ}$. Then, for any $Q \in \mathcal{Q}_h^\circ$, we have

$$\mathrm{dist}_1\left(\pi_Q, \pi_{\widetilde{Q}}\right) \overset{(a)}{\leq} \frac{8}{\kappa} \mathrm{dist}_\infty\left(Q, \widetilde{Q}\right) \overset{(b)}{\leq} \frac{8\varepsilon}{\kappa} ,$$

where (a) uses Lemma 19 and (b) appropriately chooses $\widetilde{Q}$ from $\mathcal{N}_\varepsilon^{\mathcal{Q}_h^\circ}$. Therefore,

$$\ln\left|\mathcal{N}_\varepsilon^{\widetilde{\Pi}_h}\right| \leq \ln\left|\mathcal{N}_{\kappa\varepsilon/8}^{\mathcal{Q}_h^\circ}\right| \leq \mathcal{O}(d^2) \,\mathrm{polylog}(d, K, H_\kappa, C_r, C_u, B_\dagger, C_\dagger, C_\lambda, \varepsilon^{-1}, \kappa^{-1})$$

where the second inequality uses Lemma 30. $\qquad\square$

**Definition 13** (V function class). Let $\mathcal{V}_h^r$, $\mathcal{V}_h^u$, and $\mathcal{V}_h^\dagger$ denote value function classes such that

$$\mathcal{V}_h^r := \left\{V_Q^\pi[\kappa] : \mathcal{S} \to \mathbb{R} \,\middle|\, \pi \in \widetilde{\Pi}_h \cup \{\pi_h^{\mathrm{sf}}\} \text{ and } Q \in \mathcal{Q}_h^r\right\} ,$$

$$\mathcal{V}_h^u := \left\{V_Q^\pi[0] : \mathcal{S} \to \mathbb{R} \,\middle|\, \pi \in \widetilde{\Pi}_h \cup \{\pi_h^{\mathrm{sf}}\} \text{ and } Q \in \mathcal{Q}_h^u\right\} ,$$

$$\text{and } \mathcal{V}_h^\dagger := \left\{V_Q^\pi[0] : \mathcal{S} \to \mathbb{R} \,\middle|\, \pi \in \widetilde{\Pi}_h \cup \{\pi_h^{\mathrm{sf}}\} \text{ and } Q \in \mathcal{Q}_h^\dagger\right\} ,$$

$$\text{where } V_Q^\pi[\kappa](s) := \sum_{a \in \mathcal{A}} \pi(a \mid s)(Q(s, a) - \kappa \ln \pi(a \mid s)) \;\forall s \in \mathcal{S} .$$

We let $\mathcal{N}_\varepsilon^{\mathcal{V}_h^r}$, $\mathcal{N}_\varepsilon^{\mathcal{V}_h^u}$, and $\mathcal{N}_\varepsilon^{\mathcal{V}_h^\dagger}$ be the $\varepsilon$-covers of $\mathcal{V}_h^r$, $\mathcal{V}_h^u$, and $\mathcal{V}_h^\dagger$ with the distance metric $\mathrm{dist}_\infty$.

**Lemma 32** (V covers). *When Algorithm 2 is run with $\rho = 1$ and $\kappa > 0$, for all $h$, the following statements hold:*

(i) *For all $(k, h)$, for any $\lambda \in [0, C_\lambda]$, and for both $\pi = \pi^{(k),\lambda}$ and $\pi = \pi^{\mathrm{sf}}$, we have:*
$\overline{V}_{(k),h}^{\pi,r}[\kappa] \in \mathcal{V}_h^r$, $\underline{V}_{(k),h}^{\pi,u} \in \mathcal{V}_h^u$, *and* $\overline{V}_{(k),h}^{\pi,\dagger} \in \mathcal{V}_h^\dagger$

(ii) $\ln\left|\mathcal{N}_\varepsilon^{\mathcal{V}_h^r}\right| = \mathcal{O}(d^2) \,\mathrm{polylog}(d, K, H_\kappa, C_r, C_u, B_\dagger, C_\dagger, C_\lambda, \varepsilon^{-1}, \kappa^{-1})$,
$\ln\left|\mathcal{N}_\varepsilon^{\mathcal{V}_h^u}\right| = \mathcal{O}(d^2) \,\mathrm{polylog}(d, K, H_\kappa, C_r, C_u, B_\dagger, C_\dagger, C_\lambda, \varepsilon^{-1}, \kappa^{-1})$,
*and* $\ln\left|\mathcal{N}_\varepsilon^{\mathcal{V}_h^\dagger}\right| = \mathcal{O}(d^2) \,\mathrm{polylog}(d, K, H_\kappa, C_r, C_u, B_\dagger, C_\dagger, C_\lambda, \varepsilon^{-1}, \kappa^{-1})$

*Proof.* The condition (i) immediately follow from Lemma 29 and Lemma 31 with Definition 13 and Definition 3.

We prove the second claim (ii). Let $Q \in \mathcal{Q}_h^r$ and $\widetilde{Q} \in \mathcal{N}_{\varepsilon^r}^{\mathcal{Q}_h^r}$ where $\varepsilon^r > 0$. For any two $\pi, \widetilde{\pi} : \mathcal{S} \to \mathscr{P}(\mathcal{A})$, for any $s$, we have

$$\left| \sum_{a \in \mathcal{A}} \pi(a \mid s)(Q(s,a) - \kappa \ln \pi(a \mid s)) - \sum_{a \in \mathcal{A}} \widetilde{\pi}(a \mid s)\left( \widetilde{Q}(s,a) - \kappa \ln \pi(a \mid s) \right) \right|$$

$$\leq \left| \sum_{a \in \mathcal{A}} \pi(a \mid s) Q(s,a) - \sum_{a \in \mathcal{A}} \pi(a \mid s)\widetilde{Q}(s,a) + \sum_{a \in \mathcal{A}} \pi(a \mid s)\widetilde{Q}(s,a) - \sum_{a \in \mathcal{A}} \widetilde{\pi}(a \mid s)\widetilde{Q}(s,a) \right|$$

$$+ \kappa \underbrace{\left| \sum_{a \in \mathcal{A}} \pi(a \mid s) \ln \pi(a \mid s) - \widetilde{\pi}(a \mid s) \ln \widetilde{\pi}(a \mid s) \right|}_{=: \mathscr{H}(\pi) - \mathscr{H}(\widetilde{\pi})}$$

$$\leq \sum_{a \in \mathcal{A}} \pi(a \mid s) \underbrace{\left| Q(s,a) - \widetilde{Q}(s,a) \right|}_{\leq \varepsilon^r} + \|\pi(\cdot \mid s) - \widetilde{\pi}(\cdot \mid s)\|_1 \underbrace{\left\| \widetilde{Q}(\cdot, s) \right\|_\infty}_{\leq H_\kappa} + \kappa(\mathscr{H}(\pi) - \mathscr{H}(\widetilde{\pi}))$$

$$\leq \varepsilon^r + H_\kappa \|\pi(\cdot \mid s) - \widetilde{\pi}(\cdot \mid s)\|_1 + \kappa(\mathscr{H}(\pi) - \mathscr{H}(\widetilde{\pi}))$$

where the second inequality chooses appropriate $\widetilde{Q}$. We defined entropies of $\pi$ and $\widetilde{\pi}$ as $\mathscr{H}(\pi) := \sum_{a \in \mathcal{A}} \pi(a \mid s) \ln \pi(a \mid s)$ and $\mathscr{H}(\widetilde{\pi}) := \sum_{a \in \mathcal{A}} \widetilde{\pi}(a \mid s) \ln \widetilde{\pi}(a \mid s)$, respectively.

The remaining task is to bound $H_\kappa \|\pi(\cdot \mid s) - \widetilde{\pi}(\cdot \mid s)\|_1 + \kappa(\mathscr{H}(\pi) - \mathscr{H}(\widetilde{\pi}))$. When $\pi = \pi^{\mathrm{sf}}$, choosing $\widetilde{\pi} = \pi^{\mathrm{sf}}$ trivially bounds this term by 0. Thus, we only consider the case when $\pi \in \widetilde{\Pi}_h$, i.e., $\pi(\cdot \mid s) = \mathrm{SoftMax}\left(\frac{1}{\kappa}Q^\circ(s, \cdot)\right)$ with $Q^\circ \in \mathcal{Q}_h^\circ$. We also consider $\widetilde{\pi}(\cdot \mid s) = \mathrm{SoftMax}\left(\frac{1}{\kappa}\widetilde{Q}^\circ(s, \cdot)\right)$ with $\widetilde{Q}^\circ \in \mathcal{N}_{\varepsilon^\circ}^{\mathcal{Q}_h^\circ}$, where $\varepsilon^\circ > 0$. For the entropy gap, we have

$$\mathscr{H}(\pi) - \mathscr{H}(\widetilde{\pi})$$

$$= \left| \sum_{a \in \mathcal{A}} \pi(a \mid s) \ln \pi(a \mid s) - \widetilde{\pi}(a \mid s) \ln \widetilde{\pi}(a \mid s) \right|$$

$$= \left| \sum_{a \in \mathcal{A}} (\pi(a \mid s) - \widetilde{\pi}(a \mid s)) \ln \pi(a \mid s) + \sum_{a \in \mathcal{A}} \widetilde{\pi}(a \mid s)(\ln \pi(a \mid s) - \ln \widetilde{\pi}(a \mid s)) \right|$$

$$\leq \|\pi(\cdot \mid s) - \widetilde{\pi}(\cdot \mid s)\|_1 \max_a \ln \pi(a \mid s) + \max_a |\ln \pi(a \mid s) - \ln \widetilde{\pi}(a \mid s)|$$

$$\overset{(a)}{\leq} \underbrace{\|\pi(\cdot \mid s) - \widetilde{\pi}(\cdot \mid s)\|_1}_{\leq \frac{8}{\kappa} \max_a |Q^\circ(s,a) - \widetilde{Q}^\circ(s,a)| \text{ by Lemma 19}} \max_a \ln \pi(a \mid s) + \frac{2}{\kappa} \underbrace{\max_a \left| Q^\circ(s,a) - \widetilde{Q}^\circ(s,a) \right|}_{\leq \varepsilon^\circ}$$

$$\overset{(b)}{\leq} \frac{\varepsilon^\circ}{\kappa}\left( 8 \max_a \ln \pi(a \mid s) + 2 \right),$$

where (a) utilizes a decomposition similar to the proof of Lemma 19, and (b) chooses an appropriate $\widetilde{Q}^\circ$. Finally, $\ln \pi(a \mid s)$ can be bounded as

$$\max_a \ln \pi(a \mid s) = \max_a \frac{1}{\kappa} Q^\circ(s,a) - \ln \sum_{a'} \exp\left( \frac{1}{\kappa}Q^\circ(s,a') \right) \leq \frac{B_\dagger H + H_\kappa + C_\lambda H}{\kappa},$$

where the last inequality is due to Definition 11.

Therefore, we have

$$H_\kappa \|\pi(\cdot \mid s) - \widetilde{\pi}(\cdot \mid s)\|_1 + \kappa(\mathscr{H}(\pi) - \mathscr{H}(\widetilde{\pi})) \leq \varepsilon^\circ \underbrace{\left( 2 + \frac{8}{\kappa}(B_\dagger H + 2H_\kappa + C_\lambda H) \right)}_{=: Z}.$$

Finally, by setting $\varepsilon^r = \varepsilon/2H_\kappa$ and $\varepsilon^\circ = \varepsilon/2Z$, $\ln\left|\mathcal{N}_\varepsilon^{\mathcal{V}_h^r}\right|$ is bounded as:

$$\ln\left|\mathcal{N}_\varepsilon^{\mathcal{V}_h^r}\right| \leq \ln\left(\left|\mathcal{N}_{\varepsilon/2Z}^{\mathcal{Q}_h^\circ}\right| + 1\right) + \ln|\mathcal{N}_{\varepsilon/2H_\kappa}^{\mathcal{Q}_h^r}| = \mathcal{O}(d^2)\,\mathrm{polylog}\big(d, K, H_\kappa, C_r, C_u, B_\dagger, C_\dagger, C_\lambda, \varepsilon^{-1}, \kappa^{-1}\big),$$

where the second inequality is due to Lemma 29 and Lemma 30. The claims for $\ln\left|\mathcal{N}_\varepsilon^{\mathcal{V}_h^u}\right|$ and $\ln\left|\mathcal{N}_\varepsilon^{\mathcal{V}_h^\dagger}\right|$ can be similarly proven. $\qquad\square$

### E.3 Good Events and Value Confidence Bounds for Lemma 4 Proof

**Lemma 33** (Good event 1). *Define $\mathscr{E}_1$ as the event where the following inequality holds:*

$$\sum_{k=1}^{K}\sum_{h=1}^{H}\mathbb{E}\left[\left\|\phi(s_h^{(k)},a_h^{(k)})\right\|^2_{\left(\Lambda_h^{(k)}\right)^{-1}}\;\middle|\;s_h^{(k)},a_h^{(k)}\sim\pi_h^{(k)}\right]$$

$$\leq 2\sum_{k=1}^{K}\sum_{h=1}^{H}\left\|\phi(s_h^{(k)},a_h^{(k)})\right\|^2_{\left(\Lambda_h^{(k)}\right)^{-1}}+4H\ln\frac{2KH}{\delta}\;.$$

*If Algorithm 2 is run with $\rho=1$, $\mathbb{P}(\mathscr{E}_1)\geq 1-\delta$.*

*Proof.* The claim immediately follows from Lemma 11 with $\|\phi\|_2\leq 1$ and $\rho=1$. $\qquad\square$

**Lemma 34** (Good event 2). *Define $\mathscr{E}_2$ as the event where the following condition holds:*
*For all $k,h$ and for any $V^r\in\mathcal{V}_{h+1}^r$, $V^u\in\mathcal{V}_{h+1}^u$, and $V^\dagger\in\mathcal{V}_{h+1}^\dagger$*

$$\left|\left(\left(\widehat{P}_h^{(k)}-P_h\right)V^r\right)(s,a)\right|\leq C_r\beta_h^{(k)}(s,a)\quad\forall(h,s,a)\in[\![1,H]\!]\times\mathcal{S}\times\mathcal{A}$$

$$\left|\left(\left(\widehat{P}_h^{(k)}-P_h\right)V^u\right)(s,a)\right|\leq C_u\beta_h^{(k)}(s,a)\quad\forall(h,s,a)\in[\![1,H]\!]\times\mathcal{S}\times\mathcal{A}$$

$$and\;\left|\left(\left(\widehat{P}_h^{(k)}-P_h\right)V^\dagger\right)(s,a)\right|\leq C_\dagger\beta_h^{(k)}(s,a)\quad\forall(h,s,a)\in[\![1,H]\!]\times\mathcal{S}\times\mathcal{A}\;.$$

*If Algorithm 2 is run with $\rho=1$, $C_r=\widetilde{\mathcal{O}}(dH_\kappa)$, $C_u=\widetilde{\mathcal{O}}(dH)$, and $C_\dagger=\widetilde{\mathcal{O}}(dHB_\dagger)$, we have $\mathbb{P}(\mathscr{E}_2)\geq 1-2\delta$.*

*Proof.* Using Lemma 28 with $\mathcal{N}_{1/K}^{\mathcal{V}_{h+1}^r}$, with probability at least $1-\delta$, for any $(k,h,s,a)$,

$$\left|\left(\left(\widehat{P}_h^{(k)}-P_h\right)V^r\right)(s,a)\right|$$

$$\overset{(a)}{\leq}\|\phi(s,a)\|_{\left(\Lambda_h^{(k)}\right)^{-1}}\left(\sqrt{d}H_\kappa+2H_\kappa\sqrt{\frac{d}{2}\ln(2K)}+2H_\kappa\sqrt{\ln\frac{\left|\mathcal{N}_{1/K}^{\mathcal{V}_{h+1}^r}\right|}{\delta}+4}\right)$$

$$\overset{(b)}{\leq}\|\phi(s,a)\|_{\left(\Lambda_h^{(k)}\right)^{-1}}\widetilde{\mathcal{O}}(dH_\kappa)\ln C_r\overset{(c)}{\leq}\|\phi(s,a)\|_{\left(\Lambda_h^{(k)}\right)^{-1}}C_r$$

where (a) sets $\varepsilon=1/K$ to $\mathcal{N}_\varepsilon^{\mathcal{V}_h^u}$ and uses Lemma 28, (b) uses Lemma 29, and (c) set sufficiently large $C_r=\widetilde{\mathcal{O}}(dH_\kappa)$ and uses Lemma 15. The claim for $\mathcal{V}_{h+1}^u$ and $\mathcal{V}_{h+1}^\dagger$ can be similarly proven. $\qquad\square$

**Lemma 35** (Remove clipping one-side). *Under $\mathscr{E}_2$, for any $(k,h,s,a)$, and for any $\lambda\in[0,C_\lambda]$, for both $\pi=\pi^{(k),\lambda}$ and $\pi=\pi^{\mathrm{sf}}$, we have*

$$C_r\beta_h^{(k)}(s,a)+\left(\widehat{P}_h^{(k)}\overline{V}_{(k),h+1}^{\pi,r}[\kappa]\right)(s,a)\geq\left(P_h\overline{V}_{(k),h+1}^{\pi,r}[\kappa]\right)(s,a)\geq 0\;,$$

$$-C_u\beta_h^{(k)}(s,a)+\left(\widehat{P}_h^{(k)}\underline{V}_{(k),h+1}^{\pi,u}\right)(s,a)\leq\left(P_h\underline{V}_{(k),h+1}^{\pi,u}\right)(s,a)\leq H-h\;,$$

$$and\;\;C_\dagger\beta_h^{(k)}(s,a)+\left(\widehat{P}_h^{(k)}\overline{V}_{(k),h+1}^{\pi,\dagger}\right)(s,a)\geq\left(P_h\overline{V}_{(k),h+1}^{\pi,\dagger}\right)(s,a)\geq 0$$

*Proof.* We have

$$C_r\beta_h^{(k)}(s,a) + \left(\widehat{P}_h^{(k)}\overline{V}_{(k),h+1}^{\pi,r}[\kappa]\right)(s,a)$$

$$\overset{(a)}{\geq}\left|\left(P_h - \widehat{P}_h^{(k)}\right)\overline{V}_{(k),h+1}^{\pi,r}[\kappa]\right|(s,a) + \left(\widehat{P}_h^{(k)}\overline{V}_{(k),h+1}^{\pi,r}[\kappa]\right)(s,a)$$

$$\geq\left(P_h - \widehat{P}_h^{(k)}\right)\overline{V}_{(k),h+1}^{\pi,r}[\kappa](s,a) + \left(\widehat{P}_h^{(k)}\overline{V}_{(k),h+1}^{\pi,r}[\kappa]\right)(s,a)$$

$$=P_h\overline{V}_{(k),h+1}^{\pi,r}[\kappa](s,a) \overset{(b)}{\geq} 0 ,$$

where (a) is due to $\mathscr{E}_2$ with Lemma 32 and (b) is due to $r \geq 0$ and by the definition of $\overline{V}_{(k),h+1}^{\pi,r}[\kappa]$. The claim for $\overline{V}_{(k),h+1}^{\pi,\dagger}$ can be similarly proven.

For $\underline{V}_{(k),h+1}^{\pi,u}$, we have

$$-C_u\beta_h^{(k)}(s,a) + \left(\widehat{P}_h^{(k)}\underline{V}_{(k),h+1}^{\pi,u}\right)(s,a)$$

$$\overset{(a)}{\leq}-\left|\left(\widehat{P}_h^{(k)} - P_h\right)\underline{V}_{(k),h+1}^{\pi,u}\right|(s,a) + \left(\widehat{P}_h^{(k)}\underline{V}_{(k),h+1}^{\pi,u}\right)(s,a)$$

$$\leq-\left(\widehat{P}_h^{(k)} - P_h\right)\underline{V}_{(k),h+1}^{\pi,u}(s,a) + \left(\widehat{P}_h^{(k)}\underline{V}_{(k),h+1}^{\pi,u}\right)(s,a)$$

$$=P_h\underline{V}_{(k),h+1}^{\pi,u}(s,a) \overset{(b)}{\leq} H - h ,$$

where (a) is due to $\mathscr{E}_2$ with Lemma 32 and (b) is due to $u \leq \mathbf{1}$ and by the definition of $\underline{V}_{(k),h+1}^{\pi,u}$. $\quad\square$

**Definition 14** ($Q$ estimation gap). For any $h, k$ and $\pi \in \Pi$, define $\delta_{(k),h}^{\pi,r}, \delta_{(k),h}^{\pi,u}, \delta_{(k),h}^{\pi,\dagger} : \mathcal{S} \times \mathcal{A} \to \mathbb{R}$ be functions such that:

$$\delta_{(k),h}^{\pi,r} = \mathrm{clip}\left\{C_r\beta_h^{(k)} + \left(\widehat{P}_h^{(k)}\overline{V}_{(k),h+1}^{\pi,r}[\kappa]\right), 0, H_\kappa - h_\kappa\right\} - \left(P_h\overline{V}_{(k),h+1}^{\pi,r}[\kappa]\right) ,$$

$$\delta_{(k),h}^{\pi,u} = \left(P_h\underline{V}_{(k),h+1}^{\pi,u}\right) - \mathrm{clip}\left\{-C_u\beta_h^{(k)} + \left(\widehat{P}_h^{(k)}\underline{V}_{(k),h+1}^{\pi,u}\right), 0, H - h\right\} ,$$

$$\text{and } \delta_{(k),h}^{\pi,\dagger} = \mathrm{clip}\left\{C_\dagger\beta_h^{(k)} + \left(\widehat{P}_h^{(k)}\overline{V}_{(k),h+1}^{\pi,\dagger}\right), 0, B_\dagger(H - h)\right\} - \left(P_h\overline{V}_{(k),h+1}^{\pi,\dagger}\right) ,$$

It is clear that these functions satisfy, for any $(\pi, k, h)$,

$$\overline{Q}_{(k),h}^{\pi,r}[\kappa] = Q_{P,h}^{\pi,r+\delta_{(k)}^{\pi,r}}[\kappa], \quad \underline{Q}_{(k),1}^{\pi,u} = Q_{P,h}^{\pi,u-\delta_{(k)}^{\pi,u}}, \quad \text{and} \quad \overline{Q}_{(k),h}^{\pi,\dagger} = Q_{P,h}^{\pi,B_\dagger\beta^{(k)}+\delta_{(k)}^{\pi,\dagger}} . \tag{11}$$

Additionally, let $\Delta_r^{(k)}, \Delta_u^{(k)}$, and $\Delta_\dagger^{(k)}$ be function classes such that:

$$\Delta_r^{(k)} := \left\{\delta : [\![1, H]\!] \times \mathcal{S} \times \mathcal{A} \to \mathbb{R} \;\middle|\; \mathbf{0} \leq \delta_h \leq \min\left\{2C_r\beta_h^{(k)}, H_\kappa - h_\kappa\right\} \forall h \in [\![1, H]\!]\right\}$$

$$\Delta_u^{(k)} := \left\{\delta : [\![1, H]\!] \times \mathcal{S} \times \mathcal{A} \to \mathbb{R} \;\middle|\; \mathbf{0} \leq \delta_h \leq \min\left\{2C_u\beta_h^{(k)}, H - h\right\} \forall h \in [\![1, H]\!]\right\}$$

$$\text{and } \Delta_\dagger^{(k)} := \left\{\delta : [\![1, H]\!] \times \mathcal{S} \times \mathcal{A} \to \mathbb{R} \;\middle|\; \mathbf{0} \leq \delta_h \leq \min\left\{2C_\dagger\beta_h^{(k)}, B_\dagger(H - h)\right\} \forall h \in [\![1, H]\!]\right\} .$$

**Lemma 36.** *Under $\mathscr{E}_2$, for any $k$ and for any $\lambda \in [0, C_\lambda]$, for both $\pi = \pi^{(k),\lambda}$ and $\pi = \pi^{\mathrm{sf}}$, it holds that $\delta_{(k),\cdot}^{\pi,r} \in \Delta_r^{(k)}, \delta_{(k),\cdot}^{\pi,u} \in \Delta_u^{(k)}$, and $\delta_{(k),\cdot}^{\pi,\dagger} \in \Delta_\dagger^{(k)}$.*

*Proof.* $\delta_{(k),h}^{\pi,u}(s,a) \leq H - h$ clearly holds. Additionally, we have

$$\delta_{(k),h}^{\pi,u}(s,a) \overset{(a)}{=} \left(P_h\underline{V}_{(k),h+1}^{\pi,u}\right)(s,a) - \max\left\{-C_u\beta_h^{(k)}(s,a) + \left(\widehat{P}_h^{(k)}\underline{V}_{(k),h+1}^{\pi,u}\right)(s,a), 0\right\}$$

$$\leq\left(P_h\underline{V}_{(k),h+1}^{\pi,u}\right)(s,a) + C_u\beta_h^{(k)}(s,a) - \left(\widehat{P}_h^{(k)}\underline{V}_{(k),h+1}^{\pi,u}\right)(s,a)$$

$$\leq C_u\beta_h^{(k)}(s,a) + \left|\left(P_h - \widehat{P}_h^{(k)}\right)\underline{V}_{(k),h+1}^{\pi,u}\right|(s,a) \overset{(b)}{\leq} 2C_u\beta_h^{(k)}(s,a) ,$$

where (a) is due to Lemma 35 and (b) is due to $\mathcal{E}_2$. Finally, note that

$$
\begin{aligned}
\delta^{\pi,u}_{(k),h}(s,a) &= \Big(P_h \underline{V}^{\pi,u}_{(k),h+1}\Big)(s,a) - \max\Big\{-C_u\beta^{(k)}_h(s,a) + \Big(\widehat{P}^{(k)}_h \underline{V}^{\pi,u}_{(k),h+1}\Big)(s,a), 0\Big\} \\
&\geq \underbrace{C_u\beta^{(k)}_h(s,a) + \Big(P_h \underline{V}^{\pi,u}_{(k),h+1}\Big)(s,a) - \Big(\widehat{P}^{(k)}_h \underline{V}^{\pi,u}_{(k),h+1}\Big)(s,a)}_{\geq 0 \ \text{by} \ \mathcal{E}_2} \geq 0 \ .
\end{aligned}
$$

This concludes the proof for $\delta^{\pi,u}_{(k),h}$. The claims for $\delta^{\pi,r}_{(k),h}$ and $\delta^{\pi,\dagger}_{(k),h}$ can be similarly proven. $\qquad\square$

**Lemma 37** (Restatement of Lemma 4). *Suppose $\mathcal{E}_2$ holds. For any $k$ and for any $\lambda \in [0, C_\lambda]$, for both $\pi = \pi^{(k),\lambda}$ and $\pi = \pi^{\mathrm{sf}}$, we have*

$$
\begin{aligned}
V^{\pi,r}_{P,h} &\leq \overline{V}^{\pi,r}_{(k),h} \leq V^{\pi,r+2C_r\beta^{(k)}}_{P,h}, & Q^{\pi,r}_{P,h} &\leq \overline{Q}^{\pi,r}_{(k),h} \leq Q^{\pi,r+2C_r\beta^{(k)}}_{P,h} \\
V^{\pi,B_\dagger\beta^{(k)}}_{P,h} &\leq \overline{V}^{\pi,\dagger}_{(k),h} \leq V^{\pi,B_\dagger\beta^{(k)}+2C_\dagger\beta^{(k)}}_{P,h}, & Q^{\pi,B_\dagger\beta^{(k)}}_{P,h} &\leq \overline{Q}^{\pi,\dagger}_{(k),h} \leq Q^{\pi,B_\dagger\beta^{(k)}+2C_\dagger\beta^{(k)}}_{P,h}, \\
V^{\pi,u-2C_u\beta^{(k)}}_{P,h} &\leq \underline{V}^{\pi,u}_{(k),h} \leq V^{\pi,u}_{P,h}, & Q^{\pi,u-2C_u\beta^{(k)}}_{P,h} &\leq \underline{Q}^{\pi,u}_{(k),h} \leq Q^{\pi,u}_{P,h} \ .
\end{aligned}
$$

*Proof.* The inequalities for $Q$ functions directly hold by Equation (11) and Lemma 36.

For the utility $V$ function,

$$
\begin{aligned}
\underline{V}^{\pi^{(k)},u}_{(k),h}(s) - V^{\pi^{(k)},u}_{P,h}(s) &= \sum_{a\in\mathcal{A}} \pi_h(a \mid s)\Big(\underline{Q}^{\pi,u}_{(k),h}(s,a) - Q^{\pi,u}_{P,h}(s,a)\Big) \\
&\overset{(a)}{=} \sum_{a\in\mathcal{A}} \pi_h(a \mid s) Q^{\pi,-\delta^{\pi,u}_{(k)}}_{P,h}(s) \overset{(b)}{\leq} 0 \ ,
\end{aligned}
$$

where (a) uses Equation (11) and (b) uses Lemma 36. Similarly,

$$
\begin{aligned}
\underline{V}^{\pi,u}_{(k),h}(s) - V^{\pi,u-2C_u\beta^{(k)}}_{P,h}(s) &= \sum_{a\in\mathcal{A}} \pi_h(a \mid s)\Big(\underline{Q}^{\pi,u}_{(k),h}(s,a) - Q^{\pi,u-2C_u\beta^{(k)}}_{P,h}(s,a)\Big) \\
&\overset{(a)}{=} \sum_{a\in\mathcal{A}} \pi_h(a \mid s) Q^{\pi,-\delta^{\pi,u}_{(k)}+2C_u\beta^{(k)}}_{P,h}(s) \overset{(b)}{\geq} 0 \ ,
\end{aligned}
$$

where (a) uses Equation (11) and (b) uses Lemma 36. The claims for $r$ and $\dagger$ can be similarly proven. $\qquad\square$

### E.4 Proofs for Zero-Violation Guarantee (Section 3.2.1)

#### E.4.1 Proof of Lemma 5 and Lemma 6

**Lemma 38** (Restatement of Lemma 6). *Let $f, g : \llbracket 1, H \rrbracket \times \mathcal{S} \times \mathcal{A} \to \mathbb{R}$ be functions and let $\kappa > 0$. Given $\lambda \geq 0$, let $\pi^\lambda$ be a softmax policy such that*

$$
\pi^\lambda_h(\cdot \mid s) = \mathrm{SoftMax}\left(\frac{1}{\kappa}\Big(Q^{\pi,f}_{P,h}[\kappa](s,\cdot) + \lambda Q^{\pi,g}_{P,h}(s,\cdot)\Big)\right) \ .
$$

*Then, $V^{\pi^\lambda,g}_{P,1}(s_1)$ is monotonically increasing in $\lambda$.*

*Proof.* Let $\mathcal{W} := \big\{w^\pi_{P,\cdot} : \llbracket 1, H \rrbracket \times \mathcal{S} \times \mathcal{A} \to [0,1] \,\big|\, \pi \in \Pi\big\}$ be the set of all the occupancy measures. Let $\mathscr{L} : \mathbb{R} \times \mathcal{W} \to \mathbb{R}$ be a function such that:

$$
\mathscr{L}(\lambda, w) = \sum_{h,s,a\in\llbracket 1,H\rrbracket\times\mathcal{S}\times\mathcal{A}} w_h(s,a)(f_h(s,a) + \lambda g_h(s,a)) - \kappa w_h(s,a)\ln\frac{w_h(s,a)}{\sum_{a'\in\mathcal{A}} w_h(s,a')} \ .
$$

We first show that $\mathscr{L}$ is strictly concave in $\mathcal{W}$. Let

$$
\mathscr{H} : w \in \mathcal{W} \mapsto \sum_{h,s,a\in\llbracket 1,H\rrbracket\times\mathcal{S}\times\mathcal{A}} -w_h(s,a)\ln\frac{w_h(s,a)}{\sum_{a'\in\mathcal{A}} w_h(s,a')}
$$

be the function representing the second term of $\mathscr{L}$. Then, [10]

$$\mathscr{H}\left(\alpha w^1 + (1-\alpha)w^2\right)$$

$$= -\sum_{h,s,a}\left(\alpha w_h^1(s,a) + (1-\alpha)w_h^2(s,a)\right)\log\frac{\alpha w_h^1(s,a) + (1-\alpha)w_h^2(s,a)}{\alpha\sum_{a'} w_h^1(s,a') + (1-\alpha)\sum_{a'} w_h^2(s,a')}$$

$$\overset{(a)}{\geq} -\sum_{h,s,a}\alpha w_h^1(s,a)\log\frac{\alpha w_h^1(s,a)}{\alpha\sum_{a'} w_h^1(s,a')} - \sum_{h,s,a}(1-\alpha)w_h^2(s,a)\log\frac{(1-\alpha)w_h^2(s,a)}{(1-\alpha)\sum_{a'} w_h^2(s,a')}$$

$$= \alpha\mathscr{H}\left(w_h^1\right) + (1-\alpha)\mathscr{H}\left(w_h^2\right) ,$$

for any $w^1, w^2 \in \mathcal{W}$ and $\alpha \in [0,1]$, where (a) is due to the log sum inequality $(\sum_i \mathbf{x}_i)\ln\frac{\sum_i \mathbf{x}_i}{\sum_i \mathbf{y}_i} \leq \sum_i \mathbf{x}_i\ln\frac{\mathbf{x}_i}{\mathbf{y}_i}$ for non-negative $\mathbf{x}_i$ and $\mathbf{y}_i$. Since (a) takes equality if and only if $w^1 = w^2$, $\mathscr{H}$ is strictly concave. Consequently, $\mathscr{L}(\lambda, w) = \sum_{h,s,a\in[\![1,H]\!]\times\mathcal{S}\times\mathcal{A}} w_h(s,a)(f_h(s,a) + \lambda g_h(s,a)) - \kappa\mathscr{H}(w)$ is also strictly concave in $\mathcal{W}$.

Let $w^\lambda = \arg\max_{w\in\mathcal{W}}\mathscr{L}(\lambda, w)$, which is a unique maximizer due to the strict concavity. Define $\mathscr{L}(\lambda) := \max_{w\in\mathcal{W}}\mathscr{L}(\lambda, w)$. Using Danskin's theorem (Lemma 10), $\mathscr{L}(\lambda)$ is convex and $\frac{\partial\mathscr{L}(\lambda)}{\partial\lambda} = \sum_{h,s,a\in[\![1,H]\!]\times\mathcal{S}\times\mathcal{A}} w_h^\lambda(s,a)g_h(s,a)$. Since $\mathscr{L}(\lambda)$ is convex, its derivative is non-decreasing. Therefore,

$$\frac{\partial^2\mathscr{L}(\lambda)}{\partial\lambda^2} = \frac{\partial}{\partial\lambda}\sum_{h,s,a\in[\![1,H]\!]\times\mathcal{S}\times\mathcal{A}} w_h^\lambda(s,a)g_h(s,a) \geq 0 . \tag{12}$$

Since $\pi^\lambda$ is the softmax policy, combined with the one-to-one mapping between occupancy measure and policy [35], the well-known analytical solution of regularized MDP [15] indicates that $w^\lambda$ corresponds to the occupancy measure of $\pi^\lambda$. Thus, due to Equation (12), it holds that

$$0 \leq \frac{\partial}{\partial\lambda}\sum_{h,s,a\in[\![1,H]\!]\times\mathcal{S}\times\mathcal{A}} w_h^\lambda(s,a)g_h(s,a) = \frac{\partial}{\partial\lambda}V_{P,1}^{\pi^\lambda,g}(s_1) .$$

This concludes the proof. $\qquad\square$

**Definition 15** (Softmax policy with fixed $\boldsymbol{\delta}$). For any $k \in \mathcal{U}^{\complement}$, $\boldsymbol{\delta} := (\delta^r, \delta^u, \delta^\dagger) \in \Delta_r^{(k)}\times\Delta_u^{(k)}\times\Delta_\dagger^{(k)}$ and $\lambda \geq 0$, let $\pi^{\boldsymbol{\delta},\lambda} \in \Pi$ be a policy such that

$$\pi_h^{\boldsymbol{\delta},\lambda}(\cdot\mid s) = \mathrm{SoftMax}\left(\frac{1}{\kappa}\left(Q_{P,h}^{\pi^{\boldsymbol{\delta},\lambda},B_\dagger\beta^{(k)}+\delta^\dagger}(s,\cdot) + Q_{P,h}^{\pi^{\boldsymbol{\delta},\lambda},r+\delta^r}[\kappa](s,\cdot) + \lambda Q_{P,h}^{\pi^{\boldsymbol{\delta},\lambda},u-\delta^u}(s,\cdot)\right)\right) .$$

**Lemma 39** (Existence of feasible $\lambda$). *Suppose $\kappa \leq \frac{\xi^2}{32H_\kappa^2(B_\dagger+1)}$. For any $k$ and for any $\boldsymbol{\delta} \in \Delta_\dagger^{(k)}\times\Delta_r^{(k)}\times\Delta_u^{(k)}$, there exists a $\lambda^{\boldsymbol{\delta}} \in \left[0, \frac{8H_\kappa^2(B_\dagger+1)}{\xi}\right]$ such that, $V_{P,1}^{\pi^{\boldsymbol{\delta},\lambda},u-\delta^u}(s_1) \geq b$ holds for any $\lambda \geq \lambda^{\boldsymbol{\delta}}$.*

*Proof.* Throughout the proof, we use a shorthand $r^{\boldsymbol{\delta}} := B_\dagger\beta^{(k)} + \delta^\dagger + r + \delta^r$. Consider the following entropy-regularized max-min optimization problem:

$$\max_{\pi\in\Pi}\min_{\lambda\geq 0} V_{P,1}^{\pi,r^{\boldsymbol{\delta}}}[\kappa](s_1) + \lambda\left(V_{P,1}^{\pi,u-\delta^u}(s_1) - b - \frac{\xi}{4}\right) + \frac{\kappa}{2}\lambda^2$$

$$= \min_{\lambda\geq 0}\max_{\pi\in\Pi} V_{P,1}^{\pi,r^{\boldsymbol{\delta}}}[\kappa](s_1) + \lambda\left(V_{P,1}^{\pi,u-\delta^u}(s_1) - b - \frac{\xi}{4}\right) + \frac{\kappa}{2}\lambda^2 . \tag{13}$$

where the equality holds by the strong duality of regularized CMDPs (see, e.g., **Appendix C.1** in Ding et al. [11]). Let $(\widetilde{\pi}, \widetilde{\lambda})$ be a saddle point of the problem, which is ensured to be unique thanks to the regularization. We first show the analytical forms of $(\widetilde{\pi}, \widetilde{\lambda})$.

---

[10]This proof is based of **Lemma 14** from Ding et al. [11]

**Analytical forms of** $(\widetilde{\pi}, \widetilde{\lambda})$**.** Due to the strong duality, we have

$$\max_{\pi \in \Pi} V_{P,1}^{\pi, r^\delta \widetilde{\lambda}(u-\delta^u)}[\kappa](s_1) = V_{P,1}^{\widetilde{\pi}, r^\delta + \widetilde{\lambda}(u-\delta^u)}[\kappa](s_1) \ .$$

Since the left-hand side is an entropy-regularized optimization problem in an MDP, the well-known analytical solution of regularized MDP indicates that [15]:

$$\widetilde{\pi}_h(\cdot \mid s) = \mathrm{SoftMax}\left( \frac{1}{\kappa} \left( Q_{P,h}^{\widetilde{\pi}, r^\delta}[\kappa](s, \cdot) + \widetilde{\lambda} Q_{P,h}^{\widetilde{\pi}, u-\delta^u}(s, \cdot) \right) \right) = \pi_h^{\boldsymbol{\delta}, \widetilde{\lambda}} \ , \tag{14}$$

where the last equality is due to the definition of $\pi_h^{\boldsymbol{\delta}, \widetilde{\lambda}}$. Additionally, due to the strong duality,

$$\widetilde{\lambda} \in \arg\min_{\lambda \geq 0} V_{P,1}^{\widetilde{\pi}, r^\delta}[\kappa](s_1) + \lambda \left( V_{P,1}^{\widetilde{\pi}, u-\delta^u}(s_1) - b - \frac{\xi}{4} \right) + \frac{\kappa}{2}\lambda^2 \ .$$

Since the right-hand side is a quadratic equation on $\lambda$, we have

$$\widetilde{\lambda} = \frac{1}{\kappa}\left[ b + \frac{\xi}{4} - V_{P,1}^{\widetilde{\pi}, u-\delta^u}(s_1) \right]_+ \ . \tag{15}$$

$\widetilde{\lambda}$ **upper bound.** Next, we will show that $\widetilde{\lambda}$ is upper bounded by constant. We have

$$2H_\kappa^2(B_\dagger + 1) \overset{(a)}{\geq} V_{P,1}^{\widetilde{\pi}, r^\delta}[\kappa](s_1) - \underbrace{\frac{1}{2\kappa}\left[ b + \frac{\xi}{4} - V_{P,1}^{\widetilde{\pi}, u-\delta^u}(s_1) \right]_+^2}_{\geq 0}$$

$$\overset{(b)}{=} V_{P,1}^{\widetilde{\pi}, r^\delta}[\kappa](s_1) + \widetilde{\lambda}\left( V_{P,1}^{\widetilde{\pi}, u-\delta^u}(s_1) - b - \frac{\xi}{4} \right) + \frac{\kappa}{2}\widetilde{\lambda}^2$$

$$\overset{(c)}{\geq} V_{P,1}^{\pi^{\mathrm{sf}}, r^\delta}[\kappa](s_1) + \widetilde{\lambda}\left( V_{P,1}^{\pi^{\mathrm{sf}}, u-\delta^u}(s_1) - b - \frac{\xi}{4} \right) + \frac{\kappa}{2}\widetilde{\lambda}^2$$

$$\geq \widetilde{\lambda}\left( \underbrace{V_{P,1}^{\pi^{\mathrm{sf}}, u}(s_1) - b - \frac{\xi}{4}}_{\geq 3\xi/4} - \underbrace{V_{P,1}^{\pi^{\mathrm{sf}}, 2C_\beta \beta^{(k)}}(s_1)}_{\leq \xi/2 \text{ since } k \in \mathcal{U}^{\complement}} \right) \geq \widetilde{\lambda}\frac{\xi}{4} \ ,$$

where (a) is since $\|r^\delta\|_\infty = \|B_\dagger \beta^{(k)} + \delta^\dagger + r + \delta^r\|_\infty \leq B_\dagger + B_\dagger H + 1 + H = (H+1)(B_\dagger + 1)$, (b) is due to Equation (15), (c) uses Equation (13). By reformulating the inequality,

$$\widetilde{\lambda} \leq \frac{8H_\kappa^2(B_\dagger + 1)}{\xi} \ . \tag{16}$$

**Constraint violation of** $\pi^{\boldsymbol{\delta}, \lambda}$ Finally, we will show that for any $\lambda \geq \widetilde{\lambda}$, $\pi^{\boldsymbol{\delta}, \lambda}$ guarantees zero constraint violation. Due to Equations (14), (15), and (16), we have

$$\kappa\widetilde{\lambda} = \left[ b + \frac{\xi}{4} - V_{P,1}^{\pi^{\boldsymbol{\delta}, \widetilde{\lambda}}, u-\delta^u}(s_1) \right]_+ \leq \frac{8\kappa H_\kappa^2(B_\dagger + 1)}{\xi} \ ,$$

which ensures the small violation of $\pi^{\boldsymbol{\delta}, \widetilde{\lambda}}$ when $\kappa \ll 1$. Since $V_{P,1}^{\pi^{\boldsymbol{\delta}, \lambda}, u-\delta^u}(s_1)$ is monotonically increasing in $\lambda$ due to Lemma 38, for any $\lambda \geq \widetilde{\lambda}$, $V_{P,1}^{\pi^{\boldsymbol{\delta}, \lambda}, u-\delta^u}(s_1) \geq b + \frac{\xi}{4} - \frac{8\kappa H_\kappa^2(B_\dagger + 1)}{\xi}$. Therefore, by setting $\kappa \leq \frac{\xi^2}{32H_\kappa^2(B_\dagger + 1)}$, we have $V_{P,1}^{\pi^{\boldsymbol{\delta}, \lambda}, u-\delta^u}(s_1) \geq b$. $\qquad\square$

**Lemma 40** (Restatement of Lemma 5). *If Algorithm 2 is run with $\rho = 1$, $C_\lambda \geq \frac{8H_\kappa^2(B_\dagger + 1)}{\xi}$, and $\kappa \leq \frac{\xi^2}{32H_\kappa^2(B_\dagger + 1)}$, under $\mathscr{E}_2$, it holds $\underline{V}_{(k),1}^{\pi^{(k), C_\lambda}, u}(s_1) \geq b$ for any $k \in \mathcal{U}^{\complement}$.*

*Proof.* Due to $\mathscr{E}_2$, it holds that

$$\boldsymbol{\delta} := \left( \delta_{(k), \cdot}^{\pi^{(k), C_\lambda}, r}, \delta_{(k), \cdot}^{\pi^{(k), C_\lambda}, u}, \delta_{(k), \cdot}^{\pi^{(k), C_\lambda}, \dagger} \right) \in \Delta_\dagger^{(k)} \times \Delta_r^{(k)} \times \Delta_u^{(k)} \ .$$

According to Equation (11), this $\boldsymbol{\delta}$ satisfies $\pi^{\boldsymbol{\delta}, C_\lambda} = \pi^{(k), C_\lambda}$ where $\pi^{\boldsymbol{\delta}, C_\lambda}$ is defined in Definition 15. Therefore, using Lemma 39, $\underline{V}_{(k),1}^{\pi^{(k), C_\lambda}, u}(s_1) \geq b$. This concludes the proof. $\qquad\square$

### E.4.2 Proof of Theorem 3

**Lemma 41** (Bonus summation bound). *If Algorithm 2 is run with $\rho = 1$, under $\mathscr{E}_1$ and $\mathscr{E}_2$, it holds that*

$$\sum_{k=1}^{K}\left(V_{P,1}^{\pi^{(k)},\beta^{(k)}}(s_1)\right)^2 \le 2H^2 d\ln\left(1 + \frac{K}{d}\right) + 4H^2\ln\frac{2KH}{\delta} = \widetilde{\mathcal{O}}\left(H^2 d\right)$$

$$and \quad \sum_{k=1}^{K}\left(V_{P,1}^{\pi^{(k)},\beta^{(k)}}(s_1)\right) \le H\sqrt{K}\sqrt{2d\ln\left(1 + \frac{K}{d}\right) + 4\ln\frac{2KH}{\delta}} = \widetilde{\mathcal{O}}\left(H\sqrt{dK}\right).$$

*Proof.* We have

$$
\begin{aligned}
\sum_{k=1}^{K}\left(V_{P,1}^{\pi^{(k)},\beta^{(k)}}(s_1)\right)^2 &= \sum_{k=1}^{K}\left(\sum_{h=1}^{H}\mathbb{E}\left[\beta_h^{(k)}(s_h,a_h)\;\middle|\;s_h,a_h\sim\pi^{(k)}\right]\right)^2 \\
&\overset{(a)}{\le} H\sum_{k=1}^{K}\sum_{h=1}^{H}\left(\mathbb{E}\left[\beta_h^{(k)}(s_h,a_h)\;\middle|\;s_h,a_h\sim\pi^{(k)}\right]\right)^2 \\
&\overset{(a)}{\le} H\sum_{k=1}^{K}\sum_{h=1}^{H}\mathbb{E}\left[\|\phi(s_h,a_h)\|^2_{\left(\mathbf{\Lambda}_h^{(k)}\right)^{-1}}\;\middle|\;s_h,a_h\sim\pi^{(k)}\right] \\
&\overset{(b)}{\le} 2H\sum_{k=1}^{K}\sum_{h=1}^{H}\left\|\phi(s_h^{(k)},a_h^{(k)})\right\|^2_{\left(\mathbf{\Lambda}_h^{(k)}\right)^{-1}} + 4H^2\ln\frac{2KH}{\delta} \\
&\overset{(c)}{\le} 2H^2 d\ln\left(1 + \frac{K}{d}\right) + 4H^2\ln\frac{2KH}{\delta},
\end{aligned}
$$

where (a) is due to Jensen's inequality, (b) is due to $\mathscr{E}_1$, and (c) uses Lemma 12. The second claim follows by:

$$\sum_{k=1}^{K}V_{P,1}^{\pi^{(k)},\beta^{(k)}}(s_1) \overset{(a)}{\le} \sqrt{K}\sqrt{\sum_{k=1}^{K}\left(V_{P,1}^{\pi^{(k)},\beta^{(k)}}(s_1)\right)^2} \overset{(b)}{\le} H\sqrt{K}\sqrt{2d\ln\left(1 + \frac{K}{d}\right) + 4\ln\frac{2KH}{\delta}},$$

where (a) uses Cauchy–Schwarz inequality and (b) uses the first claim. □

**Lemma 42** (Restatement of Theorem 3). *Suppose Algorithm 2 is run with $\rho = 1$ and $\mathscr{E}_1$ and $\mathscr{E}_2$ hold. Then,*

$$|\mathcal{U}| \le \frac{64C_u^2 H^2 d}{\xi^2}\ln\left(\frac{2KH}{\delta}\right) = \widetilde{\mathcal{O}}\left(\xi^{-2}H^4 d^3\right),$$

*where the last equality sets $C_u = \widetilde{\mathcal{O}}(dH)$.*

*Proof.* Using Lemma 41 and Definition 4, we have

$$|\mathcal{U}|\left(\frac{\xi}{2}\right)^2 \le \sum_{k\in\mathcal{U}}\left(V_{P,1}^{\pi^{\text{sf}},2C_u\beta^{(k)}}(s_1)\right)^2 \le 8C_u^2 H^2 d\ln\left(1 + \frac{K}{d}\right) + 16C_u^2 H^2\ln\frac{2KH}{\delta}.$$

Therefore, we have

$$|\mathcal{U}| \le \frac{32C_u^2 H^2 d}{\xi^2}\ln\left(1 + \frac{K}{d}\right) + \frac{64C_u^2 H^2}{\xi^2}\ln\frac{2KH}{\delta} \le \frac{64C_u^2 H^2 d}{\xi^2}\ln\left(\frac{2KH}{\delta}\right).$$

□

## E.5 Proofs for Sublinear Regret Guarantee (Section 3.2.2)

Suppose the good events $\mathscr{E}_1 \cap \mathscr{E}_2$ hold. We decompose the regret as follows:

$$
\begin{aligned}
&\mathrm{Regret}(K) \\
=&\sum_{k=1}^{K}\left(V_{P,1}^{\pi^\star,r}(s_1) - V_{P,1}^{\pi^{(k)},r}(s_1)\right) \\
=&\sum_{k\in\mathcal{U}}\left(V_{P,1}^{\pi^\star,r}(s_1) - V_{P,1}^{\pi^{(k)},r}(s_1)\right) + \sum_{k\in\mathcal{U}^{\complement}}\left(V_{P,1}^{\pi^\star,r}(s_1) - V_{P,1}^{\pi^{(k)},r}(s_1)\right) \\
\leq&|\mathcal{U}|H + \sum_{k\in\mathcal{U}^{\complement}}\left(\overline{V}_{(k),1}^{\pi^{(k)},r}[\kappa](s_1) - V_{P,1}^{\pi^{(k)},r}(s_1)\right) + \sum_{k\in\mathcal{U}^{\complement}}\left(V_{P,1}^{\pi^\star,r}(s_1) - \overline{V}_{(k),1}^{\pi^{(k)},r}[\kappa](s_1)\right) \\
\overset{(a)}{\leq}&\widetilde{\mathcal{O}}\left(d^3 H^4 \xi^{-2}\right) + \sum_{k\in\mathcal{U}^{\complement}}\left(\overline{V}_{(k),1}^{\pi^{(k)},r}[\kappa](s_1) - V_{P,1}^{\pi^{(k)},r}[\kappa](s_1)\right) + \sum_{k\in\mathcal{U}^{\complement}}\left(V_{P,1}^{\pi^\star,r}(s_1) - \overline{V}_{(k),1}^{\pi^{(k)},r}[\kappa](s_1)\right) + \kappa K H \ln A \\
\overset{(b)}{\leq}&\widetilde{\mathcal{O}}\left(d^3 H^4 \xi^{-2}\right) + 2C_r\underbrace{\sum_{k\in\mathcal{U}^{\complement}} V_{P,1}^{\pi^{(k)},\beta^{(k)}}(s_1)}_{①} + \underbrace{\sum_{k\in\mathcal{U}^{\complement}}\left(V_{P,1}^{\pi^\star,r}(s_1) - \overline{V}_{(k),1}^{\pi^{(k)},r}[\kappa](s_1)\right)}_{②} + \kappa K H \ln A \,,
\end{aligned}
$$

$$(17)$$

where (a) uses Lemma 42 and (b) is due to Lemma 37 with $\mathscr{E}_2$. Under $\mathscr{E}_1 \cap \mathscr{E}_2$, ① can be easily bounded by Lemma 41

$$
① \leq C_r\widetilde{\mathcal{O}}(H\sqrt{dK}) \leq \widetilde{\mathcal{O}}\left(H_\kappa^2 d^{3/2}\sqrt{K}\right)\,, \tag{18}
$$

where the last equality inserts $C_r = \widetilde{\mathcal{O}}(dH_\kappa)$.

### E.5.1 Mixture Policy Decomposition

We upper bound ② in Equation (17) by the mixture policy technique.

**Lemma 43** (Mixture policy's feasibility)**.** *Let $\alpha^{(k)} := \frac{\xi}{\xi + 2V_{P,1}^{\pi^\star,2C_u\beta^{(k)}}(s_1)}$. For any $k \in \mathcal{U}^{\complement}$ and $\alpha \in [0, \alpha^{(k)}]$, $\pi^\alpha$ defined in Definition 5 satisfies $V_{P,1}^{\pi^\alpha,u-2C_u\beta^{(k)}}(s_1) \geq b$.*

*Proof.* We have

$$
\begin{aligned}
&V_{P,1}^{\pi^\alpha,u-2C_u\beta^{(k)}}(s_1) - b \\
=&(1-\alpha)\left(V_{P,1}^{\pi^{\mathrm{sf}},u-2C_u\beta^{(k)}}(s_1) - b\right) + \alpha\left(V_{P,1}^{\pi^\star,u-2C_u\beta^{(k)}}(s_1) - b\right) \\
\geq&(1-\alpha)\frac{\xi}{2} + \alpha\left(V_{P,1}^{\pi^\star,-2C_u\beta^{(k)}}(s_1)\right)\,,
\end{aligned}
$$

where the last inequality holds because $V_{P,1}^{\pi^{\mathrm{sf}},2C_u\beta^{(k)}}(s_1) \leq \frac{\xi}{2}$ due to $k \in \mathcal{U}^{\complement}$. Thus, $V_{P,1}^{\pi^\alpha,u-2C_u\beta^{(k)}}(s_1) - b \geq 0$ holds when

$$
\alpha \leq \frac{\xi}{\xi + 2V_{P,1}^{\pi^\star,2C_u\beta^{(k)}}(s_1)}\,.
$$

$\square$

**Lemma 44** (Mixture policy's optimism)**.** *Let $B_\dagger \geq \frac{4C_u H}{\xi}$. For any $k \in \mathcal{U}^{\complement}$, $\pi^{\alpha^{(k)}}$ with $\alpha^{(k)}$ from Lemma 43 satisfies,*

$$
V_{P,1}^{\pi^{\alpha^{(k)}},r+B_\dagger\beta^{(k)}}(s_1) \geq V_{P,1}^{\pi^\star,r}(s_1) \text{ and } V_{P,1}^{\pi^{\alpha^{(k)}},u-2C_u\beta^{(k)}}(s_1) \geq b\,.
$$

*Proof.* The sufficient condition that $V_{P,1}^{\pi^\alpha,r+B_\dagger\beta^{(k)}}(s_1) \geq V_{P,1}^{\pi^\star,r}(s_1)$ to hold is

$$B_\dagger \geq \frac{V_{P,1}^{\pi^\star,r}(s_1) - V_{P,1}^{\pi^\alpha,r}(s_1)}{V_{P,1}^{\pi^\alpha,\beta^{(k)}}(s_1)} = \frac{(1-\alpha)\left(V_{P,1}^{\pi^\star,r}(s_1) - V_{P,1}^{\pi^{\mathrm{sf}},r}(s_1)\right)}{(1-\alpha)V_{P,1}^{\pi^{\mathrm{sf}},\beta^{(k)}}(s_1) + \alpha V_{P,1}^{\pi^\star,\beta^{(k)}}(s_1)}$$

$$= \frac{V_{P,1}^{\pi^\star,r}(s_1) - V_{P,1}^{\pi^{\mathrm{sf}},r}(s_1)}{V_{P,1}^{\pi^{\mathrm{sf}},\beta^{(k)}}(s_1) + \frac{\alpha}{1-\alpha}V_{P,1}^{\pi^\star,\beta^{(k)}}(s_1)}\,.$$

By inserting $\alpha^{(k)} = \frac{\xi}{\xi + 2V_{P,1}^{\pi^\star,2C_u\beta^{(k)}}(s_1)}$ into $\alpha$, i.e., $\frac{\alpha}{1-\alpha} = \frac{\xi}{2V_{P,1}^{\pi^\star,2C_u\beta^{(k)}}(s_1)}$,

$$B_\dagger \geq \frac{V_{P,1}^{\pi^\star,r}(s_1) - V_{P,1}^{\pi^{\mathrm{sf}},r}(s_1)}{V_{P,1}^{\pi^{\mathrm{sf}},\beta^{(k)}}(s_1) + \frac{\xi}{4C_u V_{P,1}^{\pi^\star,\beta^{(k)}}(s_1)}V_{P,1}^{\pi^\star,\beta^{(k)}}(s_1)} = \frac{4C_u\left(V_{P,1}^{\pi^\star,r}(s_1) - V_{P,1}^{\pi^{\mathrm{sf}},r}(s_1)\right)}{2V_{P,1}^{\pi^{\mathrm{sf}},2C_u\beta^{(k)}}(s_1) + \xi}\,.$$

Thus, when $B_\dagger \geq \frac{4C_u H}{\xi}$, it holds that $V_{P,1}^{\pi^{\alpha^{(k)}},r+B_\dagger\beta^{(k)}}(s_1) \geq V_{P,1}^{\pi^\star,r}(s_1)$. The second claim follows from Lemma 43. $\qquad\square$

We are now ready to decompose ②. Using Lemmas 43 and 44, we have

$$② = \sum_{k\in\mathcal{U}^\complement}\left(V_{P,1}^{\pi^\star,r}(s_1) - \overline{V}_{(k),1}^{\pi^{(k)},r}[\kappa](s_1)\right)$$

$$\leq \sum_{k\in\mathcal{U}^\complement}\left(V_{P,1}^{\pi^{\alpha^{(k)}},B_\dagger\beta^{(k)}}(s_1) + V_{P,1}^{\pi^{\alpha^{(k)}},r}[\kappa](s_1) - \overline{V}_{(k),1}^{\pi^{(k)},r}[\kappa](s_1)\right)$$

$$= \sum_{k\in\mathcal{U}^\complement}\Big(V_{P,1}^{\pi^{\alpha^{(k)}},B_\dagger\beta^{(k)}}(s_1) + V_{P,1}^{\pi^{\alpha^{(k)}},r}[\kappa](s_1) + \overline{\lambda}^{(k,T)}V_{P,1}^{\pi^{\alpha^{(k)}},u-2C_u\beta^{(k)}}(s_1)$$

$$\underbrace{-\overline{V}_{(k),1}^{\pi^{(k)},\dagger}(s_1) - \overline{V}_{(k),1}^{\pi^{(k)},r}[\kappa](s_1) - \overline{\lambda}^{(k,T)}\underline{V}_{(k),1}^{\pi^{(k)},u}(s_1)\Big)}_{③} \tag{19}$$

$$+ \underbrace{\sum_{k\in\mathcal{U}^\complement}\overline{V}_{(k),1}^{\pi^{(k)},\dagger}(s_1)}_{④} + \underbrace{\sum_{k\in\mathcal{U}^\complement}\overline{\lambda}^{(k,T)}\left(\underline{V}_{(k),1}^{\pi^{(k)},u}(s_1) - V_{P,1}^{\pi^{\alpha^{(k)}},u-2C_u\beta^{(k)}}(s_1)\right)}_{⑤}\,,$$

where $\overline{\lambda}^{(k,T)}$ is defined in Line 10. Using Lemma 37, the term ④ is bounded as

$$④ \leq V_{P,1}^{\pi^{(k)},(B_\dagger+2C_\dagger)\beta^{(k)}}(s_1)\,.$$

Using Lemma 41, it holds that

$$④ \leq (B_\dagger + 2C_\dagger)\widetilde{\mathcal{O}}\left(H\sqrt{dK}\right) = \widetilde{\mathcal{O}}\left(H^4 d^{5/2}\xi^{-1}\sqrt{K}\right)\,, \tag{20}$$

where the last equality inserts $B_\dagger = 4\xi^{-1}C_u H$, $C_u = \widetilde{\mathcal{O}}(dH)$, and $C_\dagger = \widetilde{\mathcal{O}}(dH B_\dagger)$. We will bound ③ and ⑤ separately.

### E.5.2 Optimistic Bounds

**Lemma 45** (Optimism in composite value function). *Suppose $\mathcal{E}_2$ holds. Then,*

$$③ = \sum_{k\in\mathcal{U}^\complement}\Big(V_{P,1}^{\pi^{\alpha^{(k)}},B_\dagger\beta^{(k)}}(s_1) + V_{P,1}^{\pi^{\alpha^{(k)}},r}[\kappa](s_1) + \overline{\lambda}^{(k,T)}V_{P,1}^{\pi^{\alpha^{(k)}},u-2C_u\beta^{(k)}}(s_1)$$

$$- \overline{V}_{(k),1}^{\pi^{(k)},B_\dagger\beta^{(k)}}(s_1) - \overline{V}_{(k),1}^{\pi^{(k)},r}[\kappa](s_1) - \overline{\lambda}^{(k,T)}\underline{V}_{(k),1}^{\pi^{(k)},u}(s_1)\Big) \leq 0\,.$$

*Proof.* Using Lemma 18, for any $k \in \mathcal{U}^{\complement}$, we have

$$\overline{V}_{(k),1}^{\pi^{(k)},B_{\dagger}\beta^{(k)}}(s_1) + \overline{V}_{(k),1}^{\pi^{(k)},r}[\kappa](s_1) + \overline{\lambda}^{(k,T)}\underline{V}_{(k),1}^{\pi^{(k)},u}(s_1)$$

$$- V_{P,1}^{\pi^{\alpha(k)},B_{\dagger}\beta^{(k)}}(s_1) - V_{P,1}^{\pi^{\alpha(k)},r}[\kappa](s_1) - \overline{\lambda}^{(k,T)}V_{P,1}^{\pi^{\alpha(k)},u-2C_u\beta^{(k)}}(s_1)$$

$$= V_{P,1}^{\pi^{\alpha(k)},f^1}(s_1) + V_{P,1}^{\pi^{\alpha(k)},f^2}(s_1) + \overline{\lambda}^{(k,T)}V_{P,1}^{\pi^{\alpha(k)},2C_u\beta^{(k)}}(s_1)$$

where $f^1 : [\![1,H]\!] \times \mathcal{S} \times \mathcal{A} \to \mathbb{R}$ and $f^2 : [\![1,H]\!] \times \mathcal{S} \times \mathcal{A} \to \mathbb{R}$ are functions such that

$$f_h^1(s,a) = \sum_{a \in \mathcal{A}} \left( \pi_h^{(k)}(a \mid s)\left( \overline{Q}_{(k),h}^{\pi^{(k)},r}[\kappa](s,a) + \overline{\lambda}^{(k,T)}\underline{Q}_{(k),h}^{\pi^{(k)},u}(s,a) - \kappa \ln \pi_h^{(k)}(a \mid s) \right) \right)$$

$$- \sum_{a \in \mathcal{A}} \left( \pi_h^{\alpha(k)}(a \mid s)\left( \overline{Q}_{(k),h}^{\pi^{(k)},r}(s,a) + \overline{\lambda}^{(k,T)}\underline{Q}_{(k),h}^{\pi^{(k)},u}(s,a) - \kappa \ln \pi_h^{\alpha(k)}(a \mid s) \right) \right)$$

$$f_h^2(s,a) = \delta_{(k)}^{\pi^{(k)},r} - \overline{\lambda}^{(k,T)}\delta_{(k)}^{\pi^{(k)},u} .$$

It is well-known that the analytical maximizer of $\max_{\pi \in \mathscr{P}(\mathcal{A})} \sum_{a \in \mathcal{A}} \pi(a)(\mathbf{x}(a) - \kappa \ln \pi(a))$ is $\mathrm{SoftMax}\left(\frac{1}{\kappa}\mathbf{x}(\cdot)\right)$. Therefore, the function $f^1$ is non-negative and thus $V_{P,1}^{\pi^{\alpha(k)},f^1}(s_1) \geq 0$.

On the other hand, using Lemma 36, we have

$$f_h^2(s,a) = \delta_{(k),h}^{\pi^{(k)},r} - \overline{\lambda}^{(k,T)}\delta_{(k),h}^{\pi^{(k)},u} \overset{(a)}{\geq} -\overline{\lambda}^{(k,T)}2C_u\beta_h^{(k)}$$

Therefore, it holds that

$$V_{P,1}^{\pi^{\alpha(k)},f^2}(s_1) + \overline{\lambda}^{(k,T)}V_{P,1}^{\pi^{\alpha(k)},2C_u\beta^{(k)}}(s_1) \geq 0 .$$

By combining all the results, we have ③ $\leq 0$. $\qquad\square$

### E.5.3 Bounds for Bisection Search

Using Lemma 43, ⑤ is further bounded by

$$⑤ = \sum_{k \in \mathcal{U}^{\complement}} \overline{\lambda}^{(k,T)}\left( \underline{V}_{(k),1}^{\pi^{(k)},u}(s_1) - V_{P,1}^{\pi^{\alpha(k)},u-2C_u\beta^{(k)}}(s_1) \right)$$

$$\leq \sum_{k \in \mathcal{U}^{\complement}} \overline{\lambda}^{(k,T)}\left( \underline{V}_{(k),1}^{\pi^{(k)},u}(s_1) - b \right) \leq C_\lambda \sum_{k \in \mathcal{U}^{\complement}} \left( \underline{V}_{(k),1}^{\pi^{(k)},u}(s_1) - b \right) .$$

We bound the last term using the bisection search in Algorithm 2. Note that we focus only the case $\underline{V}_{(k),1}^{\pi^{(k),0},u}(s_1) < b$ and $\underline{V}_{(k),1}^{\pi^{(k),C_\lambda},u}(s_1) \geq b$ due to Line 4 and Line 3 in Algorithm 2. Due to the definitions of $\overline{\lambda}^{(k,t)}$ and $\underline{\lambda}^{(k,t)}$ in Algorithm 2,

$$\underline{V}_{(k),1}^{\pi^{(k)},\underline{\lambda}^{(k,t)},u}(s_1) < b \text{ and } \underline{V}_{(k),1}^{\pi^{(k)},\overline{\lambda}^{(k,t)},u}(s_1) \geq b$$

hold for any $t \in [\![1,T]\!]$. Therefore,

$$⑤ \leq C_\lambda \sum_{k \in \mathcal{U}^{\complement}} \left( \underline{V}_{(k),1}^{\pi^{(k)},\overline{\lambda}^{(k,T)},u}(s_1) - \underline{V}_{(k),1}^{\pi^{(k)},\underline{\lambda}^{(k,T)},u}(s_1) \right)$$

To bound the right-hand side, we derive the sensitivity of $V_{(k),1}^{\pi^{(k),\lambda},u}(s_1)$ with respect to $\lambda$.

**Lemma 46** (Restatement of Lemma 8). *Let $X := K\left(1 + \frac{8(1+C_\lambda)(H_\kappa + B_\dagger H + H)}{\kappa}\right)$ and $Y := \frac{8(H_\kappa + B_\dagger H + H)}{\kappa}$. For any $k$ and $\lambda \in [0, C_\lambda]$, it holds that*

$$\left| \underline{V}_{(k),1}^{\pi^{(k),\lambda},u}(s_1) - \underline{V}_{(k),1}^{\pi^{(k),\lambda+\varepsilon},u}(s_1) \right| \leq X^H H^2 Y\varepsilon .$$

*Proof.* The proof is based on **Lemma 2** from Ghosh et al. [18]. For notational simplicity, we denote $\pi := \pi^{(k),\lambda}$ and $\pi' := \pi^{(k),\lambda+\varepsilon}$. Additionally, we use shorthand:

$$v_h^r := \left\|\overline{V}_{(k),h}^{\pi,r}[\kappa] - \overline{V}_{(k),h}^{\pi',r}[\kappa]\right\|_\infty, \qquad\qquad q_h^r := \left\|\overline{Q}_{(k),h}^{\pi,r}[\kappa] - \overline{Q}_{(k),h}^{\pi',r}[\kappa]\right\|_\infty,$$

$$v_h^\dagger := \left\|\overline{V}_{(k),h}^{\pi,\dagger} - \overline{V}_{(k),h}^{\pi',\dagger}\right\|_\infty, \qquad\qquad q_h^\dagger := \left\|\overline{Q}_{(k),h}^{\pi,\dagger} - \overline{Q}_{(k),h}^{\pi',\dagger}\right\|_\infty,$$

$$v_h^u := \left\|\underline{V}_{(k),h}^{\pi,u} - \underline{V}_{(k),h}^{\pi',u}\right\|_\infty, \qquad\qquad q_h^u := \left\|\underline{Q}_{(k),h}^{\pi,u} - \underline{Q}_{(k),h}^{\pi',u}\right\|_\infty.$$

For any $h$, we have

$$v_h^r = \left\|\pi_h \overline{Q}_{(k),h}^{\pi,r}[\kappa] - \pi_h' \overline{Q}_{(k),h}^{\pi',r}[\kappa]\right\|_\infty \le H_\kappa \|\pi_h - \pi_h'\|_1 + q_h^r$$

$$v_h^\dagger \le B_\dagger H \|\pi_h - \pi_h'\|_1 + q_h^\dagger$$

$$v_h^u \le H \|\pi_h - \pi_h'\|_1 + q_h^u.$$

Since $\pi_h$ and $\pi_h'$ are softmax policies, using Lemma 19,

$$\|\pi_h - \pi_h'\|_1 \le \frac{8}{\kappa}\left\|\overline{Q}_{(k),h}^{\pi,\dagger} + \overline{Q}_{(k),h}^{\pi,r}[\kappa] + \lambda \underline{Q}_{(k),h}^{\pi,u} - \overline{Q}_{(k),h}^{\pi',\dagger} - \overline{Q}_{(k),h}^{\pi',r}[\kappa] - (\lambda+\varepsilon)\underline{Q}_{(k),h}^{\pi',u}\right\|_\infty$$

$$\le \frac{8}{\kappa}\left(q_h^\dagger + q_h^r + C_\lambda q_h^u + \varepsilon H\right)$$

Additionally,

$$q_h^r \le \left\|\widehat{P}_h^{(k)}\left(\overline{V}_{(k),h+1}^{\pi,r}[\kappa] - \overline{V}_{(k),h+1}^{\pi',r}[\kappa]\right)\right\|_\infty \le K v_{h+1}^r$$

$$q_h^\dagger \le \left\|\widehat{P}_h^{(k)}\left(\overline{V}_{(k),h+1}^{\pi,\dagger} - \overline{V}_{(k),h+1}^{\pi',\dagger}\right)\right\|_\infty \le K v_{h+1}^\dagger$$

$$q_h^u \le \left\|\widehat{P}_h^{(k)}\left(\underline{V}_{(k),h+1}^{\pi,u} - \underline{V}_{(k),h+1}^{\pi',u}\right)\right\|_\infty \le K v_{h+1}^u,$$

where we used the fact that, for any $V : \mathcal{S} \to \mathbb{R}$,

$$\left|\widehat{P}_h^{(k)} V\right|(s,a) = \left|\phi(s,a)^\top (\mathbf{\Lambda}_h^{(k)})^{-1} \sum_{i=1}^{k-1} \phi(s_h^{(i)}, a_h^{(i)}) V(s_{h+1}^{(i)})\right|$$

$$\le \left\|(\mathbf{\Lambda}_h^{(k)})^{-1} \sum_{i=1}^{k-1} \phi(s_h^{(i)}, a_h^{(i)})\right\|_2 \|V\|_\infty \le K \|V\|_\infty.$$

By combining all the results,

$$v_h^r \le K\left(\frac{8H_\kappa}{\kappa} + 1\right)v_{h+1}^r \quad + K\frac{8H_\kappa}{\kappa}v_{h+1}^\dagger \qquad\qquad +K\frac{8H_\kappa C_\lambda}{\kappa}v_{h+1}^u \qquad +\frac{8H_\kappa}{\kappa}\varepsilon H$$

$$v_h^\dagger \le K\frac{8B_\dagger H}{\kappa}v_{h+1}^r \qquad\qquad + K\left(\frac{8B_\dagger H}{\kappa} + 1\right)v_{h+1}^\dagger \qquad +K\frac{8B_\dagger HC_\lambda}{\kappa}v_{h+1}^u \quad +\frac{8B_\dagger H}{\kappa}\varepsilon H$$

$$v_h^u \le K\frac{8H}{\kappa}v_{h+1}^r \qquad\qquad + K\frac{8H}{\kappa}v_{h+1}^\dagger \qquad\qquad +K\left(\frac{8H}{\kappa} + 1\right)C_\lambda v_{h+1}^u \quad +\frac{8H}{\kappa}\varepsilon H.$$

Let $X := K\left(1 + \frac{8(1+C_\lambda)(H_\kappa + B_\dagger H + H)}{\kappa}\right)$ and $Y := \frac{8(H_\kappa + B_\dagger H + H)}{\kappa}$. Then,

$$v_h^r + v_h^\dagger + v_h^u \le X(v_{h+1}^r + v_{h+1}^\dagger + v_{h+1}^u) + YH\varepsilon$$

$$\le X^2(v_{h+2}^r + v_{h+2}^\dagger + v_{h+2}^u) + XYH\varepsilon + YH\varepsilon$$

$$\le \cdots$$

$$\le (X^H + \cdots + X + 1)YH\varepsilon.$$

$\square$

We are now ready to bound $\circled{5}$ Applying Lemma 46 to $\circled{5}$, we obtain the following lemma.

**Lemma 47.** *When $T = \widetilde{\mathcal{O}}(H)$, it holds that*

$$\circled{5} \le C_\lambda \sum_{k \in \mathcal{U}^{\complement}} \left( \overline{V}_{(k),1}^{\pi^{(k)}, \overline{\lambda}^{(k,T)}, u}(s_1) - \underline{V}_{(k),1}^{\pi^{(k)}, \underline{\lambda}^{(k,T)}, u}(s_1) \right) \le \widetilde{\mathcal{O}}(1) \ .$$

*Proof.* Due to the bisection search update rule, $\overline{\lambda}^{(k,T)} - \underline{\lambda}^{(k,T)} = 2^{-T}$. Thus,

$$\circled{5} \le C_\lambda \sum_{k \in \mathcal{U}^{\complement}} \left( \overline{V}_{(k),1}^{\pi^{(k)}, \overline{\lambda}^{(k,T)}, u}(s_1) - \underline{V}_{(k),1}^{\pi^{(k)}, \underline{\lambda}^{(k,T)}, u}(s_1) \right) \le X^H C_\lambda K H^2 Y 2^{-T}$$

where the inequality uses Lemma 46 with $X$ and $Y$ defined in Lemma 46. Thus, $\circled{5} \le \widetilde{\mathcal{O}}(1)$ holds by setting $T = H \operatorname{polylog}(X, H, Y)$. This concludes the proof. $\qquad\square$

We are now ready to prove Theorem 4. The proof is under the parameters of: $\rho = 1$, $C_r = \widetilde{\mathcal{O}}(dH)$, $C_u = \widetilde{\mathcal{O}}(dH)$, $C_\dagger = \widetilde{\mathcal{O}}(d^2 H^3 \xi^{-1})$, $B_\dagger = \widetilde{\mathcal{O}}(dH^2 \xi^{-1})$, $\kappa = \widetilde{\Omega}(\xi^3 H^{-4} d^{-1} K^{-0.5})$, $T = \widetilde{\mathcal{O}}(H)$, and $C_\lambda = \widetilde{\mathcal{O}}(dH^4 \xi^{-2})$.

### E.5.4  Proof of Theorem 4

We condition the proof with the good events $\mathscr{E}_1 \cap \mathscr{E}_2$, which holds with probability at least $1 - 3\delta$ by Lemmas 33 and 34.

In Algorithm 2, the deployed policy switches between $\pi^{\mathrm{sf}} \in \Pi^{\mathrm{sf}}$ and the softmax policies. Since Algorithm 2 deploys the softmax policies only when $\underline{V}_{(k),1}^{\pi^{(k)},0,u}(s_1) \ge b$, due to Lemma 36 and the good events, all the deployed policies satisfy $\pi^{(k)} \in \Pi^{\mathrm{sf}}$ for all $k \in [\![1, K]\!]$. This concludes the proof of the zero-violation guarantee.

Next, we derive the regret bound. Recall from Equation (17) that

$$\operatorname{Regret}(K) \le \widetilde{\mathcal{O}}(d^3 H^4 \xi^{-2}) + \circled{1} + \circled{2} + \kappa K H \ln A \le \widetilde{\mathcal{O}}(d^3 H^4 \xi^{-2}) + \circled{1} + \circled{2} + \widetilde{\mathcal{O}}(\sqrt{K}) \ ,$$

where the second inequality is due to the value of $\kappa$.

Using Equation (18),

$$\circled{1} \le \widetilde{\mathcal{O}}\left( H^2 d^{3/2} \sqrt{K} \right).$$

Using Equation (19), $\circled{2}$ can be decomposed as:

$$\circled{2} \le \circled{3} + \circled{4} + \circled{5} \ .$$

Each term can be bounded as:

- $\circled{3} \le 0$ by Lemma 45
- $\circled{4} \le \widetilde{\mathcal{O}}\left( H^4 d^{5/2} \xi^{-1} \sqrt{K} \right)$ by Equation (20),
- $\circled{5} \le \widetilde{\mathcal{O}}(1)$ by Lemma 47

Finally, by combining all the results, we have

$$\operatorname{Regret}(K) \le \widetilde{\mathcal{O}}(d^3 H^4 \xi^{-2}) + \widetilde{\mathcal{O}}\left( H^2 d^{3/2} \sqrt{K} \right) + \widetilde{\mathcal{O}}\left( H^4 d^{5/2} \xi^{-1} \sqrt{K} \right) \ .$$

This concludes the proof of the sublinear regret guarantee.

## F   Numerical Experiments

This section presents empirical results supporting Theorem 4, which guarantees $\sqrt{K}$ regret and episode-wise safety of OPSE-LCMDP. We also evaluate how often OPSE-LCMDP deploys the safe policy $\pi^{\mathrm{sf}}$, a key technique for achieving sublinear regret (Theorem 3). All experiments were conducted within 30 minutes using eight Intel Core i7 CPUs and 32 GiB of RAM.

The source code for the experiment is available at https://github.com/matsuolab/Episode-Wise-Safe-Linear-CMDP.

We compare OPSE-LCMDP against the previous state-of-the-art linear CMDP algorithm by Ghosh et al. [18] and the tabular CMDP algorithm called DOPE [8]. Ghosh et al. [18] achieves $\widetilde{\mathcal{O}}(\sqrt{K})$ bounds for both regret and violation regret, and DOPE achieves $\widetilde{\mathcal{O}}(\sqrt{K})$ regret with zero episode-wise violation.

For a sequence of policies $\{\pi^{(k)}\}_{k \in [1,K]}$, the violation regret is defined as:

$$\mathrm{Vio}(K) := \sum_{k=1}^{K} \max\left\{ b - V_{P,1}^{\pi^{(k)},u}(s_1),\ 0 \right\} . \tag{21}$$

Clearly, if all the policies satisfy $\pi^{(k)} \in \Pi^{\mathrm{sf}}$, the violation regret is zero.

Additionally, we also report the performance of a uniform policy defined by $\pi_h(\cdot \mid s) = 1/A$ for all $h, s$, to highlight the sublinear regret of our algorithm.

**Implementations of Ghosh et al. [18] and DOPE.**   Ghosh et al. [18]'s algorithm can be implemented similarly to ours, with a few modifications: remove the $\pi^{\mathrm{sf}}$ deployment trigger, eliminate the pessimism compensation bonuses by setting $C_\dagger = B_\dagger = 0$, and apply an optimistic constraint bonus instead of our pessimistic one (i.e., use a negative sign for $C_u$). We use $C_r$ and $C_u$ to denote the bonus scaling parameters for Ghosh et al. [18]. See Algorithm 1 of Ghosh et al. [18] for further implementation details.

The DOPE algorithm can be implemented in tabular environments with a moderately small state space. It computes the policy $\pi^{(k)}$ by solving the following optimistic–pessimistic problem:

$$\pi^{(k)} \in \max_{\pi \in \Pi} \max_{P' \in \mathcal{P}^{(k)}} V_{P',1}^{\pi, r + C_r \beta^{(k)}}(s_1) \text{ such that } V_{P',1}^{\pi, u - C_u \beta^{(k)}}(s_1) \geq b , \tag{22}$$

where $\beta_h^{(k)}(s,a)$ denotes the bonus at step $h$ for the state-action pair $(s,a)$ and $\mathcal{P}^{(k)}$ denotes the confidence set for the transition kernel. Specifically, using the visitation count[11] $n_h^{(k)}(s,a,s') := \sum_{k'=1}^{k} \mathbb{1}[s_h^{(k)} = s, a_h^{(k)} = a, s_{h+1}^{(k)} = s']$, the bonus and the confidence set are defined as

$$\beta_h^{(k)}(s,a) = \sum_{s' \in \mathcal{S}} \gamma_h^{(k)}(s,a,s') \text{ where } \gamma_h^{(k)}(s,a,s') \propto \sqrt{\frac{\widehat{P}_h^{(k)}(s' \mid s,a)(1 - \widehat{P}_h^{(k)}(s' \mid s,a))}{n_h^{(k)}(s,a) \vee 1}} ,$$

$$n_h^{(k)}(s,a) := \sum_{s' \in \mathcal{S}} n_h^{(k)}(s,a,s') , \text{ and } \widehat{P}_h^{(k)}(s' \mid s,a) := \frac{n_h^{(k)}(s,a,s')}{n_h^{(k)}(s,a) \vee 1} .$$

For simplicity, we omit absolute constants and logarithmic factors, and use this simplified form in all experiments. Further implementation details can be found in Bura et al. [8].

For each environment, we select the hyperparameters of each algorithm using heuristic adjustments to balance exploration and exploitation. To ensure numerical stability, we assign relatively small values to these parameters. The detailed values are provided below.

**Synthetic tabular environments.**   To evaluate the exact regret values, we conduct experiments on tabular CMDPs with a small state space size. Tabular CMDP is the special case of linear CMDP with $d = |\mathcal{S}|$ and allows us to compute the optimal policy $\pi^\star$ by linear programming.

---

[11] $\mathbb{1}[E]$ equals 1 if the event $E$ is true, and 0 otherwise. For two scalars $a$ and $b$, we use shorthand $a \vee b := \max\{a, b\}$.

We instantiated CMDPs with $|\mathcal{S}| = 5$, $|\mathcal{A}| = 3$, $H = 4$, employing a construction strategy akin to that of Dann et al. [9]. For all $s, a, h$, the transition probabilities $P_h(\cdot \mid s, a)$ were independently sampled from $\mathrm{Dirichlet}(0.1, \ldots, 0.1)$. This transition probability kernel is concentrated yet encompasses non-deterministic transition probabilities.

The reward values for the objective $r_h(s, a)$ are set to 0 with probability 0.1 and to 1 otherwise. The utility values for the constraint $u_h(s, a)$ are assigned in the same way. The initial state $s_1$ is randomly chosen from $\mathcal{S}$ and fixed during the training. The constraint threshold is set as $b = 0.6 \max_{\pi \in \Pi} V_{P,1}^{\pi,u}(s_1)$.

We choose the hyperparameters of the algorithms as follows:

- OPSE-LCMDP: $C_r = C_u = C_d = 1.0$, $C_\lambda = 300$, $B_\dagger = 1.0$, and $\kappa = 0.1$.
- Ghosh et al. [18]: $C_r = C_u = 1.0$ and $\kappa = 0.1$.
- DOPE [8]: $C_r = C_u = 1.0$.

**Media Streaming CMDP Environments.** As a realistic environment, we also evaluate algorithms on the media streaming environment from Bura et al. [8]. In the environment, a wireless base station (agent) transmits media to a device using either a fast or slow service option, each incurring different costs. The slow and fast services correspond to actions $a = 1$ and $a = 2$, respectively.

The fast service succeeds with probability $\mu_1$, and the slow one with $\mu_2 = 1 - \mu_1$, where both follow independent Bernoulli distributions. At each environment construction, we randomly sample $\mu_1$ from $[0.5, 0.9]$. Packets received at the device are stored in a media buffer and played out according to a Bernoulli process with parameter $\rho$. We sample $\rho$ uniformly from $[0.1, 0.4]$.

Let $A_h, B_h \in \{0, 1\}$ denote the number of arriving and departing packets, respectively. The media buffer length represents the state, and transitions as $s_{h+1} = \min\{\max\{0, s_h + A_h - B_h\}, L\}$ where $L$ denotes the maximum buffer length. We set $L = 5$, $|\mathcal{S}| = L + 1$, and $H = 4$. The initial state is set to $s_1 = 0$.

The objective is to deliver enough packets to the buffer while limiting the use of the fast service. Accordingly, the agent receives a reward $r_h(s, \cdot) = \mathbb{1}\{s \geq 0.3L\}$ and incurs a constraint utility $u_h(\cdot, a) = \mathbb{1}\{a = 1\}$. The constraint threshold is set as $b = 0.6 \max_{\pi \in \Pi} V_{P,1}^{\pi,u}(s_1)$.

We choose the hyperparameters of the algorithms as follows:

- OPSE-LCMDP: $C_r = C_u = C_d = 2.0$, $C_\lambda = 300$, $B_\dagger = 1.0$, and $\kappa = 0.1$.
- Ghosh et al. [18]: $C_r = C_u = 2.0$ and $\kappa = 0.1$.
- DOPE [8]: $C_r = C_u = 1.0$.

**Synthetic linear environments.** Building on the experiment by Amani et al. [4], we randomly construct linear CMDPs in which the number of states is larger than the feature map dimension. We test the algorithms on environments with $S = 100$, $A = 3$, $d = 5$, and $H = 4$. This setup has a relatively large state space while still allowing us to analytically compute the optimal policy and exact regret.

For each $(s, a) \in \mathcal{S} \times \mathcal{A}$, the feature vector $\phi(s, a) \in \mathbb{R}^d$ is sampled from $\mathrm{Dirichlet}(0.1, \ldots, 0.1)$. Recall from Assumption 2 the definition of $\boldsymbol{\mu}_h = (\boldsymbol{\mu}_h^1, \ldots, \boldsymbol{\mu}_h^d) \in \mathbb{R}^{S \times d}$. For each $(h, i) \in [\![1, H]\!] \times [\![1, d]\!]$, we sample $\boldsymbol{\mu}_h^i$ from $\mathrm{Dirichlet}(0.1, \ldots, 0.1)$. With these $\boldsymbol{\mu}$ and $\phi$, we set $P_h(s' \mid s, a) = \boldsymbol{\mu}_h(s')^\top \phi(s, a)$. This construction ensures that $P_h(\cdot \mid s, a) = 1$ becomes a valid probability distribution for any $(h, s, a)$.

For the reward and utility functions, we sample both $\boldsymbol{\theta}_h^r$ and $\boldsymbol{\theta}_h^u$ from a uniform distribution over $[0, 1]^d$. The reward and utility functions are then constructed such that $r_h(s, a) = (\boldsymbol{\theta}_h^r)^\top \phi(s, a)$ and $u_h(s, a) = (\boldsymbol{\theta}_h^u)^\top \phi(s, a)$.

The initial state $s_1$ is randomly chosen from $\mathcal{S}$ and fixed during the training. The constraint threshold is set as $b = 0.68 \max_{\pi \in \Pi} V_{P,1}^{\pi,u}(s_1)$.

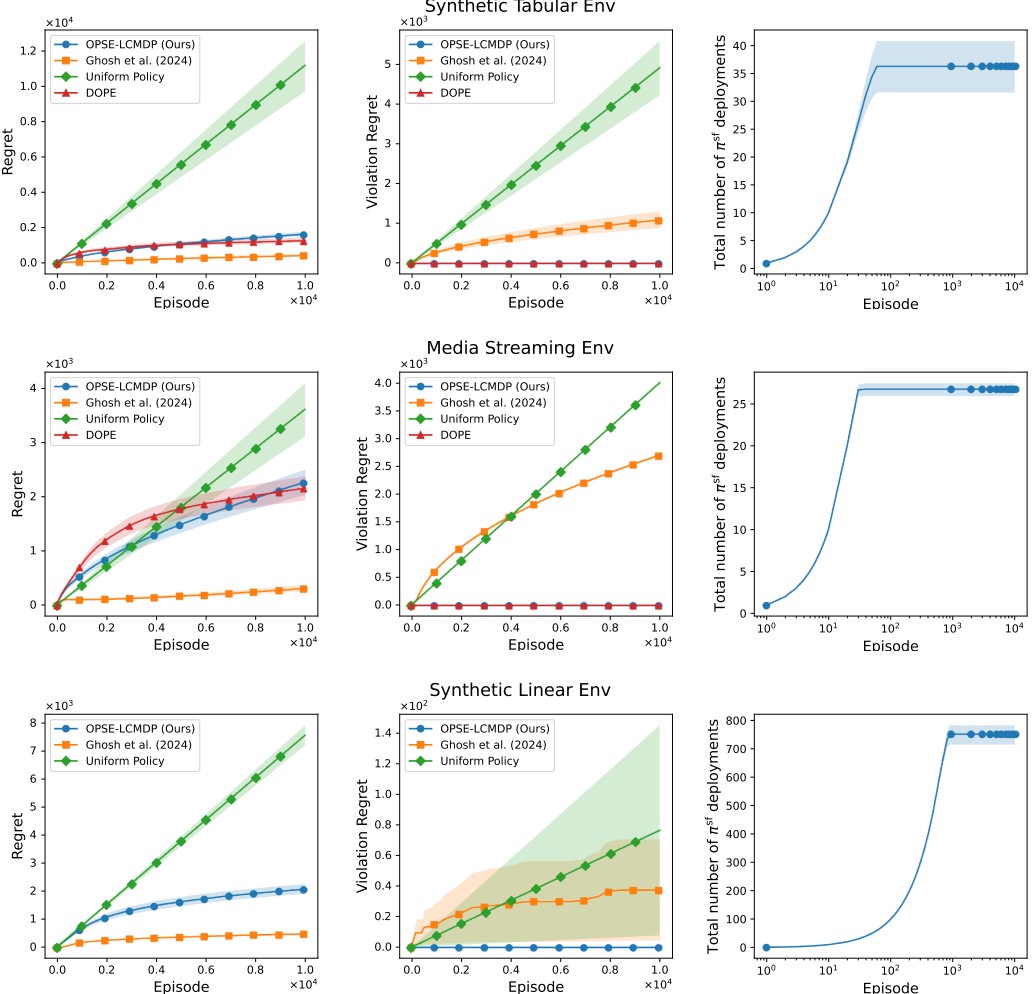

Figure 1: Numerical comparison of the algorithms in the synthetic tabular environment (**Top**), the media streaming environment (**Middle**), and the synthetic linear environment (**Bottom**). We do not run DOPE in the linear CMDP environment due to its computational intractability (see Remark 3). **Left**: regret (Equation (6)), **Middle**: violation regret (Equation (21)), and **Right**: total number of $\pi^{\mathrm{sf}}$ deployments in OPSE-LCMDP.

---

**Remark 3.** *We do not run the DOPE algorithm in this linear environment due to its heavy computational cost. Using the extended LP technique introduced by Efroni et al. [13], the optimization problem in (22) can be reformulated as the following standard LP problem:*

$$\min_{\mathbf{x}} \quad \mathbf{c}^{\top}\mathbf{x} \ \text{such that} \ A\mathbf{x} = \mathbf{b} \ \text{and} \ G\mathbf{x} \geq \mathbf{h} \,,$$

*where parameters are defined appropriately. This LP involves $HS^2A$ decision variables and more than $HS^2A$ number of constraints (see [13] for more details). Therefore, in our synthetic linear CMDP experiment, the matrices $A$ and $G$ require at least $10^{10}$ entries, which is computationally intractable in practice.*

---

**Results.** Figure 1 shows the performance of the algorithms, averaged over 10 random seeds, with regret plotted on the left, violation regret in the middle, and the total number of $\pi^{\mathrm{sf}}$ deployments on the right.

Across all settings, both OPSE-LCMDP and the algorithm by Ghosh et al. [18] exhibit sublinear regret. However, while OPSE-LCMDP maintains zero constraint violation throughout, Ghosh et al. [18] con-

sistently violates the constraint, leading to increasing violation regret. These results empirically validate Theorem 4, confirming the $\widetilde{\mathcal{O}}(\sqrt{K})$ regret and episode-wise safety guarantees of our algorithm.

While DOPE achieves sublinear regret with zero-violation, it is limited to the tabular settings where $S$ is small, as described in Remark 3. This highlights the computational tractability of our OPSE-LCMDP in large $S$, which supports Remark 2.

Finally, the right plot shows that OPSE-LCMDP explores the environment using $\pi^{\text{sf}}$ primarily during the early stages of training, and stops deploying it after approximately $10^4$ episodes. This behavior supports Theorem 3.

