# OpenReview forum: "Provably Efficient RL under Episode-Wise Safety in Constrained MDPs with Linear Function Approximation"
_NeurIPS.cc/2025/Conference — NeurIPS 2025 spotlight_

### Official Review · Reviewer_JT6p · 2025-07-01

**Clarity:** 3
**Significance:** 3
**Originality:** 3
**Rating:** 5
**Confidence:** 3

**Summary:**

The paper studies linear Constrained Markov Decision Processes (CMDPs) in the function approximation setting. Concretely, the authors present two algorithms with sublinear regret which is guaranteed to never violate the constraints, one for the bandit setting and one for the general reinforcement learning setting. This is achieved by using a known safe policy to select actions whenever the confidence bounds exceed a given threshold. Otherwise the algorithm plays a policy whose occupancy measure mixes that of the safe policy and that of the estimated best policy. The authors show that the number of episodes playing the safe policy is logarithmically bounded, ensuring sublinear regret.

**Questions:**

How does Algorithm 1 solve the optimistic-pessimistic problem? This does not seem trivial due to the constraint.

A policy with a given occupancy measure is guaranteed to exist, but how easy it is to compute such a policy?

**Ethical Concerns:**

["NO or VERY MINOR ethics concerns only"]

**Final Justification:**

The authors addressed my concerns and I am satisfied with the responses. In my view, the most important outstanding point is to add a discussion about the tractability of the optimistic-pessimistic problem.

**Limitations:**

I do not think that there are any major limitations.

**Paper Formatting Concerns:**

No concerns.

**Quality:**

3

**Strengths And Weaknesses:**

The paper is mostly well written and the authors attempt to explain all steps of the analysis. According to the authors, theirs is the first known algorithm with sublinear regret and zero violation in the linear CMDP setting. I believe the algorithm and the analysis are novel, particularly the way that the known safe policy is used to achieve a mixture policy that is guaranteed to be safe. The authors also bound the number of times that the safe policy is played on its own, which is sufficient to ensure sublinear regret.

The analysis is somewhat difficult to follow at times, though I have done my best to ensure that the proofs are correct. A couple of algorithmic concepts are not well explained, e.g. how to solve a given optimistic-pessimistic problem efficiently when constraints are involved, and how to produce a policy that matches a certain occupancy measure. However, overall I am happy with the paper and assuming that all results are correct I think the contribution is significant enough for publication at NeurIPS.

---

> ### Author Rebuttal · Authors · 2025-07-30
>
> Thank you very much for the valuable comments. We answer the question raised by the reviewer in the following. We would also be happy to provide further clarifications if suitable.
>
> ---
>
> >*How does Algorithm 1 solve the optimistic-pessimistic problem? This does not seem trivial due to the constraint.*
>
> Recall the Opt-Pes problem in the bandit setup (Equation (3)):
>
> $$
> \max_{\pi} E_{a \sim \pi} \left[f(a) \right] \text { such that } \mathbb{E}_{a \sim \pi} \left[g(a) \right] \geq b\;,
> $$
> where we simplified the notations to render the equation with Markdown properly.
>
> This is an optimization problem with a linear objective and a constraint.
> In general, not much can be said about the computational tractability without assumptions on $\mathcal{A}$.  **When $\mathcal{A}$ is finite, which is the focus of the linear CMDP section (Section 3), in this case, the problem reduces to a convex optimization problem and can be solved efficiently using linear programming, the Lagrangian method, or our softmax-policy approach with the bisection search (Section 3.2).**
>
> Since Section 2 aims to illustrate our $\pi^{\text{sf}}$ deployment trigger technique and its role in efficient exploration, we omit computational details and focus on the core ideas behind the regret analysis.
>
> We also note that the optimistic problem becomes computationally challenging beyond the finite action setting [Lattimore 2020]. Specifying what condition on $\mathcal{A}$ makes the Opt-Pes problem in the linear bandit efficiently solvable is beyond the scope of Section 2. We will add the above discussion to the final version.
>
>
> - [Lattimore 2020]: [Bandit Algorithms Section 19.3.1](https://tor-lattimore.com/downloads/book/book.pdf#page=254)
>
> ---
>
> >*A policy with a given occupancy measure is guaranteed to exist, but how easy it is to compute such a policy?*
>
> We believe the reviewer is referring to the discussion around Lemma 7. In our algorithm, we do not need to compute the policy of the corresponding mixed occupancy measure.
>
> Lemma 7 ensures the existence of a policy $\pi$ that is optimistic while satisfying the pessimistic constraint such that
>
> $$
> V_{P, 1}^{\pi, r+B_{\dagger} \beta^{(k)}}\left(s_1\right) \geq V_{P, 1}^{\pi^{\star}, r}\left(s_1\right)
> \quad \text{and}\quad
> V_{P, h}^{\pi, u-2 C_u \beta^{(k)}}  \geq b\;.
> $$
> As we show in Lemma 7, the existence is ensured by the technique of mixed occupancy measures. We use this mixture policy $\pi$ to establish the regret upper bound for term ② around line 264. *Notably, this mixture policy appears only in the regret analysis of $\pi^{(k)}$ obtained via bisection search; it is not required in the actual algorithm.*

---

### Official Review · Reviewer_R4du · 2025-07-03

**Clarity:** 3
**Significance:** 2
**Originality:** 2
**Rating:** 4
**Confidence:** 4

**Summary:**

This paper studies constrained RL problem where it maximizes the reward subject to a constraint. The settings considered in this paper are episode-wise safety and linear function approximation. It designs a computationally efficient algorithm and proves that with high probability the algorithm incurs $\tilde{O}(\sqrt{K})$ regret with an episode-wise zero-violation guarantee.

**Questions:**

Could you be more specific about the claim on “while safe exploration is well-established in tabular CMDPs, extending it to large-scale CMDPs remains a major challenge”? Given existing work with optimistic–pessimistic exploration techniques and safe algorithms with linear function approximation, could you articulate the specific challenge your paper resolves? This is a solid work but the novelty is not yet clear to me.

Some minor suggestions:

Providing a table of content would improve readability.

A small typo in Assumption 2: \mu_h(s’) should be replaced with (\mu_h)

**Ethical Concerns:**

["NO or VERY MINOR ethics concerns only"]

**Final Justification:**

The authors' response addresses my major concerns therefore I increased the score from 3 to 4.

**Limitations:**

yes

**Quality:**

3

**Strengths And Weaknesses:**

Strengths:

While prior works established such episode-wise safety with sublinear regret in tabular settings, extending these guarantees to function approximation remained open. Previous work use LP based methods which is computationally inefficient with large state space. This work design and analyze an algorithm which is computationally efficient under the linear setting.

Weaknesses:

-Although the regret is optimal in its dependence on K, the dependence on both d and H deviates a lot from the optimal dependence. Given existing results on episode-wise constrained RL and hard-constrained linear RL, the main advance seems to be computational rather than statistical (since it sacrifices on the sample complexity). Although the authors mentioned that adapting Bernstein-type bonus can improve it, but this improvement (H) is not enough to cover the loss.

-There are many existing work on constrained/safe RL with good convergence rates. Although the episode-wise safety setting considers a stricter case, it relies on an unrealistic assumption on having access to a known baseline policy that is strictly safe. Note that this assumption is not common in other constrained RL settings. This limits the practical impact of the guarantees.

-This paper studies linear function approximation setting, which is common for large state-action space RL problem. However, the empirical experiments are done in a small size CMDP.

---

> ### Author Rebuttal · Authors · 2025-07-30
>
> Thank you very much for the valuable comments. We answer the weaknesses and questions raised by the reviewer. We would also be happy to provide further clarifications if suitable.
>
> ## Weaknesses
>
> >*Although the regret is optimal in its dependence on K, the dependence on both d and H deviates a lot from the optimal dependence. Given existing results on episode-wise constrained RL and hard-constrained linear RL, the main advance seems to be computational rather than statistical (since it sacrifices on the sample complexity). Although the authors mentioned that adapting Bernstein-type bonus can improve it, but this improvement (H) is not enough to cover the loss..*
>
>
> * The reviewer appears to be mistaken on this point. **We note that none of the existing safe CMDP literature achieves episode-wise safety in linear CMDPs, which is the main contribution of our paper.** Prior works that achieve regret bounds typically consider relaxed notions of episode-wise safety or impose instantaneous constraints instead. As a result, our regret bounds are not directly comparable to existing results. We will add the discussion in the final version.
>
> * More broadly, regret or sample complexity bounds under different safety requirements are generally not comparable, even if the problem settings appear similar. For instance, [Vaswani 2022] shows that the sample complexity lower bound for tabular CMDPs scales as $(1-\gamma)^{-3}$ when small constraint violations are allowed, but increases to $(1-\gamma)^{-5}$ when a strictly safe policy is required. This highlights how even a slight change in safety requirements can significantly impact the theoretical lower bounds.
>
> * Deriving a matching lower bound for episode‑wise safety in linear CMDPs would be the next logical step toward understanding the tightness of our upper bound; we leave this important direction to future work. We will mention the above discussion in the final version.
>
> * [Vaswani 2022] Near-optimal sample complexity bounds for constrained MDPs
>
> >*There are many existing work on constrained/safe RL with good convergence rates. Although the episode-wise safety setting considers a stricter case, it relies on an unrealistic assumption on having access to a known baseline policy that is strictly safe. Note that this assumption is not common in other constrained RL settings. This limits the practical impact of the guarantees.*
>
> * **The assumption of a strictly safe policy is, in fact, necessary for episode-wise or instantaneously safe exploration.** [Pacchiano 2021] shows that without strictly safe policies, the regret lower bound becomes infinite, indicating that a strictly safe policy is necessary for efficient exploration.
> * Additionally, in practice, it is common to have a strictly safe policy. For instance, in many real-world decision-making systems, simply remaining idle or taking no action constitutes a strictly safe policy. A self-driving car that always remains stationary, or a dialogue system that remains silent, cannot cause harm to humans.
>
> * [Pacchiano 2021]: [Stochastic Bandits with Linear Constraints](https://proceedings.mlr.press/v130/pacchiano21a.html)
>
>
> For further reference, the following papers assume the existence of a strictly safe policy:
>
> * Linear CMDP with instantaneous safety: [Provably Efficient RL for Linear MDPs under Instantaneous Safety Constraints in Non-Convex Feature Spaces](https://arxiv.org/abs/2502.18655)
> * Tabular CMDP with episode-wise safety: [DOPE: Doubly Optimistic and Pessimistic Exploration for Safe Reinforcement Learning](https://arxiv.org/abs/2112.00885)
> * Linear bandit with constraint: [Stochastic Bandits with Linear Constraints](https://proceedings.mlr.press/v130/pacchiano21a.html)
>
>
> >*This paper studies linear function approximation setting, which is common for large state-action space RL problem. However, the empirical experiments are done in a small size CMDP.*
>
> * In response to the comments by fTDU and R4du about the lack of a *linear* CMDP experiment, we are conducting an experiment on linear CMDPs based on [Amani 2021], but adapted to the episode-wise safety rather instantaneous constraint. Specifically, we randomly construct linear CMDPs in which the number of states is larger than the feature map dimension. Currently, we are doing experiments with $|\mathcal{S}| = 300$ and $d = 20$, but the detailed parameters may change by the camera-ready version. This setup has a relatively large state space while still allowing us to analytically compute the optimal policy and exact regret. With additional adjustments to the experimental setup, we plan to include the results in the camera-ready version. *We also note that several fundamental linear CMDP papers (e.g., [Ghosh 2022, 2024]) achieve empirical results only for the tabular settings*.  We thus only include empirical results for the tabular settings for comparison.
> * We again remark that our main contribution is theoretical. **This is the first linear CMDP algorithm that establishes sublinear regret while being episodewise-safe**. In order to achieve that, we have to apply novel tools and techniques, as described in Section 2.1 and 3.1. Having said that, for completeness, we are conducting the experiments described above, which we will add to the final version.
>
> * [Amani 2021] Safe Reinforcement Learning with Linear Function Approximation
> * [Ghosh 2022] Provably Efficient Model-Free Constrained RL with Linear Function Approximation
> * [Ghosh 2024] Towards Achieving Sub-linear Regret and Hard Constraint Violation in Model-free RL
>
> ## Questions
>
> >*Could you be more specific about the claim on “while safe exploration is well-established in tabular CMDPs, extending it to large-scale CMDPs remains a major challenge”? Given existing work with optimistic–pessimistic exploration techniques and safe algorithms with linear function approximation, could you articulate the specific challenge your paper resolves? This is a solid work but the novelty is not yet clear to me.*
>
> Our paper highlights the technical challenges in the following sections (We will also add this in the final version):
>
> * **Section 2.1 (Technical Challenge: Zero-Violation with a Safe Policy)**: Here, we discuss why the techniques for instantaneously safe linear CMDPs are difficult to apply to our episode-wise safety setting.
>   * Basically, the instantaneous safe algorithms utilizes the vector representation of actions and assume $a = \text{[zero vector]}$ as the safe action. However, extending this safe action technique to episode-wise safe RL is non-trivial, as the episode-wise constraint is imposed on policies rather than actions, and policies in linear CMDPs may be nonlinear functions (e.g., softmax mappings from value functions) rather than single vectors. Further, as we discussed in response to Reviewer YdzS [here](https://openreview.net/forum?id=rgoSyTCTkn&noteId=1PsVpAwIQD), only satisfying instantaneous constraints can be sub-optimal for the episode-wise safety.
>
> * **Section 3.1 (Technical Challenge: Optimistic-Pessimistic Optimization in Linear CMDP)**: This section explains why the optimistic-pessimistic (Opt-Pes) exploration technique in linear CMDPs is more difficult than the tabular setting.
>     * In the tabular CMDP setting, the Opt-Pes problem can be efficiently solved using linear programming. However, in linear CMDPs with large state spaces, linear programming becomes computationally expensive and thus is inapplicable.
>     * Given that the Opt-Pes problem is a constrained problem, an alternative approach is to apply the Lagrangian method, which reformulates the constrained problem as a min-max optimization. **However, using this approach for the Opt-Pes problem is non-trivial.** Indeed, to the best of our knowledge, no existing work applies the Lagrangian method to solve the Opt-Pes problem. The main difficulty lies in the clipping operators within the Opt-Pes formulation (see Definition 2), which complicate establishing min-max duality in the Lagrangian framework. This, in turn, makes it difficult to derive performance guarantees.
>     * We address these challenges by introducing a novel adaptation of the recent composite softmax policy technique [Ghosh 2024; see our Definition 3], and by efficiently balancing the optimistic and pessimistic terms inside the softmax via bisection search.
>
> [Ghosh 2024]: Towards achieving sub-linear regret and hard constraint violation in model-free rl
>
> **Minor suggestions**
>
> > Providing a table of content would improve readability.
>
> Due to the page limitation, our paper provides the table of contents in the beginning of Appendix (page 13).

---

> > ### Author Response · Authors · 2025-08-05
> >
> > Dear Reviewer R4du,
> >
> > Since we have not heard back from you, we just wanted to check in and ask if the rebuttal clarified and answered the questions raised in your review. We would be very happy to engage further if there are additional questions!

---

> > > ### Comment · Reviewer_R4du · 2025-08-06
> > >
> > > Thank you for your response. My concerns are addressed, and I will increase my score. I am now convinced by the assumption of a strictly safe policy, and I believe this will be a good work once the technical challenges are more clearly highlighted and the use of small-scale MDPs in the experimental setup is better explained. Lastly, sorry I meant including a table of notations would be better (as the notations are tedious), not a table of contents.

---

> > > > ### Author Response · Authors · 2025-08-06
> > > >
> > > > Thank you for increasing the score and appreciating our contributions. We will definitely add the discussion in the final version. We have a mathematical notation paragraph on page 2. We will add a table in the Appendix in the final version explicitly stating all the notations for better readability.

---

### Official Review · Reviewer_fTDU · 2025-07-03

**Clarity:** 3
**Significance:** 3
**Originality:** 4
**Rating:** 5
**Confidence:** 3

**Summary:**

This paper investigates policy optimization methods where the policy must satisfy the constraint for all iterations in a linear CMDP.
Given access to a policy that strictly satisfies the constraints, the author proposed a method that achieves $O(\sqrt{d^5 H^8 K}$ regret with high probability. To achieve this, the author add standard exploration bonuses to the rewards and utilities but provides a novel condition when to switch to $\pi^{\text{sf}}$. Additionally, the authors conduct experiments in the tabular setting to support their theoretical results.

**Questions:**

1. Is it a strict requirement that the initial state $s_1$ is fixed, or can the results be extended when $s_1$ may be chosen adversarially?
2. In Figure 1, the average number of $\pi^{\text{safe}$ deployments appears consistent across different random seeds initially, but then begins to diverge around the midpoint of training as shown by the widening shaded confidence intervals (I'm guessing that this is what the shaded regions depict). What could explain this pattern of initial convergence followed by increasing variance between seeds?"
3. In Line 4 of Algorithm 1, how it be guaranteed to a $\pi^{k}$ to always exist? It wasn't clear to me why Lemma 24 implies this fact.

**Ethical Concerns:**

["NO or VERY MINOR ethics concerns only"]

**Final Justification:**

I recommend to accept this paper. I'm happy with the promise of linear CMDP experiments. Although, they are using episode-wise safety rather instantaneous constraints, it seems common in prior work to only conduct experiments in the tabular setting. All other questions were addressed during the rebuttal.

**Limitations:**

Yes, the authors sufficiently address the limitations.

**Quality:**

4

**Strengths And Weaknesses:**

**Strengths**
1. This paper provide solid theoretical results for a challenging problem instance and even recover similar regret bounds to prior work in the absence of constraints.
2. The paper is well organized, and it was to first consider the problem in a simpler bandit setting before moving to CMDPs.
3. The empirical experiments validate the theoretical results on (a bit simplistic) environments.

**Weakness**
1. While tabular CMDPs are linear CMDP,  it would be great to see experiments for linear CMDP. Otherwise, it would be interesting to see how the OPLB-SP compares to DOPE [1] in the tabular setting. I would strength my score for either addition.
2. It seems like the parameters of Algorithm 2 appear to be specified only up to big-O notation. It would be helpful to clarify what aspects of the analysis prevents the exact values to be known.

[1] Bura, A., HasanzadeZonuzy, A., Kalathil, D., Shakkottai, S., & Chamberland, J. F. (2022). DOPE: Doubly optimistic and pessimistic exploration for safe reinforcement learning. Advances in neural information processing systems, 35, 1047-1059.

---

> ### Author Rebuttal · Authors · 2025-07-30
>
> Thank you very much for the valuable comments.
> We answer the weaknesses and questions raised by the reviewer. We would also be happy to provide further clarifications if suitable.
>
> ---
>
> **Weakness**
>
> >*While tabular CMDPs are linear CMDP, it would be great to see experiments for linear CMDP. Otherwise, it would be interesting to see how the OPLB-SP compares to DOPE [1] in the tabular setting. I would strength my score for either addition.*
>
> Thank you for the suggestion. In response to the comments by fTDU and R4du about the lack of a *linear* CMDP experiment, we are conducting an experiment on linear CMDPs based on [Amani 2021], but adapted to the episode-wise safety rather instantaneous constraint.
> * Specifically, we randomly construct linear CMDPs in which the number of states is larger than the feature map dimension. Currently, we are doing experiments with $|\mathcal{S}| = 300$ and $d = 20$, but the detailed parameters may change by the camera-ready version. This setup has a relatively large state space while still allowing us to analytically compute the optimal policy and exact regret. With additional adjustments to the experimental setup, we plan to include the results in the camera-ready version. *We also note that several fundamental linear CMDP papers (e.g., [Ghosh 2022, 2024]) achieve empirical results only for the tabular settings*.  We thus only focus on tabular settings for comparison with the state-of-the-art approaches for empirical results.
>
> - [Amani 2021] Safe Reinforcement Learning with Linear Function Approximation
> - [Ghosh 2022] Provably Efficient Model-Free Constrained RL with Linear Function Approximation
> - [Ghosh 2024] Towards Achieving Sub-linear Regret and Hard Constraint Violation in Model-free RL
>
>
> Regarding the comparison with DOPE, during the rebuttal period, we have conducted an experiment under the same setting and parameters as the DOPE paper (i.e., the streaming environment).
> * **(Results)** Our algorithm achieves sublinear regret  $(\tilde{\mathcal{O}}(\sqrt{K}))$   with zero constraint violations, similar to Figure 2 in the DOPE paper. After $4\times 10^5$ episodes, our algorithm obtains a regret value of approximately $4 \times 10^5$, whereas DOPE incurs around $6\times 10^5$ regret. **This suggests that our algorithm achieves a lower regret than DOPE in this experiment.** Additionally, we remark that DOPE needs to solve an LP which can be computationally intensive for large state, whereas our approach does not.
> * For completeness, our camera-ready version will include an additional comparison with DOPE in the tabular settings.
>
> ---
>
> >*It seems like the parameters of Algorithm 2 appear to be specified only up to big-O notation. It would be helpful to clarify what aspects of the analysis prevents the exact values to be known.*
>
> * The reason is due to our careful analysis of the covering arguments and Good Events.
> * In Appendix D.2, we derive covering numbers for several function classes, starting with the Q-function classes $\mathcal{Q}^r, \mathcal{Q}^u, \mathcal{Q}^\dagger$ (Definition 10). These results are then extended to the composite function class $\mathcal{Q}^\circ$ (Definition 11), and subsequently to the policy class $\widetilde{\Pi}$ (Definition 12). These covering numbers play a crucial role in the detailed analysis of Good Events in Appendix D.3.
> * Specifying exact coefficients would make the analysis significantly more cumbersome and prone to proof errors, as even minor mistakes could propagate and distort the final results. We ensure that the analysis remains both correct and readable using the big-O notation defined in line 62. This is also a standard practice in literature. Nevertheless, the exact constants can be found in Appendix D.
>
> **Questions**
>
> >*Is it a strict requirement that the initial state $s_1$ is fixed, or can the results be extended when $s_1$ may be chosen adversarially?*
>
> * *The initial state need not be fixed, and can be drawn from some probability distribution.*
> * Extending our setting to an adversarial one is an interesting direction for future work. However, this extension is non-trivial, as our core techniques rely on the assumption of a fixed initial state.
> * For example, we control the number of safe policy deployments by evaluating the return of the bonus function $V^{\pi^{\text{sf}}, \beta^{(k)}}_{P, 1}(s_1)$ (see Definition 4), which depends on the initial state $s_1$. We bound the number of $\pi^{\text{sf}}$ deployments using Theorem 3.
> Similarly, the existence analysis of optimistic-pessimistic (Opt-Pes) policies established in Lemma 7 depends on the initial state $s_1$.
> * To handle adversarial initial states, these analyses would need to be extended to account for adversarial $s_1$ choices. Doing so poses non-trivial technical challenges and is beyond the scope of this work.
>
> >*In Figure 1, the average number of $\pi^{\text{safe}}$ deployments appears consistent across different random seeds initially, but then begins to diverge around the midpoint of training as shown by the widening shaded confidence intervals (I'm guessing that this is what the shaded regions depict). What could explain this pattern of initial convergence followed by increasing variance between seeds?"*
>
> * Since the agent has no prior knowledge of the environment, it initially deploys the safe policy for all random seeds (note that the $x$-axis in Figure 1 is on a logarithmic scale). After a sufficient number of episodes, the agent gathers enough information to switch from the safe policy to the trained policy. The exact timing of this transition may vary slightly across random seeds, resulting in the observed divergence.
>
> >*In Line 4 of Algorithm 1, how it be guaranteed to a $\pi^k$ to always exist? It wasn't clear to me why Lemma 24 implies this fact.*
>
> Recall from Definition 1 that $\mathcal{U}^c$ denotes the set of iterations when line 4 is triggered, i.e., $\mathcal{U}:=\\{k \mid C_u \beta_{\pi^{\text{sf}}}^{(k)}>\xi / 2 \\}$. Our Lemma 2 (or Lemma 24) together with Lemma 1 ensures that, when line 4 is triggered (i.e., $k \in \mathcal{U}^c$), at least the safe policy $\pi = \pi^{\mathrm{sf}}$ satisfies the constraint:
>
>
> $$ \hat{u}\_{\pi} - C_u \beta_{\pi} \geq u_{\pi}-2 C_u \beta_{\pi} \geq b  $$
>
> Here, the first constraint is due to Lemma 1 and the second constraint is due to Lemma 2. Therefore, whenever line 4 is triggered, line 4 is guaranteed to have a feasible solution and thus $\pi^k$ exists.

---

> > ### Comment · Reviewer_fTDU · 2025-08-04
> >
> > Thank you for response. I decided to increase my score to accept.

---

> > > ### Author Response · Authors · 2025-08-05
> > >
> > > We thank the reviewer for increasing the score and appreciating our contributions.

---

### Official Review · Reviewer_YdzS · 2025-07-05

**Clarity:** 2
**Significance:** 3
**Originality:** 3
**Rating:** 4
**Confidence:** 5

**Summary:**

This paper investigates the constrained MDP problems with zero-violation guarantee. The main contribution are the algorithms for linear bandit and linear CMDPs based on the linear function approximation and the optimism-pessimism exploration method. The novelty of the algorithm lies in the a new deployment rule and the composite softmax policy. The algorithms are also computationally efficient, achieving a polynomial computational cost in parameters and independent of the state space. The authors further prove that the algorithm is the first result achieving zero-violations and sub-linear $\tilde O(\sqrt{K})$ regret.

**Questions:**

Can authors add the comparison with Roknilamouki's algorithm? It seems that their algorithm has similar theoretical results and also zero-violation.

**Ethical Concerns:**

["NO or VERY MINOR ethics concerns only"]

**Final Justification:**

I appreciate the authors' efforts in the rebuttal and they have addressed my concerns. I think this paper should be accepted, although it has some limitations

**Limitations:**

See the Weakness in **Strengths And Weaknesses** part.

**Quality:**

3

**Strengths And Weaknesses:**

**Strengths**:
1. The algorithm is novel and is the first result that achieving $\tilde O(\sqrt{K})$ regret and zero-violation.
2. The algorithm achieves the polynomial computational cost on the parameters and is independent of state-space.
3. Numerical experiments further validate the performance of the algorithms.

**Weaknesses**:
1. The algorithm is only suitable for the one-constraint setting; there are often multiple constrains in real problems
2. The computational time is still dependent on action space, and the algorithm will still be computationally inefficient when meets the large or infinite action space; Bisection search may also time-consuming when the space is large.
3. The regret has higher dependence on parameters $d,H$ compared to other similar algorithms.
4. The paper could be clearer; it's a bit messy

---

> ### Author Rebuttal · Authors · 2025-07-30
>
> Thank you very much for the valuable comments. We here answer the weaknesses and the question raised by the reviewer. We would also be happy to provide further clarifications if suitable.
>
> ## Weaknesses
>
> >*The algorithm is only suitable for the one-constraint setting; there are often multiple constrains in real problems*
>
> * **We remark that even under a single constraint, no prior work achieves episode-wise safety in linear CMDPs.** Even in the single constrained setting, existing approaches are inapplicable to ensure episode-wise safe exploration as discussed in Sections 2.1 and 3.1.
> * Note that one can certainly apply grid search for multiple constraints to search over multiple dual-variables; of course, the computational complexity will increase as the binary search approach might not work.
>
> * Whether our single-constraint limitation is fundamental or merely a consequence of current techniques remains an open question. Given that even slight changes to the CMDP formulation can render the CMDP problem computationally intractable [Feinberg 2000], it is possible that this limitation is inherent to the episode-wise safe linear CMDP setting.
> Addressing this open question requires further research, which is beyond the scope of this paper.
>
> * [Feinberg 2000] Constrained Discounted Markov Decision Processes and Hamiltonian Cycles
>
> ---
>
> >*The computational time is still dependent on action space, and the algorithm will still be computationally inefficient when meets the large or infinite action space; Bisection search may also time-consuming when the space is large.*
>
> * To discuss computationally efficient algorithms for continuous action spaces, we need to make careful assumptions about the structure of the action space. Indeed, even in the unconstrained bandit setting, solving the optimistic problem becomes non-trivial when moving beyond finite action spaces [see, e.g., Lattimore 2020].
> * **Since episode-wise safe exploration is already a non-trivial challenge in the finite action setting, introducing additional assumptions for continuous actions would further complicate the regret analysis.**
> Moreover, much of the existing literature on linear CMDPs also focuses on finite action spaces [e.g., Ghosh 2024; Ding 2021; Ghosh 2022]. For these reasons, the continuous action setting is beyond the scope of this paper. We will add this discussion in the final version.
>
> * [Lattimore 2020]: [Bandit Algorithms Section 19.3.1](https://tor-lattimore.com/downloads/book/book.pdf#page=254)
> * [Ghosh 2024] Towards Achieving Sub-linear Regret and Hard Constraint Violation in Model-free RL
> * [Ding 2021] Provably Efficient Safe Exploration via Primal-Dual Policy Optimization
> * [Ghosh 2022] Provably Efficient Model-Free Constrained RL with Linear Function Approximation
>
> ---
>
> >*The regret has higher dependence on parameters $d,H$ compared to other similar algorithms.*
>
> * We note that none of the existing safe CMDP literature achieves episode-wise safety in linear CMDPs, which is the main contribution of our paper. **As a result, our regret bounds are not directly comparable to existing results.**
> * More broadly, regret or sample complexity bounds under different safety requirements are generally not comparable, even if the problem settings appear similar. For instance, [Vaswani 2022] shows that the sample complexity lower bound for tabular CMDPs scales as $(1-\gamma)^{-3}$ when small constraint violations are allowed, but increases to $(1-\gamma)^{-5}$ when a strictly safe policy is required. This highlights how even a slight change in safety requirements can significantly impact the theoretical lower bounds.
> * Deriving a formal regret lower bound for our setting would be necessary to assess the tightness of our result, but this is beyond the scope of the current paper. While that analysis lies outside the scope of this paper, we will highlight it as an important direction for future work in the final version.
>
> * [Vaswani 2022] Near-optimal sample complexity bounds for constrained MDPs
>
>
> ## Question
>
> >*Can authors add the comparison with Roknilamouki's algorithm? It seems that their algorithm has similar theoretical results and also zero-violation.*
>
> We think that the Roknilamouki's algorithm refers to this paper: [Provably Efficient RL for Linear MDPs under Instantaneous Safety Constraints in Non-Convex Feature Spaces](https://arxiv.org/abs/2502.18655).
>
> Their algorithm is designed for **instantaneous safety constrained** settings. Specifically, the instantaneous constraint considers that, for any step $h \in [H]$ and episode $k \in [K]$,
>
> $$
> \text{(Instantaneous Constraint)} \quad u_h(s_h^{(k)}, a_h^{(k)}) \geq b
> $$
>
> where $u_h$ denotes the utility function for the safety constraint. On the other hand, we consider the **episode-wise constrained setting**. For any episode $k \in [K]$, it requires that
> $$
> \text{(Episode-wise Constraint)} \quad V^{\pi^{(k)}, u}_{P, 1}(s_1) \geq b
> $$
> As we show in Appendix A.2, **the instantaneous constraint is a special case of the episode-wise constraint.** Therefore, our algorithm can handle a more general constraints compared to Roknilamouki's algorithm.
>
> Additionally, we remark that the instantaneous constraint can be overly conservative. For example, consider designing a drone controller that must avoid depleting its battery during an episode:
> * The instantaneous constraint restricts the controller from consuming more than a fixed percentage of the battery at each control step (e.g., 1% per minute).
> * In contrast, the episode-wise constraint only requires that the total battery consumption does not exceed 100% over an entire episode. This allows the controller to use more than 1% per minute as long as the overall limit is respected.
>
> Furthermore, while Roknilamouki's algorithm requires the set of features $\{\phi(s, a) \mid a \in \mathcal{A}\}$ to be star-convex (see their Assumption 3.1), **our algorithm does not require the star-convexity**. In addition, our **safe policy** assumption (Assumption 1) is weaker than their **safe action** assumption (Assumption 2.3) which assumes that there exists a safe action $a_s$ for each state $s$ such that $u_h(s, a_s) \geq b$.
>
> In summary, our algorithm is more general than Roknilamouki's algorithm for the following reasons:
> 1. $\text{Instantaneous Constraint (Roknilamouki)} \subset \text{Episode-wise Constraint (Ours)}$. Moreover, the instantaneous constraint can be overly conservative.
> 2. Our method does not require the star-convexity assumption on the feature map.
> 3. Our safe policy assumption (Assumption 1) is weaker than their safe action assumption.

---

### Note · Authors · 2025-08-12

We thank all the reviewers for their valuable feedback. We are glad that our contributions have been positively appreciated by all the reviewers.  Reviewers fTDU and R4du explicitly mentioned that our rebuttals have resolved their questions; the reviewers JT6p, and YdzS have sent Mandatory Acknowledgements, implicitly indicating that our rebuttals have satisfactorily resolved their questions.  Below, we restate our contributions and outline the planned revisions for the camera-ready version.

**Contribution.** This work presents the first statistically and computationally efficient **episode-wise safe** exploration algorithm in the **linear** CMDP setting. Previous results have been limited to **tabular** or **instantaneous-safety** settings, which either become inefficient for large state spaces or are too conservative to handle certain safety requirements (see our drone control example in the [rebuttal to YdzS](https://openreview.net/forum?id=rgoSyTCTkn&noteId=1PsVpAwIQD)). Our method thus paves the way for safe decision-making in large-scale problems, achieving a result that no existing safety algorithm has been able to match (see Introduction for details).

**Technical Challenges and Contributions (Sec. 2.1 and 3.1).** Achieving episode-wise safety with large state spaces in CMDPs is highly non-trivial. Common approaches in tabular CMDPs, such as linear programming and Lagrangian methods, are difficult to apply in this regime (Sec. 3.1). Likewise, the vectorized-action approach widely used in instantaneous-safety settings does not readily extend to the episode-wise case (Sec. 2.1). We address these challenges with two key techniques: (i) efficiently balancing weights in the softmax policy (Sec. 3) and (ii) introducing a novel safe-policy deployment condition (Sec. 2).

**Camera-ready revision.** In response to comments from reviewers fTDU and R4du, we will add experiments in the linear CMDP setting where the number of states is larger than the feature map dimension. We also plan to include our improved performance over the DOPE algorithm, which achieves episode-wise safety in the tabular setup. As discussed with YdzS, we will add clarifying statements noting that existing regret results are incomparable to our result due to the differences in the problem setting. Finally, following JT6p’s suggestion, we will elaborate on how the OptPes problem in Algorithm 1 can be solved and explain why computing the occupancy mixture policy is unnecessary.

---

### Decision · Program_Chairs · 2025-09-17

**Decision:**

Accept (spotlight)

**Comment:**

This work introduces the first algorithm achieving both sublinear regret and zero-constraint violation in the linear CMDP setting, extending guarantees beyond the tabular case. As noted by all reviewers, the analysis is both novel and rigorous. The paper is clearly written and well structured, beginning with the bandit case before moving to CMDPs, which provides readers with an intuitive understanding of the analysis. Experiments (Appendix E), though conducted on relatively simple environments, support the theoretical claims and validate the algorithm’s effectiveness. I am happy to recommend acceptance of this work to NeurIPS.